# Early season N₂O emissions under variable water management in rice systems: source-partitioning emissions using isotope ratios along a depth profile

5    Elizabeth Verhoeven[1,2], Matti Barthel[1], Longfei Yu[3], Luisella Celi[4], Daniel Said-Pullicino[4], Steven Sleutel[5], Dominika Lewicka-Szczebak[6], Johan Six[1], Charlotte Decock1[7]

[1]Department of Environmental Systems Science, ETH Zurich, 8092 Zurich, Switzerland
[2]Department of Crop and Soil Science, Oregon State University, 97331 Corvallis, Oregon, USA
[3] Department of Air Pollution and Environmental Technology, EMPA, 8600 Dübendorf, Switzerland
10   [4]Department of Agricultural, Forest and Food Sciences, University of Turin, 10095 Grugliasco, Italy
[5]Department of Soil Management, Faculty of Bioscience and Engineering, 9000 Ghent University, Belgium
[6]Thünen Institute of Climate-Smart Agriculture, 38116 Braunschweig, Germany
[7]Department of Natural Resources Management and Environmental Sciences, California State University, 93407 San Luis Obispo, California, USA

*Correspondence to*: Elizabeth Verhoeven (elizabeth.verhoeven@gmail.com)

**Abstract.** Soil moisture strongly affects the balance between nitrification, denitrification and N₂O reduction and therefore the nitrogen (N) efficiency and N losses in agricultural systems. In rice systems, there is a need to improve alternative water management practices, which are designed to save water and reduce methane emissions, but may 20   increase N₂O and decrease nitrogen use efficiency. In a field experiment with three water management treatments, we measured N₂O isotope ratios of emitted and pore air N₂O ($\delta^{15}N$, $\delta^{18}O$ and site preference, *SP*) over the course of six weeks in the early rice growing season. Isotope ratio measurements were coupled with simultaneous measurements of pore water $NO_3^-$, $NH_4^+$, dissolved organic carbon (DOC), water filled pore space (WFPS) and soil redox potential (Eh) at three soil depths. We then used the relationship between SP x $\delta^{18}O$-N₂O and SP x $\delta^{15}N$-N₂O in simple two 25   endmember mixing models to evaluate the contribution of nitrification, denitrification and fungal denitrification to total N₂O emissions and to estimate N₂O reduction rates. N₂O emissions were higher in a dry-seeded + alternate wetting and drying (DS-AWD) treatment relative to water-seeded + alternate wetting and drying (WS-AWD) and water-seeded + conventional flooding (WS-FLD) treatments. In the DS-AWD treatment the highest emissions were associated with a high contribution from denitrification and a decrease in N₂O reduction; while in the WS treatments, 30   the highest emissions occurred when contributions from denitrification/nitrifier-denitrification and nitrification/fungal denitrification were more equal. Modeled denitrification rates appeared to be tightly linked to nitrification and $NO_3^-$ availability in all treatments, thus water management affected the rate of denitrification and N₂O reduction by controlling the substrate availability for each process ($NO_3^-$ and N₂O), likely through changes in mineralization and

nitrification rates. Our model estimates of mean $N_2O$ reduction rates match well those observed in $^{15}N$ fertilizer labeling studies in rice systems and show promise for the use of dual isotope ratio mixing models to estimate $N_2$ losses.

## 1 Introduction

Atmospheric nitrous oxide ($N_2O$) concentrations continue to rise, and with a global warming potential 298 times that of $CO_2$, $N_2O$ is a significant contributor to global warming (IPCC, 2007;Ravishankara et al., 2009). Agriculture is estimated to be responsible for roughly 60% of anthropogenic $N_2O$ emissions (Smith et al., 2008). Considering this, the quantification of field scale $N_2O$ emissions has been the focus of many studies in the last decades and much progress has been made on identifying agricultural management practices, soil and climate variables that influence emissions (Mosier et al., 1998;Verhoeven et al., 2017;Venterea et al., 2012). However, it remains difficult to quantitatively determine the microbial source processes of emitted $N_2O$ in the field, and knowledge gaps remain in our understanding of how $N_2O$ production and reduction processes change with both time and depth. More specific knowledge of process dynamics is therefore needed to inform and improve biogeochemical models.

Studying N cycling in rice systems offers a unique opportunity to study processes of $N_2O$ production and reduction. Firstly, there is a strong need to develop alternative water management practices with a shortened paddy flooding period, in order to save water and mitigate methane ($CH_4$) emissions. However, such systems can cause an increase in $N_2O$ emission that may partially offset the decrease in $CH_4$ emission (Devkota et al., 2013;Miniotti et al., 2016;Xu et al., 2015). Hence, water management practices should be improved based on a better understanding of the spatiotemporal origin of $N_2O$ emissions and inorganic N precursors, nitrate and ammonium. Secondly, the complex hydrology, and variable soil moisture conditions between soil layers and within the time course of a growing season, may induce a patchwork of conditions favorable for nitrification versus denitrification versus $N_2O$ reduction. For example, it is not clear if low $N_2O$ emissions under more moist conditions are the result of lower $N_2O$ production due to substrate limitation (i.e. low nitrification rates and hence low $NO_3^-$) or rather increased $N_2O$ reduction. To date, few studies have looked at $N_2O$ processes at depth and it is not known how moisture and nutrient stratification affect the balance between $N_2O$ production and consumption processes and ultimately surface emissions. Analysis of soil $N_2O$ concentrations along a profile should help answer this. Thirdly, rice cropping systems typically suffer from a lower nitrogen use efficiency (NUE) than other major cereal crops, often attributed to high gaseous $NH_3$ and $N_2$ losses (Cassman et al., 1998;Dedatta et al., 1991;Aulakh et al., 2001;Dong et al., 2012). In improving the NUE, a better estimate of $N_2O$ reduction to $N_2$ is needed to design strategies that reduce $N_2$ losses without increasing $N_2O$ emission.

$N_2O$ is predominately produced 1) as a byproduct during nitrification, where $NH_4^+$ is oxidized to $NO_3^-$ via hydroxylamine ($NH_2OH$); this step of nitrification is sometimes referred to as hydroxylamine oxidation (Schreiber et al., 2012;Hu et al., 2015) or 2) as an intermediate in the denitrification pathway during which $NO_3^-$ is reduced to $N_2$ (Firestone et al., 1989) or 3) during nitrifier-denitrification by specific ammonia oxidizing bacteria that oxidize $NH_4^+$ to $NH_2OH$ and then to $NO_2^-$, with a small fraction of $NO_2^-$ then being reduced to NO and $N_2O$ (Kool et al., 2011;Kool et al., 2010;Wrage et al., 2001). $N_2O$ may also be produced from additional biotic and abiotic processes, such as fungal

denitrification, coupled nitrification-denitrification, dissimilatory nitrate reduction to ammonium, chemodenitrification or hydroxylamine decomposition (Butterbach-Bahl et al., 2013;Heil et al., 2015;Zhu-Barker et al., 2015). Due to the prevalence of anaerobic conditions and the use of $NH_4^+$ based fertilizers fungal denitrification and coupled nitrification-denitrification, respectively, are likely to increase in flooded rice systems. $N_2O$ is consumed during the final step of denitrification, where $N_2O$ is reduced to $N_2$ by the $N_2O$ reductase pathway. This can occur sequentially within denitrifying organisms, or $N_2O$ produced elsewhere from other processes or incomplete denitrification can be later reduced by denitrifiers. The final and dominant product of denitrification is $N_2$. While $N_2$ emissions are not of concern for global warming, the quantification of gross denitrification rates is of environmental concern because the loss of N via this process may represent a loss of N from the system and indicate reduced fertilizer N efficiency. Gross denitrification rates are difficult to measure in-situ without the use of isotope tracers due to the high atmospheric background of $N_2$, thus denitrification and $N_2$ emissions remain relatively unconstrained aspects of N budgets.

The measurement of $N_2O$ isotope ratios at natural abundance is a tool to differentiate between in-situ $N_2O$ source processes and $N_2O$ reduction (Toyoda et al., 2011;Ostrom and Ostrom, 2011;Wolf et al., 2015;Baggs, 2008), i.e. $N_2O$ source-partitioning. The evolution of analytical techniques now allows us to measure not only the bulk $\delta^{15}N\text{-}N_2O$, but also the intermolecular distribution of the $\delta^{15}N$ within $N_2O$, called site-preference (SP) and the $\delta^{15}N$ of $N_2O$ precursors, nitrate ($NO_3^-$) and ammonium ($NH_4^+$). The $\delta^{18}O$ of $N_2O$ and its precursors may also be used to constrain processes (Lewicka-Szczebak et al., 2016;Kool et al., 2009;Lewicka-Szczebak et al., 2017). Analytical methods of interpretation remain, however, only semi-quantitative due to uncertainty and overlap in isotope effects ($\varepsilon$, $\eta$ or $\Delta$) for individual processes or cumulative processes and/or multiple N and O sources for which determination of $\delta^{15}N$ and $\delta^{18}O$ remains expensive and time consuming. Theoretically, the O in $N_2O$ derives from $O_2$ during nitrification and from $NO_3^-$ during denitrification or a combination during nitrifier-denitrification (Kool et al., 2007;Snider et al., 2012, 2013;Lewicka-Szczebak et al., 2016;Kool et al., 2010). However, in the case of nitrifier-denitrification and denitrification, intermediates in the reduction pathway ($NO_2^-$ and NO) can extensively exchange O atoms with $H_2O$ (Kool et al., 2007). Such exchange lowers the measured $\delta^{18}O\text{-}N_2O$ values because the influence of relatively depleted $\delta^{18}O$ from $H_2O$, potentially leading to an underestimation of denitrification and $N_2O$ reduction (Snider et al., 2013;Lewicka-Szczebak et al., 2016). Indeed, it has been shown that the $\varepsilon^{18}O$ for denitrification should be calculated relative to $H_2O$ not $NO_3^-$, as almost 100% O exchange occurs (Lewicka-Szczebak et al., 2014;Lewicka-Szczebak et al., 2016). The use of $\delta^{15}N$ values is theoretically more straightforward and there is also a much richer body of literature on $\varepsilon^{15}N$ for various processes, which was recently compiled and reviewed by (Denk et al., 2017). The authors report a mean isotope effect for $^{15}N$ during $NH_4^+$ oxidation to $N_2O$ of $-56.6 \pm 7.3$‰ and of $-42.9 \pm 6.3$‰ for $NO_3^-$ reduction to $N_2O$. Additionally, accurate measurement of the $\delta^{15}N$ of $NH_4^+$ and $NO_3^-$ at sufficient temporal resolution remains time consuming. In comparison, the SP is thought to be independent of the initial substrate $\delta^{15}N$ values and shows distinct values for two clusters of $N_2O$ production, namely $32.8 \pm 4.0$‰ for nitrification/fungal denitrification/abiotic hydroxylamine oxidation and $-1.6 \pm 3.8$‰ for denitrification/nitrifier-denitrification (Decock

and Six, 2013a;Denk et al., 2017).  Abiotic $N_2O$ production from NO has also been reported with an SP of 16‰ (Stanton et al., 2018).

The reduction of $N_2O$ to $N_2$ enriches the pool of remaining $N_2O$ that is measured in $\delta^{15}N$ and $\delta^{18}O$ and thus changes the $\delta^{15}N$-$N_2O$, $\delta^{18}O$-$N_2O$ and SP (Decock and Six, 2013a;Zou et al., 2014).  If the $\delta$ value of $N_2O_{initial}$ (prior to reduction) can be reasonably estimated from graphical and mixing model approaches, then the subsequent enrichment of $N_2O$ can be used to estimate $N_2O$ reduction rates and thereby total denitrification rates. This is important because $N_2O$ reduction is a crucial but exceptionally poorly constrained process within the N cycle (Lewicka-Szczebak et al., 2017).  Fractionation during $N_2O$ reduction may follow dynamics of open or closed systems (Fry, 2007;Mariotti et al., 1981).

Our goal was to collect a high resolution in situ $N_2O$ isotope ratio data set that could be used to a) determine the stratification of $N_2O$ production and reduction processes in relation to water management, b) semi-quantitatively assess $N_2O$ and $N_2$ loss rates among rice water management treatments and c) push forward current natural abundance $N_2O$ isotope source-partitioning methods and interpretation at the field scale.  We compared three rice water management practices: direct dry seeding followed by alternate wetting and drying (DS-AWD), wet seeding followed by alternate wetting and drying (WS-AWD) and wet seeding followed by conventional flooding (WS-FLD).  Isotope data was determined at three depths, simultaneously with soil environmental and nutrient data and soil $N_2O$ and dissolved $N_2O$ concentrations.  We hypothesized that $N_2O$ emissions would be highest in the AWD treatments due to greater contributions from nitrification and less $N_2O$ reduction, following the order: DS-AWD > WS-AWD > WS-FLD.  We also hypothesized that $N_2$ emissions are controlled by the availability of $NO_3^-$ coming from nitrification and high soil moisture.  We considered that $NO_3^-$ would be higher under WS-AWD but soil moisture would be higher under WS-FLD; therefore we predicted $N_2$ emissions to follow in the order: WS-AWD > WS-FLD > DS-AWD.  Lastly, we hypothesized that longer periods of lowered soil moisture in the DS-AWD and WS-AWD treatments would result in greater production of $N_2O$ at depth and this higher production would increase surface emissions.

## 2 Materials and Methods

### 2.1 Field experiment

A field experiment consisting of three water management regimes was conducted at the Italian Rice Research Center (Ente Nazionale Risi), Pavia, Italy (45°14'48"N, 8°41'52"E).  Experimental work focused only on the early growing season, lasting from the 13th of May, 2016 until June 30th, 2016.  It is in this period that the highest $N_2O$ losses and N cycling dynamics had been previously observed and the largest differences among water management practices occurred.  The experiment was conducted in the 5th year of alternative water management in an existing experimental platform.  During the first three years the paddies were maintained as dry-seeding + flooding, wet-seeding + flooding and intermittent irrigation as described in (Miniotti et al., 2016;Peyron et al., 2016;Said-Pullicino et al., 2016).  In the fourth year, the intermittent irrigation treatment was changed to wet seeding + alternate wet dry (Verhoeven et al.,

2018).  In the current study dry-seeding + flooding treatment was shifted to dry-seeding + alternate wet dry, the other treatments remained as in the 4$^{th}$ year.  Irrigation and water management details are provided below.  The soil at the site has been classified as coarse silty, mixed, mesic Fluvaquentic Epiaquept (USDA-NRCS, 2010).  The mean soil texture in the upper 30 cm of the experimental plots was 26% sand, 62% silt, and 11% clay with a mean bulk density of 1.29 g cm$^{-3}$.  At the end of the 2015 growing season, mean total organic C and total N were 1.07 and 0.11% and pH 5.9 (1:2.5 H$_2$O) and 5.2 (1:2.5 0.01M CaCl$_2$), respectively.  Annual and growing season mean temperatures in 2016 were 10°C and 23°C, respectively (Fig. S1).  Annual and growing season cumulative precipitation was 618 and 258 mm, respectively.  Data for both values were retrieved from a regional weather station operated by the Agenzia Regionale per la Protezione dell'Ambiente-Lombardia, located approximately 200 m from the field site (ARPA).

Water management in the two WS treatments was identical during the first three weeks of the growing season (Table 1).  Following regional practices for water seeding, paddies were flooded for six days at the time of seeding, but then drained for ~ 2 weeks to promote germination.  During this period of 'drainage' paddies were not dry but maintained near saturation by flush irrigation as necessary (May 31$^{st}$ and June 6$^{th}$).  Flush irrigation is a practice in which the water inlet channels are opened for a few hours and then the outlet channels are opened a few hours later resulting in temporary soil saturation or even 1-2 cm ponding for 2-4 hours.  On June 10$^{th}$, approximately three weeks after seeding, treatment differentiation between the WS-FLD and WS-AWD began.  At this time the WS-FLD was flooded, while the WS-AWD was only flush irrigated.  On June 16$^{th}$, the WS-FLD was allowed to drain slowly in order to facilitate fertilizer application on June 21$^{st}$.  Following fertilizer application, the WS-FLD treatment was re-flooded and both AWD treatments were flush irrigated on June 22$^{nd}$.  In the DS-AWD treatment no flooding or irrigation water was applied prior to June 22$^{nd}$.  Soil moisture depended on rainfall, which was 75 mm during the four weeks following seeding.

In all treatments, crop residues were incorporated in the spring, before the cropping season.  All paddies were harrowed and leveled approximately one month prior to seeding in mid-April, 2016.  All treatments were pre-fertilized with phosphorus and potassium on May 13$^{th}$ (14 and 28 kg ha$^{-1}$, respectively).  A total of 160 kg N ha$^{-1}$ as urea was applied to all treatments, with one pre-plant application on May 16$^{th}$ and two in-season applications on June 21$^{st}$ and July 14$^{th}$ (Table 1).  Following best management practices for the three water management practices, a smaller pre-plant urea application was applied in the DS-AWD treatment, followed by a larger application in this treatment at the second and third fertilization.  In the DS-AWD treatment, urea was applied at 40, 70 and 50 kg N ha$^{-1}$, while these rates were 60, 60 and 40 kg N ha$^{-1}$ for the WS treatments at fertilization 1, 2 and 3, respectively.  The WS-FLD and WS-AWD treatments were seeded on May 20$^{th}$.  All treatments were harvested on September 15$^{th}$.

Each treatment consisted of two paddies, 20 x 80 m, with two plots in each paddy, n=4 (Fig. S2).  The experimental design was identical to that of (Verhoeven et al., 2018), with the addition of the DS-AWD treatment and some adjustment to plot placement in order to accommodate data logging devices and field equipment.  Each paddy was approximately 2 m apart and hydrologically separated by a levee of 50 cm above the soil surface, flanked by an

irrigation canal on either side.  Sampling for $N_2O$ surface fluxes, pore water parameters ($NO_3^-$, $NH_4^+$, DOC, dissolved $N_2O$) and pore air $N_2O$ occurred on 15-17 dates, from the 20th of May to the 30th of June, 2016 (Table S1).  Sampling dates were on average three days apart with a greater frequency before and after N application on the 21st of June.  Sub-samples of pore water from 10 to 12 dates were analyzed for $\delta^{15}N$-$NO_3^-$, $\delta^{18}O$-$NO_3^-$ and $\delta^{15}N$-$NH_4^+$.

**2.2 Soil environment: temperature, redox potential, and moisture**

Soil moisture was measured using PR2 capacitance probes (Delta T Devices, UK) at 5, 15, 25, 45 and 85 cm.  Water filled pore space (WFPS) was calculated using bulk density measurements at 5, 12.5 and 25 cm collected at the beginning of the season using a Giddings manual soil auger.  Soil temperature was measured in only one plot per paddy (n=2) at three depths (5, 12.5 and 25 cm).  Measurements were made manually at the time of surface flux gas

measurements.  Soil redox potential (Eh) was measured continuously in each plot using sturdy tip probes outfitted with 5 Pt-electrodes that were permanently connected to a 48-channel Hypnos-III data logger (MVH Consult, The Netherlands) with two Ag/AgCl-reference probes.  Soil Eh was measured every hour at six depths; 5, 12.5, 20, 30, 50 and 80 cm.  We took the average of the 20 and 30 cm readings to derive a 25 cm reading in order to correlate to other measurements.

**2.3 $N_2O$ measurements: surface emissions, pore air, and dissolved gas**

All $N_2O$ concentration measurements were measured by gas chromatography on a Scion 456-GC (Bruker, Germany) equipped with an electron capture detector (ECD).  A standard curve was derived from 10 replicates of at least 5 concentrations to determine the standard deviation for a given concentration.  For example, the error of the GC was determined to be $\pm$ 0.012 at 0.3 ppm and $\pm$ 0.024 ppm at 1.0 ppm.  $N_2O$ surface emissions ($N_2O_{emitted}$) were measured

by the non-steady state closed chamber technique (Hutchinson and Mosier, 1981).  The chamber design and deployment was identical to that of (Verhoeven et al., 2018).  Gas samples were taken at 0, 10, 20 and 30 min in each chamber and injected into pre-evacuated exetainers (Labco, UK).  At time 0 and 30 min an additional ~ 170 ml of sample was taken and injected into gas crimp neck vials sealed with Butyl injection stoppers (IVA Analysentechnik, Germany) to be used for isotope analysis.  When the accumulation of gas over the course of measurement was less

than the standard deviation associated with the highest concentration of the four measurements, the flux was determined to be below detection.  Fluxes above the detection limit were calculated by linear or non-linear regression following the method outlined by Verhoeven and Six (2014).  Soil $N_2O$ ($N_2O_{soil}$) was sampled using passive diffusion probes installed at 5, 12.5 and 25 cm.  The probe design and sampling strategy has been previously described in (Verhoeven et al., 2018).  In brief, the samples were collected in He flushed and pre-evacuated 100 ml glass crimp

neck vials (actual volume 110 ml, IVA Analysentechnik, Germany) and after sampling topped with high purity He gas to prevent leakage into under-pressurized vials.  The final $N_2O$ concentration was determined by gas chromatography, as described above, on a subsample, while the remainder of the sample was retained for isotope analysis.  The final $N_2O$ concentration was calculated by accounting for sample dilution based on the pressure after evacuation, after sampling and after topping with He gas.  Samples for dissolved $N_2O$ ($N_2O_{dissolved}$) were collected by

injecting a 5 ml subsample of pore water, collected as described in section 2.4,  into $N_2$ flushed and filled exetainers

that also contained 50μl of 50% ZnCl to stop microbial activity. Samples were stored at 4°C until the end of the experimental campaign and transported back to the lab for analysis, therefore there was adequate time for the equilibration between the headspace and aqueous phases. The molar concentration of $N_2O$ was calculated by applying the solubility constant of $N_2O$ at the time of analysis (i.e. lab temperature) to Henry's law (Lide, 2004;Weiss and Price, 1980;Wilhelm et al., 1977), taking into account the vial volume and headspace.

**2.4 Pore water measurements**

Two MacroRhizon pore water samplers (Rhizosphere Research Products, The Netherlands) were installed at each depth (5, 12.5 and 25 cm) in every plot. Pore water was then collected in two polypropylene 60 ml syringes at each depth and later pooled together at sample processing. The syringes were attached to the MacroRhizon sample tubes with two-way leur lock valves and propped open using a wedge, which served to create a low vacuum; the syringes were left to collect water for 2-4 h. Samples were stored at 4°C and processed within 36 h. During pore water processing ~ 15 ml of solution was allocated for analysis of $NO_3^-$ and $NH_4^+$ and $\delta^{15}N$, $\delta^{18}O$-$NO_3^-$, ~ 15 ml for $\delta^{15}N$-$NH_4^+$, 5 ml for dissolved $N_2O$, 3-5 ml for dissolved $Fe^{2+}$ and $Mn^{2+}$ and 5 ml for DOC/TDN analysis. All samples, aside from those for dissolved $N_2O$, were frozen at -5°C until analysis. $NO_3^-$ and $NH_4^+$ were determined by spectrophotometry following the procedure of (Doane and Horwáth, 2003). DOC and TDN were determined by first acidifying the water sample to pH <2 by addition of concentrated HCl and then analysis on a multi N/C 2100S:TOC/TN Analyzer (Analytik Jena, Germany).

**2.5 Determination of $\delta^{15}N$, $\delta^{18}O$ and isotope ratios in $N_2O_{emitted}$ and $N_2O_{soil}$**

Surface and pore air gas samples were taken in 100 ml glass crimp neck vials (actual volume 110 ml, IVA Analysentechnik, Germany) as described in section 2.3. Pore air gas samples were preconditioned with 1 ml of 1M NaOH solution prior to analysis due to very high $CO_2$ concentrations in many samples (> 5000 ppm). The intramolecular site-specific isotopic composition of the $N_2O$ molecule was measured using a gas preparation unit (Trace Gas, Elementar, UK) coupled to an isotope ratio mass spectrometer (IRMS; IsoPrime100, Elementar, UK). The gas preparation unit was modified with an additional chemical trap (½'' diameter stainless steel), located immediately downstream from the autosampler. This pre-trap was filled with NaOH, $Mg(ClO_4)_2$, and activated carbon in the direction of flow and is designed to further scrub $CO_2$, $H_2O$, CO and VOCs which otherwise would cause mass interference during measurement. Before final injection into the IRMS the purified gas sample is directed through a Nafion drier and subsequently separated in a gas chromatograph column (5Å molecular sieve).

The IRMS consists of five Faraday cups with *m/z* of 30, 31, 44, 45, 46, measuring $\delta^{15}N$ and $\delta^{18}O$ of $N_2O$ and $\delta^{15}N$ from the $NO^+$ fragments dissociated from $N_2O$ during ionization in the source. The $^{15}N/^{14}N$ ratio of the NO molecule is used to calculate the α (central) position of the initial $N_2O$, thus allowing measurement of the site-specific isotopic composition of $N_2O$ (SP). Site preference is defined as $\delta^{15}N^{SP} = \delta^{15}N^\alpha - \delta^{15}N^\beta$ with α denoting the $^{15}N/^{14}N$ ratio of the central N atom and β the $^{15}N/^{14}N$ ratio of the terminal N atom of the linear NNO molecule. $\delta^{15}N^\beta$ is indirectly obtained from rearrangement of:

$\delta^{15}N^{bulk} = (\delta^{15}N^{\alpha} + \delta^{15}N^{\beta})/2$

which represents the average $^{15}N$ content of the $N_2O$ molecule.

For IRMS calibration three sets of two working standards ($\sim$ 3 ppm $N_2O$ mixed in synthetic air) with different isotopic composition ($\delta^{15}N^{\alpha}$ = 0.954 ± 0.123 ‰ and 34.446 ± 0.179 ‰; $\delta^{15}N^{\beta}$ = 2.574 ± 0.086 ‰ and 35.98 ± 0.221 ‰; $\delta^{18}O$ = 39.741 ± 0.051 ‰ and 38.527 ± 0.107 ‰) were used. These standards have been analyzed at EMPA using TREX-QCLAS versus standards with assigned δ-values by Tokyo Institute of Technology (Mohn *et al.*, 2014). These working standards were run in triplicate, evenly spaced throughout a run. Sample peak ratios are initially reported against a $N_2O$ reference gas peak (100% $N_2O$, Carbagas, Switzerland) and are subsequently corrected for drift and span using the working standards. Further correction procedures, such as $^{17}O$ mass overlap and scrambling, as reported elsewhere, were not applied as the data was inherently corrected by regression between true and measured values of the triplicate working standards. Long-term measurement quality was ensured using a control standard at low $N_2O$ concentration ($\sim$ 0.4 ppm) treated as a sample. Instrument linearity and stability was frequently checked by injection of 10 reference gas pulses of either varying or identical height respectively, with accepted levels of <0.03‰/nA. Since instrument linearity could only be achieved for either $N_2O$ or NO, the instrument had been tuned for the former and $\delta^{15}N^{\alpha}$ subsequently corrected using sample peak height assuming a non-linearity of 0.1 ‰ $nA^{-1}$. Such linearity complications have been previously reported using Elementar (Ostrom et al., 2007) and ThermoFinnigan IRMS (Röckmann et al., 2003). Tropospheric air was regularly measured (n=42) and used as a confirmation of correction procedures, yielding consistent and reliable results: $\delta^{15}N^{SP}$ = 18.77 ± 1.08 ‰; $\delta^{15}N^{bulk}$ = 5.96 ± 0.35 ‰; $\delta^{15}N^{\alpha}$ = 15.34 ± 0.70 ‰, $\delta^{15}N^{\beta}$ = -3.43 ± 0.60 ‰; $\delta^{18}O$ = 43.67 ± 0.41 ‰. All $^{15}N/^{14}N$ sample ratios are reported relatively to the international isotope ratio scale AIR-N2 while $^{18}O/^{16}O$ are reported versus Vienna Standard Mean Ocean Water (V-SMOW). Relative differences are given using the delta notation (δ) in units of ‰:

$$\delta^{Z}X \, [‰] = \frac{R_{sample}}{R_{reference}} - 1$$

(1)

where *R* is referring to the molar ratio of $^{15}N/^{14}N$ or $^{18}O/^{16}O$ and $^{Z}X$ to the abundance of the heavy stable isotope *Z* of element *X*.

**2.6 Determination of $\delta^{15}N$-$NO_3^-$, $\delta^{18}O$-$NO_3^-$ and $\delta^{15}N$-$NH_4^+$**

Pore water $NO_3^-$ samples were analyzed for $\delta^{15}N$ and $\delta^{18}O$ at the University of California, Davis, Stable Isotope Facility (http://stableisotopefacility.ucdavis.edu/), using the denitrifier method developed by (Sigman et al., 2001;Casciotti et al., 2002;McIlvin and Casciotti, 2011). $\delta^{15}N$-$NH_4^+$ in pore water was determined by micro-diffusion onto acidified disks followed by persulfate digestion (Lachouani et al., 2010;Stephan and Kavanagh, 2009) and lastly by the denitrifier method. For $\delta^{15}N$-$NH_4^+$, all steps and analyses were done in-house, including the denitrifier method. Our limit of quantification for $\delta^{15}N$-$NH_4^+$ was 0.75 mg $L^{-1}$ or ~42 µM $NH_4^+$, below this the diffusion gradient was too low for reliable diffusion. Briefly, samples were run in sets of 40 with 24 samples and a combination of 16 standards

and blanks. Each run contained at least two $\delta^{15}$N-NH$_4^+$ isotope standards (IAEA N2 = 20.3‰; IAEA N1 = 0.4‰; USGS 25 = -30.4‰) at two or three concentrations in duplicate or triplicate in addition to two blanks and two working standards. NH$_4^+$ isotope standards were diffused, digested and run through the denitrifier method in parallel with samples and therefore an overall correction and concentration offset was derived and applied for each batch. The denitrifier method was executed using the updated protocol described by (McIlvin and Casciotti, 2011) using *Pseudomonas aureofaciens* (ATCC 13985). An IAEA KNO$_3^-$ standard ($\delta^{15}$N = 4.7‰) was included at the denitrifier method step to ensure accurate conversion of NO$_3^-$ to N$_2$O. A propagated error across all steps of $\delta^{15}$N-NH$_4^+$ quantification was calculated from the working standards included in each batch (n=18). We excluded three values that were well outside the expected range; our overall precision was 1.9‰. The largest sources of error were incomplete diffusion or persulfate digestion. For $\delta^{15}$N-NO$_3^-$ and $\delta^{18}$O-NO$_3^-$ analyzed at SIF, UC-Davis, the limit of quantification was 0.125 mg L$^{-1}$ NO$_3^-$ or 2.0 µM NO$_3^-$ or, with a precision of 0.4‰ and 0.5‰ for $\delta^{15}$N and $\delta^{18}$O, respectively.

Using N$_2$O$_{poreair}$ and NO$_3^-$ and NH$_4^+$ in pore water we calculated the $\Delta^{15}$N of NO$_3^-$ reduction to N$_2$O and of NH$_4^+$ oxidation to N$_2$O using equation 2 and 3, respectively.

$$\Delta^{15}N_{N_2O-NO_3} \quad = \delta^{15}N_{N_2O} - \delta^{15}N_{NO_3} \tag{2}$$

$$\Delta^{15}N_{N_2O-NH_4} \quad = \delta^{15}N_{N_2O} - \delta^{15}N_{NH_4} \tag{3}$$

The calculation of $\Delta^{15}$N$_x$ can be compared to the net isotope effects for nitrification and denitrification derived N$_2$O, as found in the literature. In reality the processes in equations 1 and 2 entail a series of sequential reactions each of which has a unique isotope effect ($\varepsilon_{k,1}$, $\varepsilon_{k,2}$, $\varepsilon_{k,3}$,…). It is not possible to measure the isotope values of many of the intermediaries in these reactions series, particularly in in situ field settings, therefore we report the $\Delta^{15}$N$_x$. For the calculation of $\Delta^{15}$N$_x$ we assume open system dynamics because all measurements were in situ where substrates, products and intermediaries could be replenished by other processes.

**2.7 Determination of N$_2$O source contribution and N$_2$O reduction**

**2.7.1 Two endmember mixing models using SP and $\delta^{18}$O signatures: closed and open systems**

We used two mixing models where N$_2$O reduction was modeled under 'open' and 'closed' system dynamics following the theory outlined originally by (Fry, 2007) and (Mariotti et al., 1981), respectively. The two modeling methods are henceforth referred to as 'open' and 'closed'. In reality, the heterogeneity in microbial microhabitat within the soil most likely results in a mixture of closed versus open system dynamics. Therefore, final data interpretations were made for the average findings across open versus closed systems dynamics. A schematic of our closed system approach is given in Fig. 1. For both open and closed methods, two possible scenarios were considered as described by (Lewicka-Szczebak et al., 2017); scenario 1 (sc1), where N$_2$O is produced and reduced by denitrifiers before mixing with N$_2$O derived from nitrification or scenario two (sc2) where N$_2$O is produced from both processes, mixed, and then reduced. In both models, N$_2$O is originally produced from two possible endmembers; denitrification/nitrifier-denitrification (denoted by subscript *den*) and nitrification/fungal denitrification (denoted by subscript *nit*). Our intention was to keep the derivation of endmember values consistent between this study and Lewicka-Szczebak et al.

(2017). Our SP endmember values ($SP_{den}$ and $SP_{nit}$) and $N_2O$ reduction fractionation factors ($\varepsilon^{18}O_{red}$ and $\varepsilon SP_{red}$) were taken directly from Lewicka-Szczebak et al. (2017) (Table 2). For $\delta^{18}O$-$N_2O_{(x)}$ endmember values we could not directly use the values reported in Lewicka-Szczebak et al. (2017) because these were reported relative to $\delta^{18}O$-$H_2O$ (as $\delta^{18}O$-$N_2O(N_2O/H_2O)$) and we did not measure the isotope signature of water in our study. Therefore, $\delta^{18}O$-$N_2O_{nit}$ was re-calculated using the original mean values ($\delta^{18}O$-$N_2O$ as opposed to $\delta^{18}O$-$(N_2O/H_2O)$ of the six studies referenced by (Lewicka-Szczebak et al., 2017), this yielded a mean of 36.5‰ (Heil et al., 2014;Sutka et al., 2006;Sutka et al., 2008;Frame and Casciotti, 2010;Rohe et al., 2014;Maeda et al., 2015). For $\delta^{18}O$-$N_2O_{den}$ we adjusted the value used in Lewicka-Szczebak et al. (2017) by an estimate of $\delta^{18}O$-$H_2O$ of water for our site rather than re-calculate from the four studies originally referenced by Lewicka-Szczebak et al. (2017) (Lewicka-Szczebak et al., 2014;Lewicka-Szczebak et al., 2016;Frame and Casciotti, 2010;Sutka et al., 2006). We used a $\delta^{18}O$-$H_2O$ value of -8.3‰, as reported by Rapti-Caputo and Martinelli (2009) for an uncontained aquifer of the Po River delta. We chose to do this because some of the mean values used in calculations by Lewicka-Szczebak et al. (2017) were themselves calculated from data originally reported.

Closed system fractionation for $N_2O$ reduction was modeled following the method described in (Lewicka-Szczebak et al., 2017) (Fig.1). A detailed protocol for these calculations can also be found on ResearchGate (DOI:10.13140/RG.2.2.17478.52804). In brief, sample SP and $\delta^{18}O$-$N_2O$ values are used to derive sample specific intercepts that pass through the sample and reduction line (sc1) or the sample and the mixing line (sc2). A fixed slope for the reduction line can be calculated from $\varepsilon SP_{red} / \varepsilon^{18}O_{red}$ (i.e. in our case, -5/-15). In sc1, the intercept of the mixing and reduction line represents $N_2O$ that has been produced from denitrification/nitrifier-denitrification and partially reduced but not yet mixed with $N_2O$ produced from nitrification/fungal denitrification. In sc2, the intercept of these lines represents $N_2O$ that has been produced by the two endmember pools, mixed, but not yet reduced. The Y axis (i.e. SP) value of these respective intercepts can be used in a generalized Rayleigh equation (Eq. 4) to calculate the extent of $N_2O$ reduction, represented by the fraction of residual $N_2O$ not reduced.

$$SP_{resid.N2O} \approx SP_{N2O-unreduced} + \varepsilon SP_{red} \cdot \ln(rN_2O_{net}) \tag{4}$$

In sc1 the $rN_2O$ is determined with respect to $N_2O$ from denitrification/nitrifier-denitrification only, therefore to calculate the residual fraction of total production (i.e. $N_2 + N_2O$) we calculate gross $rN_2O$:

$$gross\ rN_2O_{sc1} = \frac{1}{fracDenit_{net}/rN_2O_{net} + 1 - fracDenit_{net}} \quad (sc1, in\ sc2\ rN_2O_{net} = rN_2O_{gross}) \tag{5}$$

To calculate the fraction of denitrification of the total initially produced $N_2O$ (emitted as $N_2O$ and $N_2$) we calculate the gross denitrification fraction:

$$gross\ frac_{DEN\ sc1-closed} = \frac{fracDenit_{net}/rN_2O_{net}}{fracDenit_{net}/rN_2O_{net} + 1 - fracDenit_{net}} \quad (sc1) \tag{6}$$

To calculate the fraction of denitrification/nitrifier-denitrification to the net $N_2O$ produced, we use Eq. 7. For simplicity and comparison with open system calculations, we call this *DenContribution*.

$$net\ frac_{DENsc1-closed} = \frac{SP_{sample} - SP_{nit}}{SP_{resid.N2O} - SP_{nit}} \quad (sc1) \ = \text{DenContribution}_{closed-sc1} \tag{7}$$

In this case, SP$_{resid.N2O}$ is the signature of residual bacterial $N_2O$ after partial reduction but before mixing. This was determined from the graphical method (Lewicka-Szczebak et al., 2017). In sc2 both net and gross fractions of denitrification are equal and can be expressed as:

$$DenContribution_{closed-sc2} = \frac{SP_{N2O-unreduced}-SP_{nit}}{SP_{den}-SP_{nit}} \quad (sc2) \tag{8}$$

Here, SP$_{N2O-undreduced}$ is the signature of $N_2O$ mixed from nitrification/fungal denitrification and denitrification/nitrifier-denitrification, but before reduction. This was determined from the graphical method (Lewicka-Szczebak et al., 2017).

To predict $r$N$_2$O in open systems we set up a series of mass balance equations using our measured $N_2O$ flux or $N_2O_{poreair}$ concentrations and measured $\delta^{18}O$ and SP values. We used the same endmember values listed in Table 2 for all equations. As above, we can model the interaction between mixing and reduction assuming sc1 (Eqs 9-11) or sc2 (Eqs 9,12,13). In Eqs 9-13, we use $k_{nit}$, $k_{den}$ and $k_{red}$ to represent the gross process rates or concentrations of $N_2O$ attributable to nitrification, denitrification and $N_2O$ reduction, respectively.

$$N_2O_{flux}(or\ N_2O_{poreair}) = k_{nit} + k_{den} - k_{red} \quad note: k_{den} = \text{total denitrification } (N_2O + N_2) \tag{9}$$

$$SP - N_2O_{measured} = \frac{SP_{nit}k_{nit}+\left(SP_{den}-\varepsilon SP_{red}\left(\frac{k_{red}}{k_{den}}\right)\right)(k_{den}-k_{red})}{k_{nit}+k_{den}-k_{red}} \quad (sc1) \tag{10}$$

$$\delta^{18}O - N_2O_{measured} = \frac{(\delta^{18}ON_2O_{nit})k_{nit}+\left(\delta^{18}ON_2O_{den}-\varepsilon^{18}O_{red}\left(\frac{k_{red}}{k_{den}}\right)\right)(k_{den}-k_{red})}{k_{nit}+k_{den}-k_{red}} \quad (sc1) \tag{11}$$

$$SP - N_2O_{measured} = \frac{(SP_{nit}k_{nit}+SP_{den}k_{den})}{k_{nit}+k_{den}} - \varepsilon SP_{red}\left(1-\frac{N_2O_{flux}}{k_{nit}+k_{den}}\right) \quad (sc2) \tag{12}$$

$$\delta^{18}O - N_2O_{measured} = \frac{(\delta^{18}ON_2O_{nit})k_{nit}+(\delta^{18}ON_2O_{den})k_{den}}{k_{nit}+k_{den}} - \varepsilon^{18}O_{red}\left(1-\frac{N_2O_{flux}}{k_{nit}+k_{den}}\right) \quad (sc2)$$
(13)

These two sets of equations (Eq. 9,10,11) or (Eq. 9,12,13), representing each scenario, were applied to measured surface fluxes to produce process rates in g $N_2$O-N ha$^{-1}$ d$^{-1}$ or were applied to $N_2O_{poreair}$ concentrations to produce concentrations of $N_2O$ in µg $N_2$O-N L$^{-1}$. By rearranging these process rates or concentrations we can calculate gross $r$N$_2$O, frac$_{DEN}$ and the contribution of denitrification to $N_2O$ using Eqs. 14-16.

$$gross\ frac_{DEN\ sc1,sc2-open} = \frac{k_{den}}{k_{nit}+k_{den}} \tag{14}$$

$$gross\ rN_2O_{sc1,sc2-open} = \frac{k_{nit}+k_{den}-k_{red}}{k_{nit}+k_{den}} \tag{15}$$

$$DenContribution_{sc1,sc1-open} = \frac{(k_{den}-k_{red})}{[N_2O]} \quad , [N_2O] = N_2O_{flux}\ or\ N_2O_{poreair} \tag{16}$$

Plausible solutions for $k_{red}$, $k_{den}$, and $k_{red}$ were estimated based on minimizing the sum of squares between the modeled and measured $N_2O$ flux (or concentration), $\delta^{18}O$ and SP values using a Generalized Reduced Gradient (GRG) nonlinear algorithm in the *Solver* function of excel. Example calculations for the open system modeling are given in an Excel supplementary material file. Solutions with a minimum sum of squares over 500 were considered implausible (8.3% of solutions) (Table S2). Both models produced some non-plausible solutions, i.e. fractional contributions over 1 or under 0. Only solutions with a gross $r$N$_2$O, gross frac$_{DEN}$ and DenContribution between 0 to 1 and an open system minimum sum of squares < 500 were retained. In sc1, roughly 75% of solutions met these criteria. For sc2, less than

10% of solutions in the open system met this criteria, therefore we do not proceed to analyze and discuss solutions from sc2 (Table S2 and Fig. S3).

**2.8 Statistical analyses**

Response variables were analyzed using a linear mixed effects ANCOVA model with treatment, date, and depth (if applicable) as fixed effects and plot as a random effect. The longitudinal position in the field (Y position) measured in meters from the central driveway (Fig. S2), was used as a covariate to account for potential heterogeneity in the longitudinal direction. In the case of non-normally distributed data, data was transformed to obtain a normal distribution of residuals. Due to the non-normal distribution of many variables, Spearman correlations were used to analyze the relationship between $N_2O_{emitted}$ fluxes, isotope ratios, soil environmental and substrate variables. Post-hoc analysis of treatment and depth within a given day was performed using the *lsmeans* function with a Tukey adjustment for multiple comparisons. For the analysis of modeling results we eliminated the 25 cm depth due to poor data availability. All data analysis was done in R version 3.3.2.

**3 Results**

**3.1 Yield**

At the end of the growing season yield was measured in the larger plots in which are sampling plots were situated. The DS-AWD treatment had a significantly lower yield, 6.6 t/ha, relative to 8.9 and 8.2 t/ha in the WS-FLD and WS-AWD, respectively (Table 1).

**3.2 $N_2O$ fluxes, dissolved and pore air $N_2O$ concentrations**

**3.2.1 Temporal patterns in $N_2O$ fluxes and concentrations**

After the first basal fertilization (May 16th) and prior to the second topdressing fertilization (June 21st), emissions were significantly higher in the DS-AWD treatment than in WS-AWD and WS-FLD on eight and six of the 11 sampling days, respectively (Fig. 2). During this time four peaks in emissions were observed in the DS-AWD treatment, on May 20th, June 1st-3rd, June 7-9th, and June 20th, averaging $39.5 \pm 5.1$ g $N_2O$-N ha$^{-1}$ d$^{-1}$. A peak in emissions following the second fertilization (June 21st) was observed in all treatments; in the DS-AWD treatment emissions peaked at $108.2 \pm 4.2$ g $N_2O$-N ha$^{-1}$ d$^{-1}$ on June 23rd, while in the WS-AWD and WS-FLD treatments, emissions peaked one day earlier reaching $49.4 \pm 17.9$ and $77.67 \pm 10.6$ g $N_2O$-N ha$^{-1}$ d$^{-1}$, respectively. In the WS-AWD treatment, emissions remained slightly elevated following this fertilization until the end of the monitoring campaign, while in the DS-AWD and WS-FLD, emissions declined after June 22 or 23rd, respectively.

If we exclude $N_2O_{dissolved}$ measurements from the DS-AWD treatment following the second fertilization (i.e. after the 22nd of June, when concentrations reached as high as $594.4 \pm 112.6$ µg $N_2O$-N L$^{-1}$ at 5 cm), concentrations throughout the profile of all treatments remained under 20 µg $N_2O$-N L$^{-1}$. Due to the large differences between dates and treatments we present the concentrations on a log$_{10}$ scale (Fig. 2) and non-transformed scale (Fig. S4). Peak

concentrations in the WS treatments occurred at 5 cm on the first day of measurement, reaching $17.7 \pm 5.1$ and $18.5 \pm 2.8$ µg $N_2O$-N $L^{-1}$ in the WS-AWD and WS-FLD, respectively.  In comparison, in the DS-AWD treatment peak concentrations prior to the second fertilization were observed at 25 cm on June 3rd, reaching $18.5 \pm 8.3$ µg $N_2O$-N $L^{-1}$.

As with dissolved $N_2O$, pore air $N_2O$ concentrations were highly variable between treatments and between sampling days and are again presented on a $log_{10}$ scale (Fig. 2) and non-transformed scale (Fig. S4).  In both WS treatments, the highest concentrations were observed on the first day of measurement, May 20th, reaching $2903.3 \pm 1103.6$ and $1321 \pm 998.0$ µg $N_2O$-N $L^{-1}$ at 5 cm in the WS-FLD and WS-AWD, respectively.  Elevated concentrations of $N_2O_{poreair}$

were also observed in the DS-AWD on the first day of measurement but were 70.1 µg $N_2O$-N $L^{-1}$ at 5 cm (roughly 40x lower than in WS-FLD on this date).   Maximum concentrations in the DS-AWD treatment were observed two days after the second fertilizer application, reaching 1902.2 µg $N_2O$-N $L^{-1}$; in contrast no change was observed in the WS treatments following this fertilizer application.  In all treatments the majority of $N_2O_{poreair}$ concentrations were orders of magnitude lower than these peaks.  There was a tendency of lower $N_2O_{poreair}$ concentrations in the DS-AWD

treatment relative to the WS treatments; this pattern was most evident at 5 cm (Fig. 2).  However, treatment differences in $N_2O_{poreair}$ were not significant ($p=0.08$, Table S3) and there was a significant date x treatment interaction.

### 3.2.2 Relation of N₂O fluxes and concentrations with soil environment, substrates and N₂O isotope ratios

We evaluated the correlation of $N_2O_{emitted}$ with Eh, WFPS, $NO_3^-$, $NH_4^+$, dissolved and pore air $N_2O$ concentrations and $N_2O$ isotope ratios at 5 cm (Table 3).  Among these variables, $N_2O$ emissions in the WS treatments were negatively

correlated with pore water $NH_4^+$ and DOC in the WS-AWD treatment.  In the DS-AWD treatment, emissions positively correlated with $N_2O_{poreair}$, WFPS, and $NO_3^-$ and negatively with $N_2O$ isotope ratios.  Examining the isotope ratios of $N_2O_{emitted}$, we observed that $N_2O_{emitted}$ was negatively correlated with $\delta^{18}O$-$N_2O_{emitted}$ in all treatments, negatively with $\delta^{15}N$-$N_2O_{emitted}$ in the DS-AWD treatment and negatively with SP-$N_2O_{emitted}$ in the WS-FLD and DS-AWD.  Interestingly, a positive correlation between $N_2O_{emitted}$ and SP-$N_2O_{emitted}$ was observed in the WS-AWD

treatment.  Relative to the DS-AWD, the WS treatments had fewer significant correlations between $N_2O$ isotope ratios, soil environment or pore air $N_2O$ isotope ratios.  DOC was positively correlated with $\delta^{15}N$-$N_2O_{emitted}$ in the WS-AWD and with $\delta^{18}O$-$N_2O_{emitted}$ in the WS-FLD.  SP-$N_2O_{emitted}$ was positively correlated to Eh and negatively to WFPS in the WS-AWD treatment.  In comparison, in the DS-AWD treatment, $N_2O_{emitted}$ isotope ratios were positively correlated to that of $N_2O_{poreair}$ for all three isotopes.  Furthermore, $N_2O$ isotope ratios in the DS-AWD treatment were negatively

correlated with $N_2O_{poreair}$ concentrations, WFPS, $NO_3^-$ ($\delta^{15}N$-$N_2O_{emitted}$ only) and $N_2O_{dissolved}$ ($\delta^{18}O$-$N_2O_{emitted}$ and SP-$N_2O_{emitted}$ only).  It should be noted that $N_2O_{dissolved}$ in the DS-AWD treatment was not measurable at the 5 cm depth on 10 of the 16 sampling dates due to low soil moisture and low pore water volumes.

### 3.3 Spatiotemporal patterns of N₂O isotope ratios

#### 3.3.1 $\delta^{15}$N-N₂O

A consistent temporal pattern of higher N₂O$_{poreair}$ concentrations and N₂O$_{emitted}$ fluxes in association with lower $\delta^{15}$N was observed in the DS-AWD treatment. In the WS treatments, high N₂O$_{emitted}$ fluxes on June 23$^{rd}$, following the second fertilization, were associated with lower $\delta^{15}$N (Fig. 3), this was not the case for a high flux in the WS-AWD on June 17$^{th}$. N₂O$_{poreair}$ at 5cm in the WS-AWD treatment tended to be higher in concentration and lower in $\delta^{15}$N relative to other depths, however, in general a consistent relationship between concentration and $\delta^{15}$N was less evident in the two WS treatments. On average, the $\delta^{15}$N of N₂O$_{emitted}$ was lower relative to N₂O$_{poreair}$ in the DS-AWD treatment. In contrast, in the WS treatments N₂O$_{emitted}$ was depleted in $^{15}$N relative to N₂O$_{poreair}$ at all depths only immediately before and after the second fertilization. In these treatments, $\delta^{15}$N-N₂O$_{poreair}$ was generally lower at 5 cm relative the other depths but tended to increase and reach similar values as the other depths over the experimental period. As a result, N₂O$_{emitted}$ was often enriched in $^{15}$N relative to N₂O$_{poreair}$ at 5 cm in these treatments, particularly in the WS-AWD treatment.

#### 3.3.2 $\delta^{18}$O-N₂O

As with $\delta^{15}$N, $\delta^{18}$O isotope ratios spanned a large range, particularly in the emitted N₂O (Fig. 3). $\delta^{18}$O-N₂O$_{poreair}$ in the DS-AWD followed a temporal pattern similar to $\delta^{15}$N and similarly, $\delta^{18}$O was generally lower in N₂O$_{emitted}$ relative to N₂O$_{poreair}$. The highest $\delta^{18}$O-N₂O$_{poreair}$ was seen in the DS-AWD treatment at moderate N₂O$_{poreair}$ concentrations where $\delta^{18}$O isotope ratios were higher than other concentrations in the DS-AWD or any concentration in the WS treatments. These samples were also nearly always taken from 12.5 or 25 cm. In all treatments, lower $\delta^{18}$O values were observed in N₂O$_{poreair}$ and N₂O$_{emitted}$ on the first day of sampling, global mean of $35.1 \pm 1.1$ and $29.6 \pm 1.7$‰ relative to $46.9 \pm 0.4$ and $43.9 \pm 1.7$‰, respectively. Otherwise, no distinct patter with depth, time, or concentration was observed in the WS treatments.

#### 3.3.3 SP-N₂O

The SP of N₂O$_{emitted}$ ranged from $4.5 \pm 0.4$ to $25.6 \pm 8.1$‰, from $2.9 \pm 1.0$ to $37.2$‰ (un-replicated) and from $5.8 \pm 0.6$ to $40.6 \pm 12.4$‰, in the DS-AWD, WS-AWD, and WS-FLD treatments, respectively (Fig. 3). In contrast to $\delta^{15}$N and $\delta^{18}$O isotope ratios, the SP-N₂O$_{poreair}$ tended to increase with time, but only in the WS treatments. As with $\delta^{15}$N-N₂O and $\delta^{18}$O-N₂O, moderate and lower concentration N₂O$_{poreair}$ samples showed higher SP values relative to higher concentration N₂O$_{poreair}$ samples. For example, two days after the second fertilizer application (June 23$^{rd}$), SP values decreased in conjunction with increased N₂O$_{poreair}$ concentrations in the DS-AWD treatment. On this date mean SP values at 5 cm demonstrated the largest treatment differences with values of: $0.7 \pm 4.5$, $27.6 \pm 2.1$, and $39.9 \pm 2.7$‰ in the DS-AWD, WS-AWD, and WS-FLD treatments, respectively. On this date, the pattern between the treatments was consistent throughout the three depths.

### 3.3.4 Relationships between $N_2O$ isotope ratios

Considering all depths and emitted data together, $\delta^{18}O$-$N_2O$ significantly and positively correlated with $\delta^{15}N$-$N_2O$ and SP across all treatments. The slope of $\delta^{18}O$-$N_2O$ vs. $\delta^{15}N$-$N_2O$ was 0.67, 0.28, and 0.52 (Fig. S5) and 0.67, 0.54 and 0.31 for SP vs. $\delta^{18}O$-$N_2O$ in the DS-AWD, WS-AWD, and WS-FLD treatments, respectively (Fig. 4a). There was no correlation between SP and $\delta^{15}N$-$N_2O$ in the two WS treatments, but a positive correlation for the DS-AWD was found, with a slope of 0.62 (Fig. 4b). Examining these relationships by depth, we saw the strongest relationship and highest slope in the $N_2O_{emitted}$ and at 25 cm for $\delta^{18}O$-$N_2O$ vs. $\delta^{15}N$-$N_2O$ (Fig. S5). While the SP vs $\delta^{18}O$-$N_2O$ showed no correlation among the surface fluxes in the WS treatments, the two isotope ratios were positively correlated in $N_2O_{poreair}$ at all depths and treatments (Fig. S6). A contrasting relationship between SP and $\delta^{15}N$-$N_2O$ was observed for the WS-FLD treatment in the $N_2O_{emitted}$ and $N_2O_{poreair}$ where the two isotope ratios were negatively correlated in $N_2O_{emitted}$ and positively in $N_2O_{poreair}$ (Fig. S7).

### 3.4 $NO_3^-$ and $NH_4^+$ concentrations and isotope ratios

### 3.4.1 Spatiotemporal trend in $NO_3^-$ and $NH_4^+$ concentration and $\delta^{15}N$ and $\delta^{18}O$ isotope ratios

In all treatments, pore water $NH_4^+$ concentrations were highest at 5 cm relative to the other depths (Fig. 2). In the DS-AWD treatment concentrations were almost null prior to the second fertilization, remaining below 0.85 mg $NH_4^+$-N $L^{-1}$ across all depths. Following this fertilization, concentrations increased at all depths, most notably at 5 cm. An opposing pattern was observed in the WS treatments where $NH_4^+$ was nearly always significantly higher than in DS-AWD for each corresponding depth leading up to the second fertilization, but dropped to near zero following the fertilization. Nitrate concentrations were exclusively less than 1.5 mg $NO_3$-N $L^{-1}$ in both WS treatments throughout the experimental period. In sharp contrast, $NO_3^-$ concentrations in the DS-AWD were at times more than 75 times higher than in WS treatments, peaking on June 1st at $113.6 \pm 22.4$ mg $NO_3$-N $L^{-1}$. Following this spike, concentrations steadily declined and dropped to null following the second fertilization.

### 3.4.2 $\delta^{15}N$-$NO_3^-$ , $\delta^{15}N$-$NH_4^+$ and isotope enrichment factors: $\Delta^{15}N_{N2O/NO3}$ and $\Delta^{15}N_{N2O/NH4}$

Concentrations of $NO_3^-$ or $NH_4^+$ were often too low for isotope measurements. Hence, we could only obtain sufficient replication for statistical analysis across depths and treatments on five days for $NO_3^-$ (May 24th, 27th, June 1st, 14th, 23rd) and two days for $NH_4^+$ (May 24th and June 23rd) (Fig. S9). Daily mean $\delta^{15}N$-$NO_3^-$ ranged from -4.3 to 28.3‰ across all treatments and depths. In the DS-AWD treatment a consistent depth pattern was observed with $^{15}N$ enrichment of $NO_3^-$ at 25 cm > 12.5 cm = 5 cm. $\delta^{15}N$-$NO_3^-$ increased with time at 5 cm, rising from $-4.3 \pm 1.5$‰ to $22.0 \pm 4.9$‰. Significant treatment and depth differences were observed on May 24th, 27th and June 1st, but no differences were observed on later dates, June 14th or 23rd. Following the second fertilizer application, $\delta^{15}N$-$NO_3^-$ values in the DS-AWD treatment rose by approximately 10‰ at all depths. Daily mean $\delta^{15}N$-$NH_4^+$ ranged from -6‰ to 15.2‰ (Fig. S9). Averaging across the experimental period and depths, mean $\delta^{15}N$ values of $NO_3^-$ and $NH_4^+$ were similar, 8.4 and 7.0‰, respectively (Table S5). There was no evident temporal or depth trend in $\delta^{15}N$-$NH_4^+$ in any of the treatments. The only significant difference was lower $\delta^{15}N$-$NH_4^+$ in the DS-AWD on June 23rd. $\delta^{15}N$-$NO_3^-$ values positively correlated to $N_2O_{poreair}$ concentrations in the DS-AWD and WS-FLD treatments and were negatively

correlated to $NO_3^-$ concentrations and to $\delta^{15}N\text{-}NH_4^+$ in the DS-AWD treatment (Table 4). $\delta^{15}N\text{-}NH_4^+$ was negatively correlated to $N_2O_{poreair}$ concentrations and $NH_4^+$ concentrations and positively to $\delta^{15}N\text{-}N_2O_{poreair}$ in the DS-AWD treatment.

Largely reflecting the depth pattern of $\delta^{15}N\text{-}NO_3^-$ in the DS-AWD, the calculated $\Delta^{15}N_{N2O/NO3}$ tended to be highest at 5 cm, mean $-7.2 \pm 2.7‰$, while mean values at 12.5 and 25 cm were slightly lower, $-9.5 \pm 2.0$ and $-16.0 \pm 2.1‰$, respectively (Fig. S9). At 5 cm $\Delta^{15}N_{N2O/NO3}$ values in the DS-AWD were significantly higher than in the WS treatments; at 12.5cm they tended to be higher as well but the difference was not significant. Two days after the second fertilizer application, the $\Delta^{15}N_{N2O/NO3}$ in the DS-AWD markedly decreased at all depths to a treatment mean of $-23.6 \pm 2.6‰$. In comparison, WS treatment $\Delta^{15}N_{N2O/NO3}$ values rose one (WS-FLD) or two (WS-AWD) days following the fertilization. In the WS-FLD, the increase in $\Delta^{15}N_{N2O/NO3}$ values lasted only one day; unfortunately low $NO_3^-$ concentrations precluded $\delta^{15}N\text{-}NO_3^-$ analysis on many dates making temporal patterns difficult to observe. Mean depth by treatment isotope effects calculated relative to $\delta^{15}N\text{-}NH_4^+$ ($\Delta^{15}N_{N2O/NH4}$) were $-12.7 \pm 3.2‰$, $-24.5 \pm 2.6‰$ and $-20.6 \pm 2.2‰$ at 5 cm; $-9.9 \pm 4.0‰$, $-12.8 \pm 2.8‰$ and $-15.9 \pm 1.9‰$ at 12.5 cm; $-17.0 \pm 5.9‰$, $-6.4 \pm 1.7‰$ and $-5.8 \pm 2.7‰$ at 25 cm for DS-AWD, WD-AWD and WD-FLD, respectively. Data for $\Delta^{15}N_{N2O/NH4}$ was scarce in the DS-AWD treatment due to low $NH_4^+$ concentrations, in the WS treatments $\Delta^{15}N_{N2O/NH4}$ increased with depth, but these differences were not significant.

$\delta^{18}O\text{-}NO_3^-$ was significantly depleted in the DS-AWD treatment relative to both WS treatments (Fig. S9). Prior to the second fertilization, values were remarkably consistent in the DS-AWD at all depths, ranging from 0.1 to 7.5‰. Two days after this fertilizer application, $\delta^{18}O\text{-}NO_3^-$ rose to a mean of 7.6‰ across depths. In comparison the $\delta^{18}O\text{-}NO_3^-$ of both WS treatments was more variable between sampling dates, fluctuating between 12.2 to 38.8 and 10.4 to 32.7‰ leading up the second fertilization in the WS-AWD and WS-FLD, respectively. Two days after the second fertilizer application values rose to a mean of 23.7 and 27.4‰ across depths in the WS-AWD and WS-FLD, respectively. We calculated the net isotope effect for $\delta^{18}O\text{-}N_2O$ relative to water ($\Delta^{18}O_{N2O/H2O}$). The $\Delta^{18}O_{N2O/H2O}$ in all treatments and depths tended to rise over the course of the measurement period, with the most consistent rise observed at 5 cm. Here values rose from a global mean of $43.8 \pm 1.0‰$ on May 20th to $58.5 \pm 1.0‰$ on June 30th. There was a pattern of higher $\Delta^{18}O_{N2O/H2O}$ in the DS-AWD treatment relative to the two WS treatments. A drop in $\Delta^{18}O_{N2O/H2O}$ of ~ 10‰ was observed in all depths on June 23rd, two days after the second fertilization with urea, in the DS-AWD only.

**3.5 SP x $\delta^{18}O\text{-}N_2O$ two endmember mixing model to estimate $N_2O$ reduction, source contributions, and $N_2O$ reduction**

To further quantitatively interpret our isotope ratio data, we employed a graphical two end-member mixing model (Lewicka-Szczebak et al., 2017), based on the relationship between SP and $\delta^{18}O\text{-}N_2O$ (Fig. 1 and 4). Data was modeled for open and closed fractionation dynamics under two scenarios. In sc1 reduction of $N_2O$ from the denitrification/nitrifier-denitrification endmember pool occurs prior to mixing with nitrification/fungal denitrification derived $N_2O$; in sc2, mixing of $N_2O$ from both endmember pools occurs before reduction. For sc2 our model yielded implausible results for the contribution of denitrification/nitrifier-denitrification to $N_2O$ emissions in about 90% and

20% of observations under open and closed system dynamics, respectively (Table S2). The poorer outcomes from sc2 in the open system indicate that the assumptions underlying this scenario are likely false in open systems or vice versa. In order to have comparable data between open and closed systems we discuss only results coming from sc1 simulations.

Temporal trends in the gross rates of $r$N$_2$O (extent of N$_2$O reduction) predicted by open and closed system N$_2$O fractionation were nearly identical (Fig. 5b). Gross $r$N$_2$O was estimated to be higher (i.e. lower N$_2$O reduction) under closed system fractionation dynamics. In reality, it can be assumed that neither perfect open or closed systems exist in nature and processes likely reflect a mixture of these dynamics. The use of one or the other case may bias results,

therefore we chose to take the mean of the two systems to estimate N$_2$O reduction, nitrification/fungal denitrification and denitrification/nitrifier-denitrification derived N$_2$O emissions (Decock and Six, 2013b;Wu et al., 2016). Due to a disproportionate number of missing values at 25 cm in the two WS treatments, we chose not to include data from this depth in our analysis and discussion. Therefore, further values refer to the mean of open and closed systems and N$_2$O$_{emitted}$ or N$_2$O$_{poreair}$ at 5 cm and 12.5 cm unless explicitly stated otherwise. Gross $r$N$_2$O fractions tended to be higher

in N$_2$O$_{emitted}$ (treatment means 0.14 to 0.19) relative to the subsurface (treatment means 0.06 to 0.15). While water management treatment had a significant effect on process contributions to N$_2$O$_{emitted}$ and N$_2$O$_{poreair}$ (Table 5), significant interactions with depth and date were observed. Gross $r$N$_2$O fractions in N$_2$O$_{poreair}$ were significantly lower in the DS-AWD relative to the WS-FLD on six of 15 days, with the WS-AWD falling in between. In the N$_2$O$_{emitted}$, the opposite pattern was mostly observed with gross $r$N$_2$O fractions often being higher in the DS-AWD than one or

the other WS treatments, significantly so on four of 15 days. Aggregated across depths, the contribution of denitrification/nitrifier-denitrification to N$_2$O$_{poreair}$ were higher in the DS-AWD relative to one or both WS treatments on four dates and lower on three dates (Fig. 5a). The mean contribution of denitrification/nitrifier-denitrification to N$_2$O$_{emitted}$ ranged from 43 to 49% in all treatments (Fig. 6). Denitrification/nitrifier-denitrification contributions to N$_2$O$_{emitted}$ were higher in the DS-AWD relative to the WS treatments on June 9[th] and 23[rd] and relative to WS-AWD

only they were also higher on June 28[th] and lower on June 21[st].

### 4 Discussion

#### 4.1 Patterns of N$_2$O$_{emitted}$, N$_2$O$_{poreair}$ and N$_2$O isotope ratios

In accordance with results from past studies (Miniotti et al., 2016;Peyron et al., 2016;Cai et al., 1997) and in line with our hypothesis, we observed higher N$_2$O emissions on most days in the DS-AWD relative to the two WS treatments

(Fig. 2). A belated divergence in water management between the WS-FLD and WS-AWD (Table 1), in addition to a relatively wet early summer, likely contributed to similar observed soil environmental conditions and N substrates among these two treatments. Therefore, given the similarities in soil conditions, it is not surprising that N$_2$O fluxes and isotope ratio differences between these two treatments were generally fewer than expected. The lower yield in the DS-AWD treatment likely contributed additional differences in pore water N concentrations because lower N

demand in this treatment should have resulted in higher soil N concentrations.

Mean daily $\delta^{15}N$, $\delta^{18}O$ and SP values of $N_2O_{emitted}$ and $N_2O_{poreair}$ per depth and treatment ranged from -27.9 to 12.3‰, 30.9 to 63.0‰ and -14.0 to 53.2‰, respectively (Fig. 3). These values are similar in magnitude to those observed by (Yano et al., 2014) in the early growing season of rice, where ranges of -24 to 6‰, 24 to 66‰ and 4 to 25‰ were reported. Our values are also similar in magnitude to those observed in other field studies which have included depth sampling (Koehler et al., 2012;Zou et al., 2014). Relative to these two studies we observed higher $\delta^{15}N$-$N_2O$ and both higher and lower SP ratios. This was likely due to a higher sampling frequency, which covered more variable soil environments and generally higher soil moisture in our study than in the others. For example, it has been shown that organic matter decomposition and DOC availability in rice systems can decline with the introduction of wet-dry cycles or dry seeding (Said-Pullicino et al., 2016;Yao et al., 2011); thus it is likely that conditions promoting complete denitrification declined in the AWD treatments. In contrast, saturated conditions favoring complete denitrification certainly prevailed in the WS treatments at times. Working in a denitrifying aquifer, (Well et al., 2012) observed very large ranges in $\delta^{15}N$ and SP ratios, varying from -55.4 to 89.4‰ and 1.8 to 97.9‰, respectively.

**4.2 Source partitioning N₂O production**

One method to source partition emissions is to calculate net isotope effects and compare these to literature values derived from controlled and pure culture experiments where isotope effects were determined for individual processes. The calculated $\Delta^{15}N_{N2O/NO3}$ in the DS-AWD treatment, with depth means of -7.2 to -16.0‰, was consistently much higher (i.e. less strong fractionation) than literature values reported for denitrification of $NO_3^-$, mean: -42.9 ± 6.3‰ (Denk et al., 2017)(Fig. S9). At 5 cm in the two WS treatments, the mean $\Delta^{15}N_{N2O/NO3}$ was lower than in the DS-AWD (-23.2 and -21.5 in the WS-AWD and WS-FLD, respectively), but still nearly 20‰ higher than literature values. In a rice system, (Yano et al., 2014) observed an $\Delta^{15}N_{N2O/NO3}$ of -6.7‰, thus very well within the range of our calculated $\Delta^{15}N_{N2O/NO3}$. Similarly, the global mean of our $\Delta^{15}N_{N2O/NH4}$ values was -14.8‰, thus on average much higher than those reported in the literature for nitrification, -46.9‰ (Sutka et al., 2006) or -56.6 ± 7.3‰ (Denk et al., 2017). For both isotope effects, similar scenarios may explain our high observed $\Delta^{15}N_X$ (i.e. low fractionation). Namely, i) non-steady state reactions, for example rapid refreshing of the $NO_3^-$ and $NH_4^+$ pools or near complete substrate consumption or ii) significant reduction of $N_2O$ serving to increase $\delta^{15}N$-$N_2O$ values and thereby reduce the net isotope effect.

Considering the moist conditions and high reduction rates, it seems most likely that strong $N_2O$ reduction was the largest contributor to the greater degree of isotopic discrimination observed. To check this, we estimated *initial* $\delta^{15}N$-$N_2O$ values before $N_2O$ reduction using our modeled $N_2O$ reduction fraction ($rN_2O$), measured $\delta^{15}N$-$N_2O$ values and a $^{15}N$ isotope effect during reduction of -6.6‰ (Denk et al., 2017) in the Rayleigh equation. We could then estimate amended $\Delta^{15}N_{N2O/NO3}$ values if $N_2O$ reduction effects were accounted for, from the difference between our *initial* $\delta^{15}N$-$N_2O$ estimates and $\delta^{15}N$-$NO_3^-$. These calculations yielded a $\Delta^{15}N_{N2O/NO3}$ from -25.0 to -36.5‰, -32.6 to -42.3‰ and -29.0 to -51.1‰ in the DS-AWD, WS-AWD and WS-FLD across depths (Table S6). These amended $\Delta^{15}N_{N2O/NO3}$ values do decrease and especially for the WS treatments, come relatively close to literature values for $\Delta^{15}N_{N2O/NO3}$

values during denitrification. Thus, significant $N_2O$ reduction can likely explain much of the high $\Delta^{15}N_{N2O/NO3}$ values observed, particularly in the WS treatments. Yet other factors were also likely at play to some degree. For example, in the DS-AWD, where we observed evidence of significant nitrification, it is quite possible to envision isolated enrichment of $NO_3^-$ in anaerobic microsites where $N_2O$ is produced, while the bulk soil $NO_3^-$ pool remained less enriched. It is also true that we could not always measure $\delta^{15}N$ values of $NO_3^-$ or $NH_4^+$ because the concentrations were too low, thus we could not calculate isotope effects. This highlights a persistent dilemma, which is true for all isotope ratios, that we cannot accurately measure isotope ratios at very low concentrations. Hence, until more sensitive methodologies are developed, in-situ measurements such as these will always be biased toward higher concentration scenarios where perhaps the strongest and most interesting effects of substrate enrichment are missed.

The use of any one isotope signature alone is confounded by overlap in the isotope effects between processes, unknown and possibly rapidly changing substrate $\delta$ values and the unknown contribution of $N_2O$ reduction effects. To overcome these drawbacks, graphical interpretations of dual $N_2O$ isotope ratios have been used in field studies to interpret datasets similar to ours (Well et al., 2012;Koehler et al., 2012). For a more quantitative assessment of source-partitioning, mixing models using a dual isotope approach can be used (Yano et al., 2014;Toyoda et al., 2011;Koba et al., 2009;Lewicka-Szczebak et al., 2017;Zou et al., 2014). In the subsequent analysis we employ both approaches using our samples values plotted in SP x $\delta^{18}O$ and SP x $\delta^{15}N$ space (Fig. 4 and Figs.S10-S12).

In both SP x $\delta^{18}O$ and SP x $\delta^{15}N$ plots our sample values mostly fell between the mixing and reduction lines predicted by either isotope relationship (Fig. 4) and somewhat surprisingly showed stronger enrichment, indicative of greater $N_2O$ reduction in the DS-AWD treatment relative to the WS treatments. In the DS-AWD and to a lesser extent in the WS-AWD treatment, high pore air $N_2O$ concentrations were associated with denitrification or nitrifier-denitrification, while mid-range concentrations were associated with a higher degree of $N_2O$ reduction and the lowest concentrations fell neatly in between. Similarly, in the WS-FLD treatment, denitrification or nitrifier-denitrification associated samples almost exclusively coincided with high $N_2O_{poreair}$. Most likely the moderate $N_2O_{poreair}$ concentrations derived from $N_2O$ reduction following high denitrification/nitrifier-denitrification production. This analysis is supported by data showing a trend of enrichment over the course of the measurement period (Fig. S10) and high WFPS values associated with the most enriched $N_2O_{poreair}$ in the DS-AWD (Fig. S12). All treatments showed an enrichment of SP with time (Fig. S10), but interestingly only in the DS-AWD did $\delta^{18}O$ and $\delta^{15}N$-$N_2O$ enrich over the course of the experiment. This may reflect an increase over time in $\delta^{15}N$ and $\delta^{18}O$ of $NO_3^-$, which was observed in the DS-AWD treatment, albeit not strongly (Fig. S9). More $NO_3^-$ was available for denitrification in the DS-AWD treatment, thus for greater enrichment of this pool to occur we propose that more $NO_3^-$ was trapped in denitrifying microsites as the soil dried or $O_2$ was consumed.

In the WS treatments we observed a minimized trend of $N_2O$ reduction compared to the DS-AWD treatment, more scattered high SP values and more values intermediate to the two end-member pools. These results may partially be explained by greater contributions from abiotic hydroxylamine decomposition (SP ~ 34-35‰, Heil et al. (2014)) or

fungal denitrification (SP ~ 35‰, Rohe et al. (2014)). Zhou et al. (2001) showed that fungal denitrification requires minimal oxygen to proceed, similarly Seo and DeLaune (2010) found that fungal denitrification dominated relative to bacterial denitrification at modest reducing conditions to weakly oxidizing conditions (Eh >250 mV). Indeed, there is some evidence that high scattered SP values corresponded to more moderate WFPS (70-90%) in the WS-FLD treatment (Fig. S12). Abiotic hydroxylamine decomposition requires nitrification for the production of $NH_2OH$, and iron or manganese (hyrdr)oxides as electron acceptors to proceed (Bremner et al., 1980). Given the moist conditions, nitrification rates were likely low in the WS treatments. Feasible co-occurrence of these species could really only occur directly in the rhizosphere of a flooded rice soil, were $O_2$ is transported to the immediate root zone by the aerenchyma. Tightly coupled nitrification-denitrification in the rhizosphere of rice plants has been shown before (Arth and Frenzel, 2000) as has coupling of nitrogen – iron transformations (Ratering and Schnell, 2000) but we cannot say the extent to which this may have occurred in our system.

It is necessary to contextualize $N_2O$ isotope data with our measured substrate concentrations and soil environmental data. Based on our observations of low $NH_4^+$ concentrations, high $NO_3^-$ concentrations, an Eh over 400 mV and WFPS often below 60% (5 cm) or below 85% (12.5 and 25 cm) in the DS-AWD treatment, we can safely deduce that extensive nitrification of either basal urea fertilizer or of indigenous soil N occurred in this treatment (Fig. 2). Furthermore, the $\delta^{18}O$-$NO_3^-$ in the DS-AWD treatment ranged from 0.1 to 14.8 (Fig. 7), thus falling in the range attributed to $NO_3^-$ produced from nitrification (Kendall and McDonnell, 2012). Additionally, we observed that both $\delta^{15}N$-$NO_3^-$ and $\delta^{15}N$-$NH_4^+$ were negatively correlated to substrate concentrations in the DS-AWD treatment, indicative of active consumption of both N substrates (Table 4). In the DS-AWD, there also was a positive correlation between $\delta^{15}N$-$NO_3^-$ and $N_2O_{poreair}$ but a negative correlation between $\delta^{15}N$-$NH_4^+$ and $N_2O_{poreair}$. The former likely indicates $N_2O$ production via denitrification and subsequent enrichment of the $NO_3^-$ pool. The latter is more difficult to interpret, but we attributed this to higher emissions associated with fresh inputs of $NH_4^+$ (from urea or mineralization) which should have a $\delta^{15}N$ value around 0‰. Together this data shows that coupled nitrification-denitrification was responsible for the majority of $N_2O$ emissions. Similar results were also reported by (Dong et al., 2012) for an AWD system. The separation of isotope ratios by date, $N_2O$ concentration and WFPS suggests that $NO_3^-$ produced early in the growing season was progressively denitrified and reduced over the course of the sampling period. Similarly, $N_2O$ produced early in the growing season may have been progressively reduced.

**4.3 Inferring the extent of $N_2O$ reduction**

It has been suggested that the slope of SP/$\delta^{18}O$, SP/ $\delta^{15}N$ and $\delta^{18}O$ $\delta/^{15}N$ or their isotope effects can be used to estimate the extent of $N_2O$ reduction (Jinuntuya-Nortman et al., 2008;Well and Flessa, 2009;Lewicka-Szczebak et al., 2017;Ostrom et al., 2007). However, many studies deriving these relationships have taken place under controlled conditions when $N_2O$ supply was often limited. Therefore fractionation following closed system dynamics would result in larger fractionation effects on the residual substrate than under open system dynamics. The positive and significant relationship between all isotopes and across all depths in the DS-AWD treatment suggests an influence of reduction at all depths. In contrast, in the WS treatments we observed no relationship between SP and $\delta^{18}O$ within

$N_2O_{emitted}$ (Fig. S7) and only a weak relationship between SP and $\delta^{15}N$ at 25 cm in the WS-AWD, and even a negative relationship between SP and $\delta^{15}N$ in the WS-FLD $N_2O_{emitted}$ (Fig. S8). The range of observed $\delta^{18}O/\delta^{15}N$ slopes, 0.21 to 0.90, (Fig. S5) were substantially lower than those observed in many $N_2O$ reduction studies (1.94 to 2.6; Jinuntuya-Nortman et al. (2008); Ostrom et al. (2007); Well and Flessa (2009); Lewicka-Szczebak et al. (2017)), but closer to the 0.45 slope observed by Yano et al. (2014) in an in situ rice field study. When a significant relationship was observed, overall or $N_2O_{poreair}$ SP/$\delta^{15}N$ slopes ranged from 0.49 to 0.83 (Fig. 4b). These slopes are either close to those of other field studies, 0.48 to 0.52 (Yano et al., 2014;Wolf et al., 2015) or intermediary between field studies and controlled $N_2O$ reduction studies, 0.59 to 1.01 (Well and Flessa (2009); Lewicka-Szczebak et al. (2017). From controlled $N_2O$ reduction studies, a SP/$\delta^{18}O$ slope between 0.2 to 0.4 has been observed (Jinuntuya-Nortman et al., 2008;Well and Flessa, 2009), thus in this case the $N_2O_{poreair}$ slopes observed in our study were substantially higher (Fig. 4a and Fig. S7). The lower overall SP and $\delta^{18}O$ slope in the WS treatments was due to inclusion of the $N_2O_{emitted}$ values, which individually showed no relationship in these treatments.

A deviation in slopes compared to those observed in controlled $N_2O$ reduction studies likely points to a growing influence of open system dynamics where substrates are continuously refreshed. It has been demonstrated that when mixing processes dominate over reduction processes, the SP/$\delta^{18}O$ slope rises (Lewicka-Szczebak et al., 2017). It is also plausible that high rates of oxygen exchange during denitrification served to partially mask an increase in $\delta^{18}O$-$N_2O$ values, resulting in the higher observed SP/$\delta^{18}O$ slopes or lower $\delta^{18}O/\delta^{15}N$ slopes. To estimate the extent of oxygen exchange with denitrification precursors (NOx) we plotted $\delta^{18}O$-$N_2O/\delta^{18}O$-$NO_3^-$ by $\delta^{18}O$-$H_2O/\delta^{18}O$-$NO_3^-$ following (Snider et al., 2009). The slope of this relationship ranged from 0.7 to 2.1 (data not shown). Thus we assume oxygen exchange was effectively 100% across treatments during denitrification. In summary, the observed positive relationships between the isotope pairs is indicative of an influential role of $N_2O$ reduction in the DS-AWD treatment. This is less clear in the WS treatments where relationships were more erratic, suggesting a stronger influence of changing nitrification and denitrification process rates or changing $\delta^{15}N$ of N substrates. It is likely that isotope ratios in the WS treatments were affected by near complete denitrification to $N_2$. Well et al. (2012) observed highly variable isotope ratios in a strongly denitrifying aquifer and concluded that $N_2O$ reduction was strongly progressed but variable. However, it should be noted that their system had abundant $NO_3^-$ while ours did not. The inconsistent relationships between $N_2O_{emitted}$ and $N_2O_{poreair}$ for SP/$\delta^{15}N$ and SP/$\delta^{18}O$ in the WS treatments and the stronger enrichment observed in the DS-AWD $N_2O_{emitted}$ (Fig. 4) demonstrate a disconnection between subsurface $N_2O_{emitted}$ and $N_2O_{poreair}$ across treatments. Such results suggest that $N_2O$ reduction may not have had as strong of an influence on the signature of $N_2O_{emitted}$ as it did on $N_2O_{poreair}$, particularly in the WS treatments. A de-coupling between subsurface $N_2O$ concentrations and surface emissions, and their isotope ratios has been observed in other studies (Van Groenigen et al., 2005;Goldberg et al., 2010a). This phenomenon is most simply explained by emitted $N_2O$ truly coming from a mix of sources and depths, while subsurface $N_2O$ is representative of a much smaller spatial zone and more likely to be dominated by one process. While difficult to practically measure, processes at shallow depths above 5 cm, were also likely influential to surface emissions.

**4.4 Complementary evidence from a two endmember mixing model approach**

To quantitatively estimate the extent of $N_2O$ reduction (gross $rN_2O$), $N_2O$ production and reduction rates, and the contribution of denitrification to $N_2O$ emissions, we used an open and closed system two endmember mixing model based on SP-$N_2O$ and $\delta^{18}O$-$N_2O$ relationships. As described in section 2.7, we tested our models under two scenarios;

in scenario one (sc1) $N_2O$ is produced and reduced by denitrifiers before mixing with $N_2O$ derived from nitrification, in scenario two (sc2) $N_2O$ is produced from both processes, mixed, and then reduced (Fig. 1). While we could estimate gross $rN_2O$ and the fraction of denitrification from both scenarios, sc2 yielded mostly implausible solutions for the contribution of denitrification to $N_2O$ in open systems (Fig. S3 and Table S2). We thus conclude that the assumptions underlying this scenario in open systems were not valid in our system. In a closed system $N_2O$ is

progressively consumed and not replenished, resulting in a stronger isotope effect and faster enrichment of the remaining $N_2O$; thus a smaller degree of $N_2O$ reduction is needed to achieve an equivalent enrichment as in open systems. Our results for open and closed systems align well with this theory on $N_2O$ fractionation. However, we feel strongly that with in situ measurements in a heterogeneous soil environment, a combination of closed and open system dynamics likely exits, therefore the following interpretation of our data is based on an average of open and closed

system values. Given the lower moisture and evidence of extensive nitrification occurring in the DS-AWD treatment, we expected a higher contribution of nitrification/fungal denitrification in this treatment, coming from an increase in nitrification. However, this was not the case and denitrification/nitrifier-denitrification contributions tended to be higher in the DS-AWD treatment relative to WS treatments (Fig. 5a, Fig. 6). Treatment differences were significant in the surface fluxes, however there was a significant interaction with sampling day; there was no treatment effect on

denitrification contribution in the subsurface (Table 5). The equivalent or higher contributions of nitrification/fungal denitrification in the WS treatments (Fig. 6) are most easily explained by higher fungal denitrification; in their laboratory experiments, Lewicka-Szczebak et al. (2017) also observed relatively high fungal denitrification contributions under very wet conditions. Larger contributions from fungal denitrification would also help explain the less clear reduction trends as fungal denitrifiers are thought to largely produce $N_2O$ as an end-product rather than $N_2$.

It should be noted that due to low surface fluxes or $N_2O_{poreair}$, we had fewer data points in the WS treatments. Previous studies have attributed significant amounts of $N_2O$ emissions in paddy systems to nitrification in periods of low soil moisture (Lagomarsino et al., 2016;Verhoeven et al., 2018). Yet, such studies were not able to quantitatively source-partition emissions. Given our results here, it is possible that $N_2O$ produced either via nitrifier-denitrification or coupled nitrification-denitrification has been previously underestimated.

The modeled gross $rN_2O$ fractions indicate high levels of $N_2O$ reduction for all treatments and depths, ($rN_2O$: 0.06 to 0.19) even in the DS-AWD where soil moisture was frequently below 60% at 5 cm (Fig. 2). These results are at first surprising, but there is still much we do not know about subsurface $N_2O$ production and consumption. Direct measurements of $N_2O$ reduction at depth are few. Using membrane inlet mass spectrometry, (Zhou et al., 2017)

detected higher $N_2O$ reduction to $N_2$ in paddy soil water at 20 cm versus 60 or 80 cm and could relate this to higher DOC concentrations at 20 cm. Other studies suggest high subsurface $N_2O$ reduction based on the inference of declining $N_2O$ concentration accompanied by isotope enrichment moving up a soil profile (Goldberg et al.,

2008;Clough et al., 1998;Van Groenigen et al., 2005).  We are also methodologically limited by our inability to measure $N_2O$ isotopes at near, or complete $N_2O$ reduction because there is too little remaining $N_2O$ to measure.  We assume this was more often the case in the WS treatments, therefore we postulate that the signature of $N_2O$ reduction was stronger in the DS-AWD largely because there was more $N_2O$ left to measure.  In their experiments to validate the mixing model we used, (Lewicka-Szczebak et al., 2017) found that the model routinely underestimated gross $rN_2O$ rates relative to measured rates in an oxic mineral soil, but performed better under anoxic conditions and in an organic soil.  Therefore, an underestimation of $rN_2O$ rates, particularly in the DS-AWD treatments, remains possible.  However, considering the strong indication of $N_2O$ reduction from other isotope relationships (i.e. SP and $\delta^{15}N$ and $\delta^{15}N$ and $\delta^{18}O$) we believe that subsurface $N_2O$ reduction rates were simply high in our system, regardless of water management.

In the subsurface, the contribution of denitrification/nitrifier-denitrification to $N_2O$ concentrations was positively correlated to $N_2O_{poreair}$ concentrations and WFPS in all treatments, indicating an increasing contribution of denitrification/nitrifier-denitrification at times of higher $N_2O$ production in conjunction with rising soil moisture (Table 6).  In the two AWD treatments, the contribution of denitrification/nitrifier-denitrification negatively correlated to $\delta^{15}N$ signature of $N_2O_{poreair}$ and $N_2O_{emitted}$ (DS-AWD only).  Many studies have demonstrated that high subsurface $N_2O$ production is correlated to depleted $\delta^{15}N$-$N_2O$ (Goldberg et al., 2008;Goldberg et al., 2010b;Van Groenigen et al., 2005).  These results further support the conclusion that high $N_2O_{poreair}$ and $N_2O_{emitted}$ were produced from denitrification/nitrifier-denitrification associated with more depleted $\delta^{15}N$-$N_2O$.  Higher gross $rN_2O$ (less $N_2O$ reduction) was associated with higher $N_2O_{emitted}$ in all treatments and higher $N_2O_{poreair}$ (WS-AWD only), demonstrating that higher $N_2O$ resulted not only from increased denitrification/nitrifier-denitrification but also from a decrease in $N_2O$ reduction.  Interestingly, higher $rN_2O$ in $N_2O_{emitted}$ of the DS-AWD was also associated with higher WFPS.  Such a result can only be explained by a dependency of reduction on $N_2O$ production.  Overall, there was a negative relationship between $rN_2O$ and $\delta^{15}N$-$N_2O$, yet the relationship was not consistently strong or significant between treatments.  A negative relationship supports an isotope enrichment effect with greater $N_2O$ reduction.  Considering the above, it appears that maximum $N_2O$ production and emissions occurred during periods of increased contribution from denitrification/nitrifier-denitrification, which were accompanied by small declines in $N_2O$ reduction.  These relationships were most robust in the DS-AWD treatment.  Correlations within the $N_2O_{emitted}$ dataset were undoubtedly affected by lower data availability, particularly in the WS treatments, and should be taken with caution.  Despite the high estimates of $N_2O$ reduction for all treatments, we still observed relevant contributions from nitrification/fungal denitrification on many dates (Fig. 6).  Nevertheless, the highest fluxes in the DS-AWD aligned with higher contributions from denitrification/nitrifier-denitrification, while the highest fluxes in the WS treatment had nitrification/fungal denitrification contributions of ca. 50%.  In the WS treatments we again postulate that fungal denitrification rates increased because conditions were not ideal for high nitrification.  Studies have shown that fungal denitrification and co-denitrification can play a significant role in soil $N_2$ and $N_2O$ emissions from soil (Long et al., 2013;Laughlin and Stevens, 2002).

From our modeling results we could estimate $N_2$ production or emissions based on our calculated $N_2O$ reduction rates (Fig.S13). Due to poor data availability and high variability we could neither confidently estimate $N_2$ production at 25 cm nor surface $N_2$ emissions on many dates of the WS treatments, but we have more confidence in the estimates obtained for the DS-AWD treatment. Mean daily $N_2$ emissions found in our study were $236 \pm 53$ (n=43), $194 \pm 37$ (n=41) and $197 \pm 35$ (n=31) g N ha$^{-1}$ d$^{-1}$ in the DS-AWD, WS-AWD and WS-FLD, respectively. To our knowledge only one other study by (Yano et al., 2014) has conducted similar calculations to estimate $N_2$ emissions in rice systems from isotope ratios. The authors also found high rates of $N_2O$ reduction, around 80 to 85%, corresponding to an $rN_2O$ of 0.15 to 0.20 and $N_2$ emissions between 0.1 to 422 µg N m$^2$ hr$^{-1}$ (or 0.024 to 101.4 g ha-1 d$^{-1}$). Therefore, the estimated extent of $N_2O$ reduction was quite similar to our surface emitted reduction rates, with somewhat lower $N_2$ emissions corresponding to somewhat lower $N_2O$ emissions. Using labeled $^{15}N$ urea, (Lindau et al., 1990) measured $N_2$ emissions of 254 g ha$^{-1}$ d$^{-1}$, while (Dong et al., 2012) observed similar rates of 194 g $N_2$-N ha$^{-1}$ d$^{-1}$ for an AWD treatment. Considering that these results only account for $N_2$ derived from fertilizer, the modeled mean daily $N_2$ emissions found in our study are plausible. Differences between the treatment means were not significant for $N_2O_{poreair}$ or $N_2O_{emitted}$ (p=0.431 and p=0.858), thus do not indicate a higher potential for $N_2$ losses in the WS treatments. We must reject our hypothesis that higher $NO_3^-$ in the WS-AWD relative to the WS-FLD would drive higher denitrification and $N_2$ losses because we observed no differences in final modeled $N_2$ production and $NO_3^-$ concentrations were essentially null for both WS treatments. Our results show there is promise for estimating $N_2$ emissions from $N_2O$ isotope ratios using simple models, but the precision of these estimates remains constrained the limitations discussed below.

All modeling attempts to date rely on isotope signatures and effects determined in laboratory studies and thus changes in these values in response to environmental or microbial population dynamics in the field remains a large question. As this was an in situ field experiment, conditions were not constant across treatments or throughout the sampling time frame, yet it has been shown that isotope effects, particularly for $N_2O$ reduction change with shifts in environmental conditions such as increasing water filled pore space (Jinuntuya-Northman et al., 2008). Therefore, the use of fixed isotope effects in our model is a simplification. Future modeling efforts may be improved by the incorporation of variable isotope effects based on soil moisture or $O_2$ for example. Careful, controlled experiments across a range of soils with different management histories are necessary to determine if consistent variation in isotope effects in relation to specific environmental parameters can be determined or if such parameters are site specific. The microbial $\delta^{18}O$ signature for denitrification used in our model were calculated relative to $\delta^{18}O$-$H_2O$. We therefore assumed complete exchange between $N_2O$ substrates, intermediaries and water during denitrification. We based this off of previous work showing that O exchange is high and that the isotope effect between water and $N_2O$ is relatively stable (Lewicka-Szczebak et al., 2016;Lewicka-Szczebak et al., 2017;Snider et al., 2013;Kool et al., 2007). In reality, results over time and between treatments may have been affected by varying degrees of $^{18}O$ exchange between $N_2O$, intermediaries and water and by variation in $\delta^{18}O$-$H_2O$ values. We recommend that future studies measure the $\delta^{18}O$-$H_2O$ to better constrain results. Modeling results would also be more robust if complete $\delta^{15}N$ -$N_2O$, -$NH_4^+$ and –$NO_3^-$

across treatments and times were available, allowing for complimentary modelling of SP x $^{15}$N(N$_2$O/NO$_3^-$ or N$_2$O/NH$_4$). Employing iterative simulation techniques where a range of literature values for N$_2$O signatures and isotope effects are used to draw from would help to highlight model sensitivity to specific isotope values and improve its accuracy. Lastly, more work needs to be done to validate results such as those generated here which rely on laboratory derived values, with complimentary measurements of microbial community dynamics, such as that by Snider et al. (2015).

## 5 Conclusions

The relatively dry conditions in the DS-AWD treatment and application of urea fertilizer led to extensive nitrification, subsequent denitrification and denitrification derived N$_2$O emissions. Even with evidence of nitrification and relatively aerobic conditions in the DS-AWD treatment, both graphical and two endmember mixing model results indicated significant N$_2$O reduction in all treatments and graphically most convincingly in the DS-AWD treatment. Treatment differences may also reflect paddy history as this was the 5$^{th}$ year of alternative water management at the site. Yields were also lower in the DS-AWD, which likely lowered N demand and increased soil N concentrations in this treatment. Differences between depths were often more evident in N$_2$O$_{poreair}$, NO$_3^-$, NH$_4^+$ and DOC concentrations than in N$_2$O isotope signatures at the various depths, particularly for the WS treatments. In the DS-AWD treatment, isotope signatures of $\delta^{18}$O-N$_2$O and SP values demonstrated notably lower values at 5 cm relative to other depths, mostly likely indicating higher N$_2$O production and less reduction in the upper layer. Overall, the highest N$_2$O production and emissions were associated with an increasing contribution from denitrification/nitrifier-denitrification accompanied by decreases in N$_2$O reduction in the AWD treatments. Our isotope data suggests that contributions from fungal denitrification to N$_2$O emissions may have increased in the WS-FLD treatment. The role of fungal denitrification in paddy rice systems should be further investigated with the use of fungal inhibitors. Surface emitted N$_2$O reduction rates were similar for all treatments, therefore our hypothesis of a greater potential for gaseous N$_2$ losses in the WS-AWD is refuted. Despite the difficulty in obtaining a full dataset for all treatments and the inherent spatiotemporal variability in the original measured fluxes, we came to good agreement with the magnitude of N$_2$ emissions reported from previous $^{15}$N labeled fertilizer studies. Thus natural abundance isotope methods do show promise for estimating N$_2$ emissions and closing N budgets, even without the $\delta^{15}$N of N substrates. Model results would likely improve with controlled incubations to determine site-specific isotope effects and whether these effects change in a consistent manner with specific environmental conditions. In saturated or partly saturated systems, future studies should probe the disconnection between subsurface and emitted N$_2$O isotopes by employing methods that allow for larger subsurface spatial integration along vertical and horizontal planes. It appears that to effectively manage N losses in alternative water management paddy systems inhibition of nitrification is necessary, particularly very early in the growing season when N availability exceeds crop N demand.

**Dataset availability**

Verhoeven, Elizabeth. (2018). CastelloD'Agogna_waterMgmt2015,2016_dataset (Version 1.0) [Data set]. Zenodo. http://doi.org/10.5281/zenodo.1251895

**Acknowledgements**

This work was financially supported by the Swiss National Science Foundation (40FA40_154246) through the Joint Programming Initiative on Agriculture, Food Security and Climate Change (FACCE-JPI). This work would not have been possible without the support and assistance of the staff at Ente Nazionale Risi in Castello D'Agogna, Italy, in particular Marco Romani, Elenora Miniotti and Daniele Tenni.

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

**Figure and table captions**

**Figure 1.** Mapping approach scheme used in the closed system modeling. Adapted from (Lewicka-Szczebak et al., 2017).

**Figure 2.** N$_2$O surface emissions, log$_{10}$ of dissolved and pore air N$_2$O concentrations, NH$_4^+$, NO$_3^-$, DOC, Eh and WFPS throughout the field measurement period in the three water management treatments (WS-FLD = water-seeding + conventional flooding; WS-AWD = water-seeding + alternate wetting and drying; DS-AWD = direct dry seeding + alternate wetting and drying). The dashed vertical line indicates the date of fertilization (60 kg urea-N ha$^{-1}$). Blue shaded areas represent periods of flooding, shaded areas that last only one day indicate a 'flush irrigation' = flooding for < 6 hrs. The error bars represent the standard error of the mean. Red and orange dashed vertical lines represent the date of seeding and fertilization in each treatment, respectively.

**Figure 3.** Time course of δ$^{15}$N-N$_2$O, δ$^{18}$O-N$_2$O and SP-N$_2$O in N$_2$O$_{emitted}$ and N$_2$O$_{poreair}$ across the three depths and water management treatments (WS-FLD = water-seeding + conventional flooding; WS-AWD = water-seeding + alternate wetting and drying; DS-AWD = direct dry seeding + alternate wetting and drying). The errors bars represent the standard error of the mean. Red and orange dashed vertical lines represent the date of seeding and fertilization in each treatment, respectively.

**Figure 4.** Graphical two-end member mixing plot after Lewicka-Szczebak *et al.* (2017) where sample values are plotted in SP x δ$^{18}$O-N$_2$O space (A) and two-end mixing plot after Toyoda *et al.* (2011) where sample values are plotted in SP x δ$^{15}$N-N$_2$O space (B). In panel (a) the black dots indicate the mean literature end-member values used in our modeling scenarios and the boxes represent a range of values derived from the literature attributed to each process, see section 2.7 and Table 2. To calculate the range of N$_2$O potentially produced by nitrification or denitrification in (B) we used the mean isotope effects, ε$^{15}$N$_{N2O/NO3}$ and ε$^{15}$N$_{N2O/NH4}$, reported in Denk *et al.* (2017) to represent denitrification and nitrification derived N$_2$O, respectively, and then added the minimum and maximum δ$^{15}$N-NO$_3^-$ and δ$^{15}$N-NH$_4^+$ values observed in each treatment (Supplementary Table 1.4). The linear relationship between each isotope pair is indicated in italics for all points together and for N$_2$O$_{poreair}$, only. The three water management treatments were: WS-FLD = water-seeding + conventional flooding; WS-AWD = water-seeding + alternate wetting and drying; DS-AWD = direct dry seeding + alternate wetting and drying.

**Figure 5.** Modeled denitrification/nitrifier-denitrification contribution and gross $r$N$_2$O of open (grey bars), closed (blue bars) and mean (purple points and line) systems predicted by a two-endmember mixing model using δ$^{18}$O-N$_2$O and SP values. For open and closed system dynamics, the shaded bars represent the standard deviation range for each treatment x depth combination. The purple error bars represent the standard deviation around the mean. Red and orange dashed vertical lines represent the date of seeding and fertilization in each treatment, respectively.

**Figure 6.** Estimated contribution of denitrification/nitrifier-denitrification and nitrification/fungal denitrification to N$_2$O surface emissions in the three water management treatments (WS-FLD = water-seeding + conventional flooding; WS-AWD = water-seeding + alternate wetting and drying; DS-AWD = direct dry seeding + alternate wetting and drying). Estimates were derived from the mean of open and closed dynamics in a two endmember mixing model

using $\delta^{18}O$-$N_2O$ and SP values. Red and orange dashed vertical lines represent the date of seeding and fertilization in each treatment, respectively.

**Figure 7.** Relationship of $\delta^{18}O$-$NO_3^-$ to $\delta^{15}N$-$NO_3^-$ in pore water samples of the three water management treatments (WS-FLD = water-seeding + conventional flooding; WS-AWD = water-seeding + alternate wetting and drying; DS-AWD = direct dry seeding + alternate wetting and drying). After Kendall and McDonnell (2012). The black arrow represents the trajectory of $NO_3^-$ reduction effects. The black asterisk signifies the $\delta^{18}O$ value atmospheric $O_2$ (25.3 ‰) while the dashed black line indicates the range of $\delta^{18}O$ in soil water. $\delta^{18}O$-$H_2O$ was not directly measured in our study. We assumed a value of -8.3‰ taken from an uncontained aquifer in the region by Rapti-Caputo and Martinelli (2009). The symbol colors indicate the concentration of $NO_3^-$ in each sample (mg L$^{-1}$).

**Table 1.** Dates of management activities during the experimental period in the three water management treatments (WS-FLD = water-seeding + conventional flooding; WS-AWD = water-seeding + alternate wetting and drying; DS-AWD = direct dry seeding + alternate wetting and drying).

**Table 2.** Endmember values used for modeling of the fraction of residual $N_2O$ not reduced (gross $r$$N_2O$) and the fraction of $N_2O$ + $N_2$ attributed to denitrification (gross frac$_{DEN}$ ) for both open and closed $N_2O$ reduction fractionation dynamics.

**Table 3.** Spearman correlations of $N_2O_{emitted}$ with $N_2O_{emitted}$ isotope ratios, $N_2O$ driving variables and $N_2O_{poreair}$ isotope ratios measured at 5 cm in the three water management treatments (WS-FLD = water-seeding + conventional flooding; WS-AWD = water-seeding + alternate wetting and drying; DS-AWD = direct dry seeding + alternate wetting and drying). Significance indicated by: **** <0.0001, *** < 0.001, **<0.01, *<0.05

**Table 4.** Spearman correlations between $\delta^{15}N$-$NO_3^-$ and $\delta^{15}N$-$NH_4^+$ with $N_2O_{poreair}$ concentration, $\delta^{15}N$-$N_2O_{poreair}$, $NO_3^-$ and $NH_4^+$ concentrations in the three water management treatments (WS-FLD = water-seeding + conventional flooding; WS-AWD = water-seeding + alternate wetting and drying; DS-AWD = direct dry seeding + alternate wetting and drying).

**Table 5.** ANCOVA results of modeled residual $N_2O$ not reduced (gross $r$$N_2O$), fraction of total $N_2$ + $N_2O$ production coming from denitrification (gross frac$_{DEN}$) and the fraction of $N_2O$ attributed to denitrification (DenContribution) derived from $N_2O_{emitted}$ and $N_2O_{poreair}$. The Y position was used a co-variate and represents the longitudinal position of each replicate within field.

**Table 6.** Spearman correlations between modeled $r$$N_2O$-gross, frac$_{DEN}$ –gross and *DenContribution* with soil environmental variables and inorganic N substrates and $\delta^{15}N$-$N_2O$. Results are for the mean of open and closed system dynamics. Subsurface correlations were performed on data aggregated across 5 and 12.5 cm depths. Significance indicated by: **** <0.0001, *** < 0.001, **<0.01, *<0.05

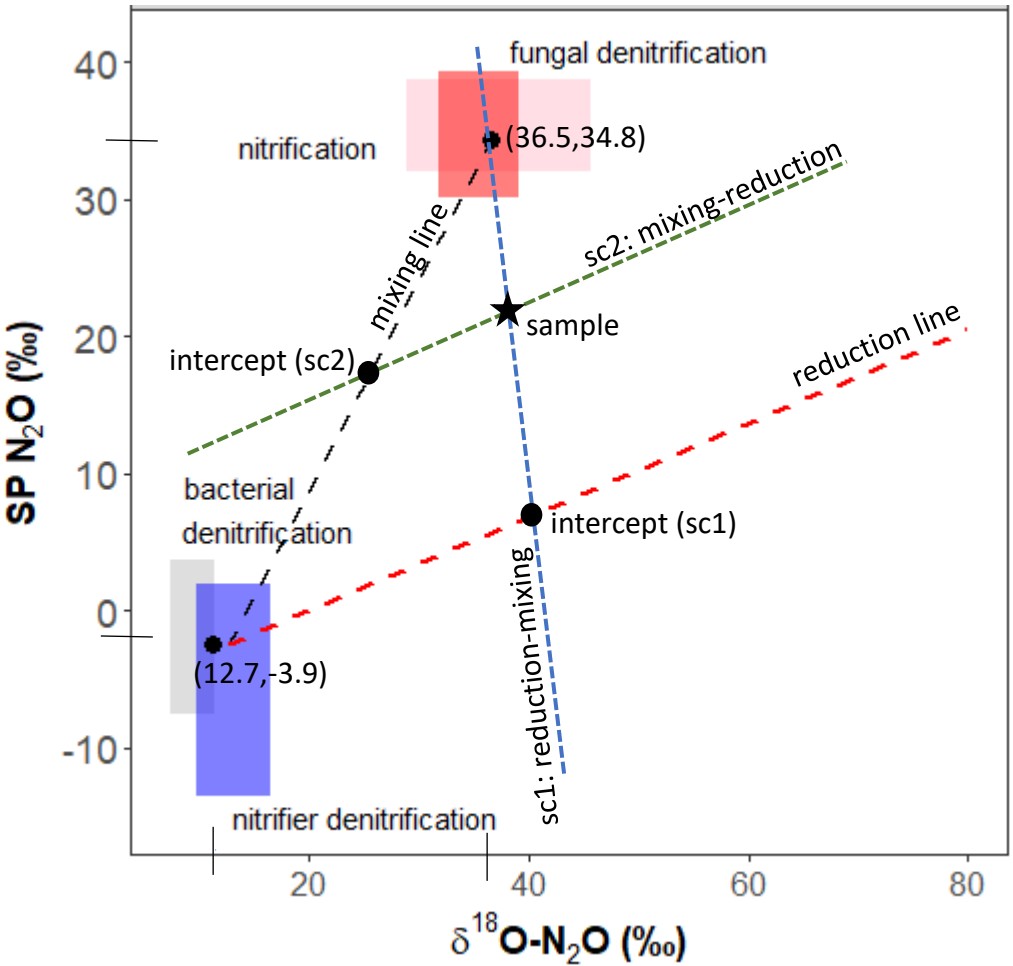

**Figure 1.**

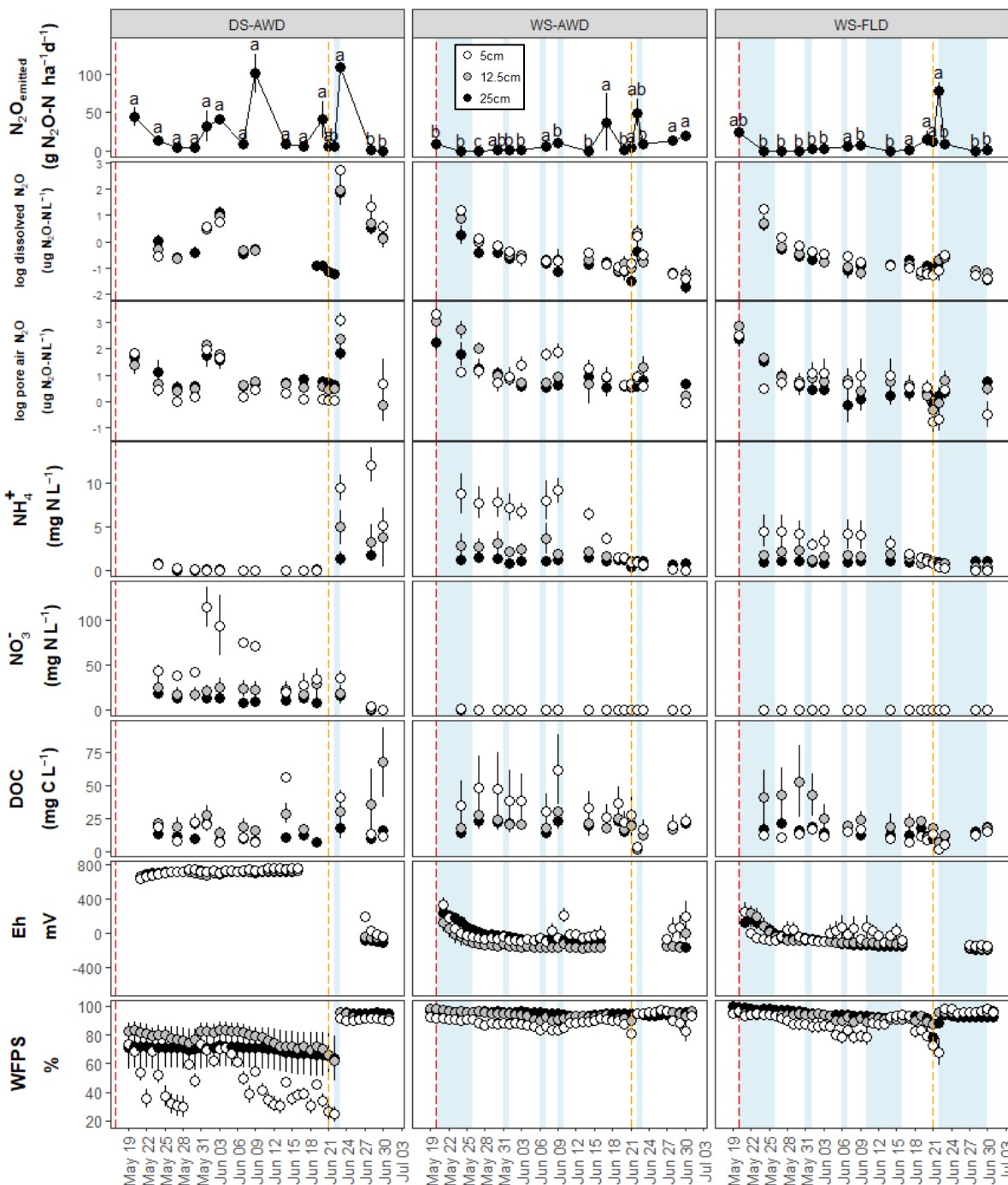

**Figure 2.**

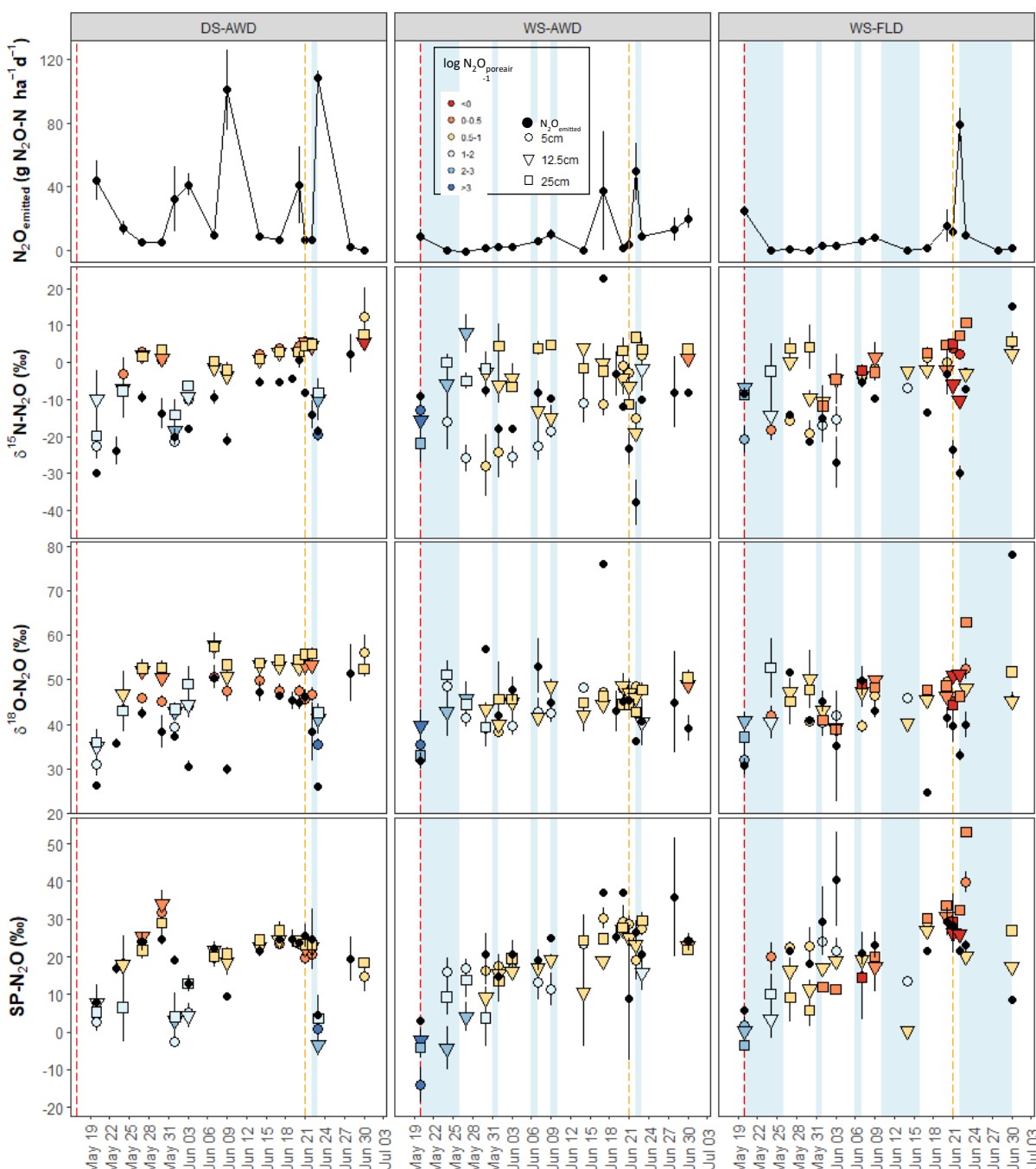

**Figure 3**.

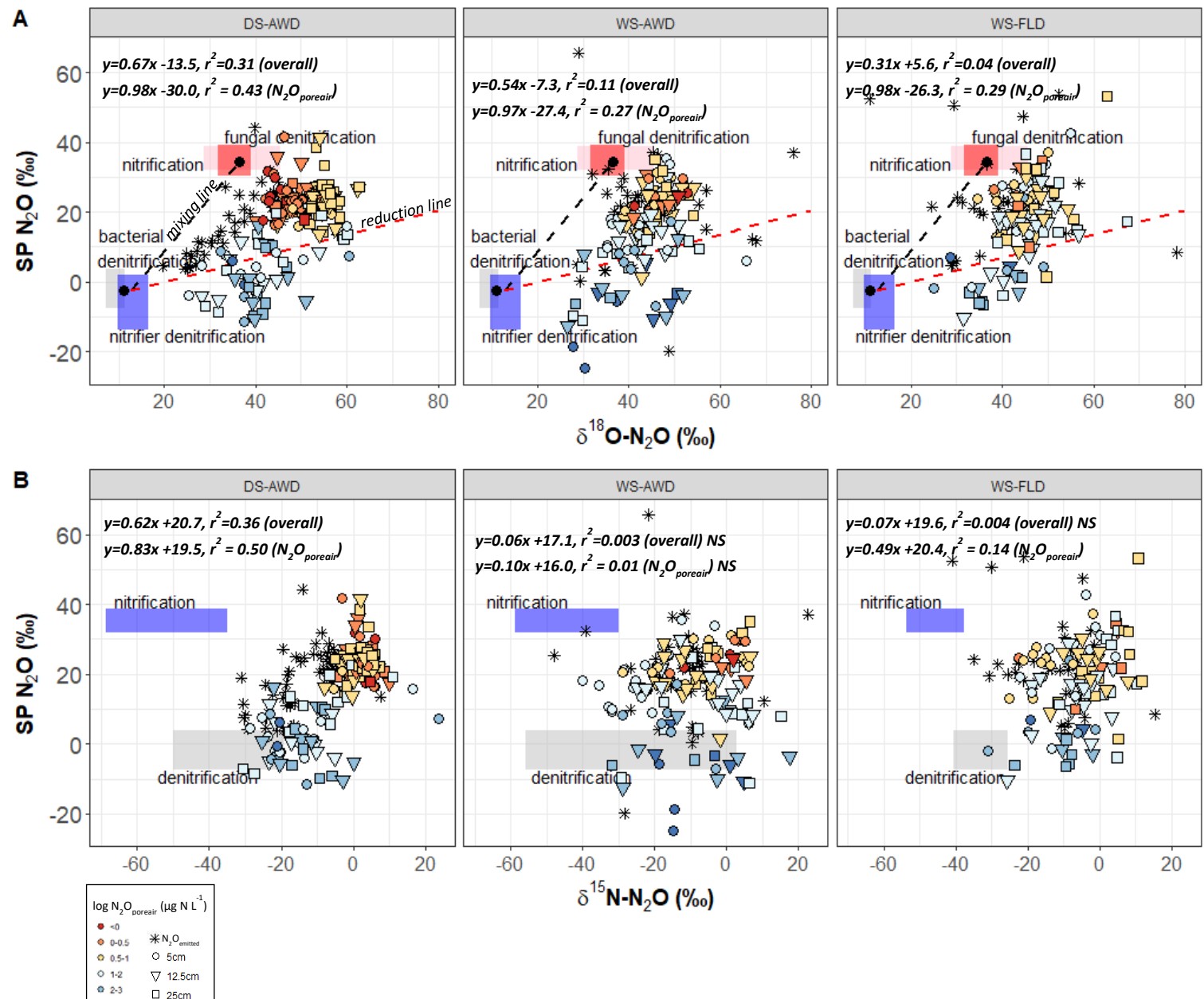

**Figure 4.**

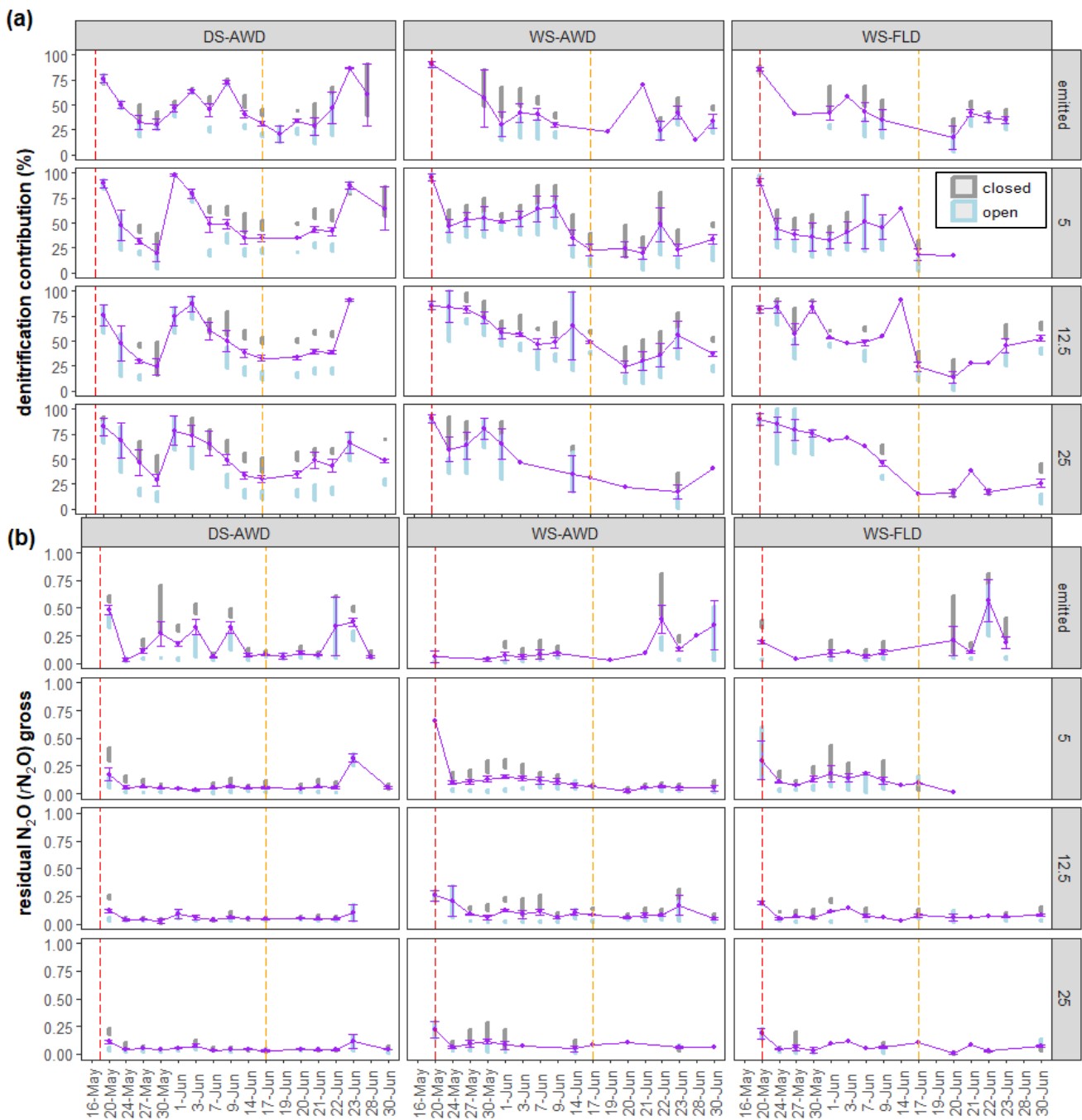

**Figure 5.**

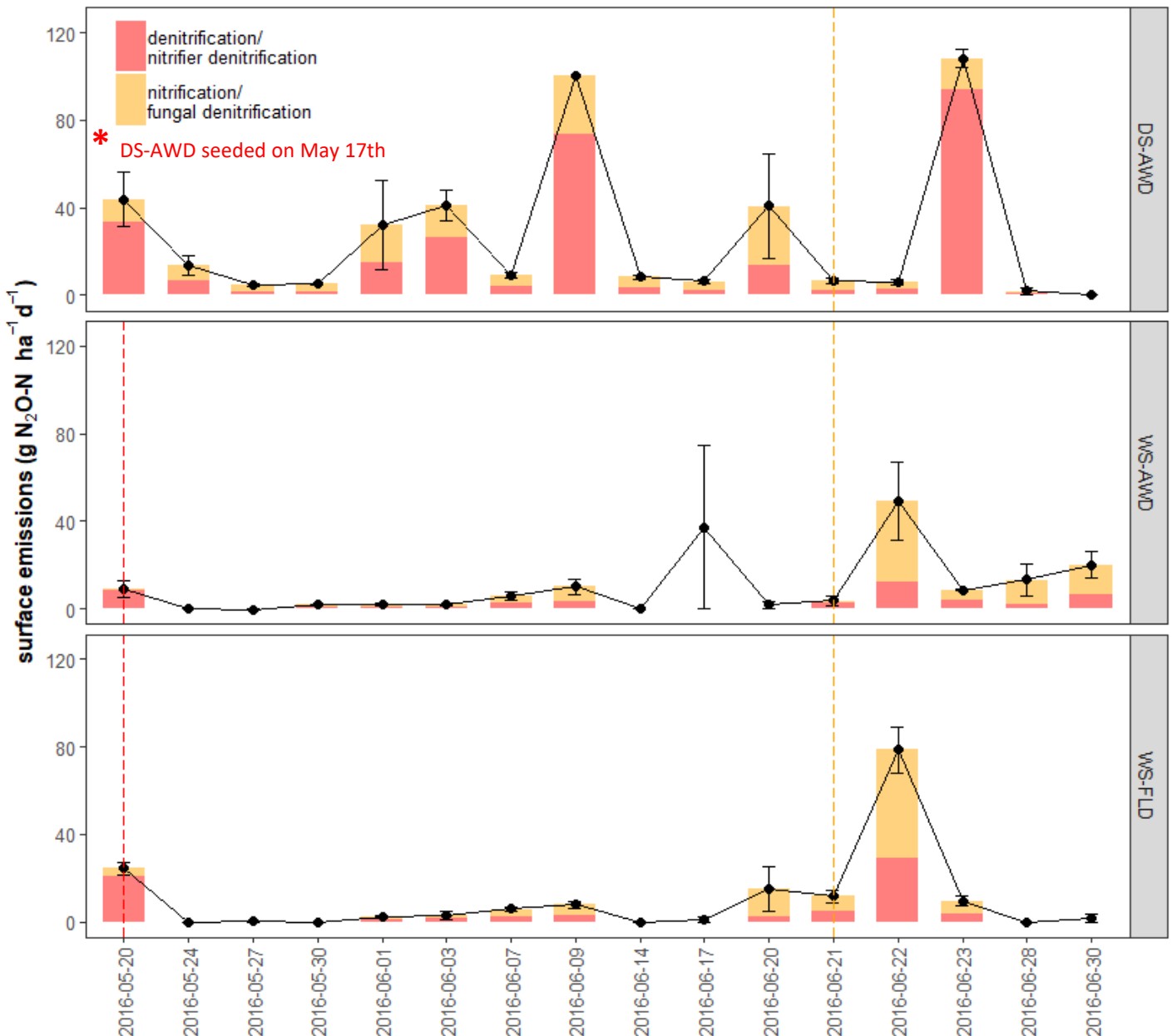

**Figure 6.**

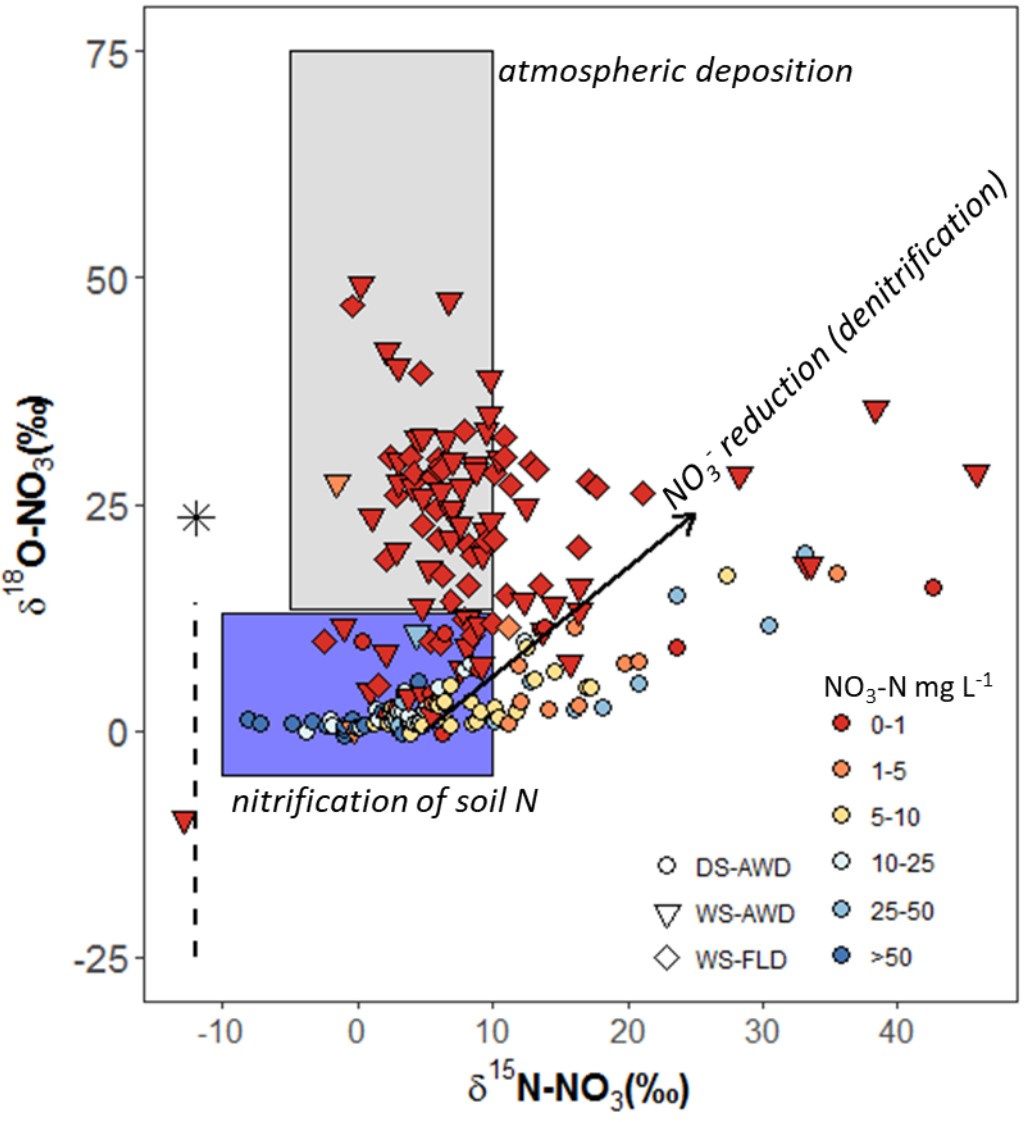

**Figure 7.**

**Table 1.** Dates of management activities during the experimental period in the three water management treatments (WS-FLD = water-seeding + conventional flooding; WS-AWD = water-seeding + alternate wetting and drying; DS-AWD = direct dry seeding + alternate wetting and drying).

| Management | WS-FLD | WS-AWD | DS-AWD |
|---|---|---|---|
| ploughing; leveling | 4-Apr; 12-Apr | 4-Apr; 12-Apr | 4-Apr; 12-Apr |
| Fertilization P-K | 13-May (14-28 kg ha$^{-1}$) | 13-May (14-28 kg ha$^{-1}$) | 13-May (14-28 kg ha$^{-1}$) |
| Fertilization N | 16-May (60 kg ha$^{-1}$) | 16-May (60 kg ha$^{-1}$) | 16-May (40 kg ha$^{-1}$) |
| Flooding | 19-May | 19-May | |
| Seeding | 20-May | 20-May | 17-May |
| Drainage | 26-May | 26-May | |
| Flush irrigation | 31-May;6-Jun | 31-May;6-Jun;10-Jun | |
| Flooding | 10-Jun | | |
| Drainage | 16-Jun | | |
| Fertilization N | 21-Jun (60 kg ha$^{-1}$) | 21-Jun (60 kg ha$^{-1}$) | 21-Jun (70 kg ha$^{-1}$) |
| Flooding | 22-Jun | | |
| Flush irrigation | | 22-Jun | 22-Jun |
| … | | | |
| Fertilization N | 14-July (40 kg ha$^{-1}$) | 14-July (40 kg ha$^{-1}$) | 14-July (50 kg ha$^{-1}$) |
| ... | | | |
| Harvest | 15-Sep | 15-Sep | 15-Sep |
| Yield (t/ha) | 8.9a | 8.2a | 6.6b |

**Table 2.** Endmember values used for modeling of the fraction of residual $N_2O$ not reduced (gross $rN_2O$) and the fraction of $N_2O + N_2$ attributed to denitrification (gross frac$_{DEN}$ ) for both open and closed $N_2O$ reduction fractionation dynamics.

| Process(s) | $\delta^{18}O$-$N_2O_{(x)}$ | $SP_{(x)}$ | references |
|---|---|---|---|
| denitrification, nitrifier-denitrification | 12.7 | -3.9 | $\delta^{18}O$ and SP: Lewicka-Szczebak *et al.* (2017) *$\delta^{18}O$ uncorrected for $\delta^{18}O$-$H_2O$ |
| nitrification, fungal denitrification | 36.5 | 34.8 | SP: Lewicka-Szczebak *et al.* (2017); $\delta^{18}O$: Sutka *et al.* (2006); Sutka *et al.* (2008); Frame and Casciotti (2010); Heil *et al.* (2014); Rohe *et al.* (2014); Maeda *et al.* (2015) |
| | $\epsilon^{18}O_{red}$ | $\epsilon SP_{red}$ | |
| $N_2O$ reduction | -15 | -5 | Lewicka-Szczebak *et al.* (2017) |

*Lewicka-Szczebak *et al.* (2017) originally report $\delta_0^{18}O$-$N_2O(N_2O/H_2O)$. Thus, to calculate a pure $\delta_0^{18}O$-$N_2O$, we added the $\delta^{18}O$-$H_2O$ value used in
5  our study, -8.3‰.

**Table 3.** Spearman correlations of $N_2O_{emitted}$ with $N_2O_{emitted}$ isotope ratios, $N_2O$ driving variables and $N_2O_{poreair}$ isotope ratios measured at 5 cm in the three water management treatments (WS-FLD = water-seeding + conventional flooding; WS-AWD = water-seeding + alternate wetting and
10  drying; DS-AWD = direct dry seeding + alternate wetting and drying). Significance indicated by: **** <0.0001, *** < 0.001, **<0.01, *<0.05

| | $N_2O_{emitted}$ | | | $\delta^{15}N$-$N_2O_{emitted}$ | | | $\delta^{18}O$-$N_2O_{emitted}$ | | | $\delta SP$-$N_2O_{emitted}$ | | |
|---|---|---|---|---|---|---|---|---|---|---|---|---|
| | WS-FLD | WS-AWD | DS-AWD | WS-FLD | WS-AWD | DS-AWD | WS-FLD | WS-AWD | DS-AWD | WS-FLD | WS-AWD | DS-AWD |
| $N_2O_{emitted}$ | | | | -0.16 | 0.03 | -0.51*** | -0.46** | -0.45** | -0.58**** | -0.42* | 0.36* | -0.68**** |
| $N_2O_{dissolved, 5cm}$ | -0.25 | 0.01 | 0.36 | 0.07 | -0.39* | -0.3 | 0.14 | -0.15 | -0.56* | -0.07 | 0.21 | -0.58* |
| $N_2O_{poreair, 5cm}$ | 0.00 | -0.05 | 0.48*** | 0.11 | 0.15 | -0.60**** | -0.29 | -0.11 | -0.64**** | -0.3 | -0.32 | -0.64**** |
| $WFPS_{5cm}$ | -0.23 | -0.02 | 0.31* | 0.25 | -0.02 | -0.49*** | -0.09 | -0.29 | -0.50**** | -0.22 | -0.3 | -0.64**** |
| $Eh_{5cm}$ | -0.03 | 0.15 | 0.25 | 0.05 | -0.09 | 0.15 | -0.03 | -0.29 | 0.26 | -0.02 | 0.44* | 0.22 |
| $DOC_{5cm}$ | -0.08 | -0.43** | -0.05 | 0.2 | 0.43** | 0.13 | 0.40* | 0.28 | -0.03 | -0.33 | 0.06 | -0.03 |
| $NO_3$-$N_{porewater, 5cm}$ | -0.21 | 0.1 | 0.52*** | -0.25 | -0.29 | -0.64**** | -0.23 | -0.15 | -0.27 | -0.13 | -0.11 | -0.21 |
| $NH_4$-$N_{porewater, 5cm}$ | -0.29* | -0.32* | -0.31 | 0.05 | -0.02 | 0.23 | 0.29 | 0.43** | 0.01 | 0.07 | -0.16 | -0.03 |
| $\delta^{15}N$-$N_2O_{poreair, 5cm}$ | 0.24 | 0.09 | -0.51**** | -0.02 | 0.07 | 0.71**** | 0.1 | -0.24 | 0.64**** | 0.1 | 0.1 | 0.65**** |
| $\delta^{18}O$-$N_2O_{poreair, 5cm}$ | -0.07 | 0.07 | -0.39** | -0.13 | -0.1 | 0.46*** | 0.02 | -0.03 | 0.48*** | 0.33 | 0.47** | 0.41** |
| $\delta SP$-$N_2O_{poreair, 5cm}$ | -0.27 | -0.1 | -0.55**** | 0.18 | -0.22 | 0.62**** | 0.14 | 0.21 | 0.49*** | 0.47* | 0.55** | 0.67**** |

**Table 4.** Spearman correlations between $\delta^{15}$N-NO$_3^-$ and $\delta^{15}$N-NH$_4^+$ with N$_2$O$_{poreair}$ concentration, $\delta^{15}$N-N$_2$O$_{poreair}$, NO$_3^-$ and NH$_4^+$ concentrations in the three water management treatments (WS-FLD = water-seeding + conventional flooding; WS-AWD = water-seeding + alternate wetting and drying; DS-AWD = direct dry seeding + alternate wetting and drying).

|  | $\delta^{15}$N-NO$_3^-$ | | | $\delta^{15}$N-NH$_4^+$ | | |
|---|---|---|---|---|---|---|
|  | DS-AWD | WS-AWD | WS-FLD | DS-AWD | WS-AWD | WS-FLD |
| $\delta^{15}$N-NO$_3^-$ |  |  |  | -0.54* | -0.03 | -0.05 |
| $\delta^{15}$N-NH$_4^+$ | -0.54* | -0.03 | -0.05 |  |  |  |
| N$_2$O$_{poreair}$ | 0.34** | 0.07 | 0.38** | -0.72*** | 0.04 | 0.22* |
| $\delta^{15}$N-N$_2$O$_{poreair}$ | 0.00 | 0.00 | -0.14 | 0.46* | -0.03 | 0.14 |
| NO$_3^-$ | -0.66**** | -0.01 | -0.28 | -0.41 | 0.11 | 0.27* |
| NH$_4^+$ | 0.01 | 0.13 | -0.06 | -0.54* | -0.23* | -0.12 |

**Table 5.** ANCOVA results of modeled residual N$_2$O not reduced (gross $r$N$_2$O), fraction of total N$_2$ + N$_2$O production coming from denitrification (gross frac$_{DEN}$) and the fraction of N$_2$O attributed to denitrification (DenContribution) derived from N$_2$O$_{emitted}$ and N$_2$O$_{poreair}$. The Y position was used a co-variate and represents the longitudinal position of each replicate within field.

|  | NumDF | N$_2$O$_{poreair}$ $r$N$_2$O-gross | N$_2$O$_{poreair}$ frac$_{DEN}$-gross | DenContribution (N$_2$O$_{poreair}$) | NumDF | N$_2$O$_{emitted}$ $r$N$_2$O-gross | N$_2$O$_{emitted}$ frac$_{DEN}$-gross | DenContribution (N$_2$O$_{emitted}$) |
|---|---|---|---|---|---|---|---|---|
| treatment | 2 | **0.004** | **<0.001** | 0.188 | 2 | 0.146 | 0.931 | **0.016** |
| day | 14 | **<0.001** | **0.001** | **<0.001** | 16 | **<0.001** | **<0.001** | **<0.001** |
| depth | 1 | **0.019** | **0.007** | **0.008** |  |  |  |  |
| Y position | 1 | 0.844 | **0.016** | 0.375 | 1 | 0.451 | 0.373 | 0.818 |
| trmt:day | 28 | **0.001** | **<0.001** | **<0.001** | 19 | **0.009** | **0.024** | **<0.001** |
| trmt:depth | 2 | 0.330 | 0.082 | 0.052 |  |  |  |  |
| day:depth | 14 | 0.185 | **<0.001** | **0.002** |  |  |  |  |
| trmt:day:depth | 23 | **0.022** | **0.047** | 0.189 |  |  |  |  |

**Table 6.** Spearman correlations between modeled $r$N$_2$O-gross, frac$_{DEN}$–gross and *DenContribution* with soil environmental variables and inorganic N substrates and $\delta^{15}$N-N$_2$O. Results are for the mean of open and closed system dynamics. Subsurface correlations were performed on data aggregated across 5 and 12.5 cm depths. Significance indicated by: **** <0.0001, *** < 0.001, **<0.01, *<0.05

| | frac$_{DEN}$ -gross | | | $r$N$_2$O - gross | | | *DenContribution* | | |
|---|---|---|---|---|---|---|---|---|---|
| | DS-AWD | WS-AWD | WS-FLD | DS-AWD | WS-AWD | WS-FLD | DS-AWD | WS-AWD | WS-FLD |
| | | | | | *subsurface* | | | | |
| [N$_2$O$_{poreair}$] | 0.34*** | 0.2 | 0.31* | 0.01 | 0.60**** | 0.17 | 0.67**** | 0.70**** | 0.59**** |
| WFPS | 0.21* | 0.21* | 0.39** | -0.11 | 0 | -0.06 | 0.34*** | 0.22* | 0.47*** |
| Eh | -0.04 | 0.01 | 0.01 | 0.04 | 0.04 | 0.07 | -0.03 | -0.12 | 0.06 |
| NO$_3^-$ | 0.16 | 0.01 | 0.16 | 0.13 | 0.15 | 0.04 | 0.28* | 0.18 | 0.31* |
| NH$_4^+$ | -0.22 | -0.06 | -0.19 | 0.21 | 0.41*** | 0.23 | -0.06 | 0.33** | -0.03 |
| $\delta^{15}$N-N$_2$O$_{poreair}$ | -0.35*** | 0.14 | 0.12 | -0.03 | -0.48**** | -0.34** | -0.61**** | -0.30** | -0.24 |
| | | | | | *surface* | | | | |
| [N$_2$O$_{emitted}$] | -0.21 | -0.73**** | -0.40* | 0.46*** | 0.77**** | 0.74**** | 0.64**** | -0.11 | 0.27 |
| WFPS | -0.12 | -0.24 | 0.18 | 0.39** | 0.29 | 0.1 | 0.60**** | 0.09 | 0.13 |
| Eh | 0.15 | -0.22 | 0.08 | -0.13 | 0.15 | -0.17 | -0.18 | -0.39 | -0.13 |
| NO$_3^-$ | -0.44** | -0.17 | -0.28 | 0.32 | 0.19 | 0.31 | 0.19 | 0.06 | 0.01 |
| NH$_4^+$ | 0.39* | 0.52** | 0.59** | -0.18 | -0.58** | -0.51** | 0.11 | 0.02 | 0.18 |
| $\delta^{15}$N-N$_2$O$_{emitted}$ | 0.60**** | 0.29 | 0.36 | -0.80**** | -0.33 | -0.44* | -0.53**** | 0.19 | -0.11 |

