# Peer review of "Early season $N_2O$ emissions under variable water management in rice systems: source-partitioning emissions using isotope ratios along a depth profile"

_Biogeosciences, 2018_

## Referee Comment (RC1) · Anonymous Referee #1 · 13 Aug 2018

The study by Verhoeven et al. attempts at partitioning N2O fluxes to the source process groups nitrification/fungal denitrification and (nitrifier)denitrification, under consideration of N2O reduction to N2. Information on the relative contribution of the source processes to total N2O emission is valuable for assessing options for agricultural management that aim at minimizing N losses. In addition and intertwined in the process, the authors present an approach that allows to estimate N2 emission, which is one of the major unknown fluxes in the N cycle. From this perspective, this manuscript is well suited for BG, and an appreciated contribution for the scientific

community.

The study is very detailed, and based on field data, so that I support publication of this manuscript. However, some points should be addressed:

1. One of the objectives is to "semi-quantitatively assess N2O and N2 losses among rice water management treatments". Though this objective is set at prominent position, there is hardly information in form of tables or figures. One would expect such information in view of the objectives.

2. In view of N losses, Crop yields would be very interesting as well. It would probably be wise to add such data in view of objective b

3. The core of the study clearly is the comparison of open and closed system calculations, and their plausibility. The manuscript stops short of clearly presenting and comparing the results of the associated calculations in form of a figure. Such a figure would help the reader to understand why some scenarios were excluded. In addition, the exclusion of open system dynamics could be presented in more detail

4. The supporting information is frequently used in the manuscript, which is ok, but in view of the complex calculations described in section 2.7, I suggest that an example data point is used to show the calculation procedure, and why a sum of squares of 500 was considered meaningful.

5. The authors present calculated Net isotope effects, however the authors are not clear with regard to their assumptions (open/closed system), and the calculation applied violates some basic assumptions of Rayleigh distillation (details below). Though the authors attempt to provide information why the calculated values do not agree with literature isotope effects, the approach is constructed and in my opinion does not bring the manuscript any further. I suggest considering to skip this section.

6. Nutrient concentrations are quite variable. I suggest adding nutrient concentrations and measured fluxes for an appropriate time interval prior to experiment start to show the comparability of the treatments. Please also add seeding dates and all fertilizer applications to the figures 2,3,5 and 6.

See some more detailed comments below.

Title

ok

Abstract

P1L18: please add emissions after N2O

L24: please add and and in front of "fungal denitrification"

Introduction

P2L9: I suggest changing from "biological" to "microbial source processes"

L25: please check the comma after while

P4L4: the "which serves to enrich" construction of the sentence sounds odd to me. What about "The reduction of N2O to N2 enriches the pool of remaining N2O that is measured in 15N and 18O and, thus changes d15N-N2O, d18O-N2O and SP.

L9 onwards: This segment on calculation approaches leaves the reader a little confused. Will there be calculations in the manuscript? Why this segment? Please add an explanatory sentence, or consider skipping this segment. It is also not necessarily true that closed system calculations lead to higher substrate enrichment. This depends very much on the amount of reacted substrate. In general, I am missing some background information: Rice is one of the dominant crops in the world, consumes a tremendous amount of water, even in water-scarce regions, and flooded rice production also contributes greatly to the global methane budget. Saving methane may be counterbalanced by N2O emissions . . . .

Materials and Methods

P5L26: why did the DS treatment receive less fertilizer than the WS treatments? At first glance, this does not make a lot of sense. Please clarify.

P6L15: do I understand correctly that the precision of the GC was +- 12ppb / 24 ppb? This would be a quite low precision, however for the fluxes it may be less severe. Chamber height controls the sensitivity of the chamber so that I suggest giving also a detection limit at, for instance, 0.6 ppm maximum headspace concentration.

P9L11-14: I am not sure if I understand this correctly: is 15N-N2O in this case the isotopic composition in soil water, or in emitted soil air? Please clarify. I suppose, the authors use 15N-N2O in pore water. I don't agree with the authors that this calculation is valid, for the following reasons:

1) 15N-N2O is not necessarily formed from exactly the location of which the nitrate originates, and may have formed from no3- / nh4+ as well.

2) the reaction coordinate is unknown, i.e., there is no knowledge on how much of the nitrate / nh4+ has been transformed. The equation is only valid, if the n2o has formed in an infinitesimal time after consumption of the substrate.

3) there are other possible intermediates in these conversions, all of which obscure this calculation. This needs to be clarified in detail.

P9L19: I am not sure what "Additionally" means in this context. I would assume that for both open and closed systems, two possible scenarios were considered. To clarify this I suggest: "For both the open and closed modeling methods, two possible scenarios were considered..."

P9L25-32: This segment is unclear to me. I guess it is most straightforward to tell my understanding of it, and you clarify in the text: there are 5 publications reporting d18O-N2O for a pure culture experiment during which exclusively N2O was produced, which gives you a good estimate for d18O-N2Oden. You want to add the value measured by Lewicka 2017 to this database (reason remains unclear, I can only encourage mentioning the really careful experiments by Lewicka 2017 as reason to extend the database). However, Lewicka 2017 was corrected for 18O-H2O. Maybe I am right in this assumptions. It became more clear to me after having a look an Figure 1. If so, I suggest you mention Figure 1 in line 22-23, and add 18O-N2Oden, 18O-N2Onit, and the corresponding SP values to figure 1, with an extra tick mark at the corresponding axis, and have the label in the plot region close to the axis. The whole approach may become more clear then. I also suggest not starting with the special case of the 18O-values corrected for water 18O, but start with the general explanation and then describe the detail.

In view of the following text, I don't understand why the orange sc2-line does not cross the sample. For my understanding, this is not correct. Please clarify.

Results

P13L20: from figure 3, this pattern is not obvious for 15N-N2O. Please clarify.

P15L2: Nutrient concentrations are quite different for the treatments. Please add an appropriate time period prior to experiment start to show that initial nutrient concentrations were equal.

P15L28: see comments above on net isotope effects.

Discussion

P18L3-4: The sentence starts with while, it seems like the sentence has not been finished correctly.

P19L11: it is unclear what you mean with a stronger trajectory towards N2O reduction.

P19L22: not clear if the denitrifying microsites are assumed to be more abundant in WS treatments? Please clarify.

P19L24: How can abiotoc N2O formation explain the high SP values greater than 30 in WS-FLD, i.e., the scatter? As you point out, this pathway is associated with SP of 35.

―――――――――――――――――

---

## Referee Comment (RC2) · N. Ostrom (Referee) · 25 Oct 2018

Review of Verhoeven et al. Biogeosciences By Nathaniel E. Ostrom and Jenie Gil Lugo, Michigan State University

This is an impressive manuscript directed toward using natural abundance isotope ratios to evaluate the production and reduction of nitrous oxide in rice agroecosystems in response to water management practices. The manuscript is extremely thorough in terms of its sampling design and extent of measurements but also in the thoroughness

with which the authors review the literature and logic with which they present their arguments. There are relatively few manuscripts that take this care; particularly in the early stages of the submission process. The presentation of both open and closed models is innovative and the explanations of the models are quite clear. We commend the authors for their efforts. There are three significant areas in this manuscript that need to be addressed and a number of minor issues that we list below. First, we appreciate the authors' use of the term "isotopocule" to more accurately describe the bulk and site dependent isotopic composition of nitrous oxide but, regrettably, their use of this term is incorrect (see Ostrom and Ostrom, 2017). Isotopocule is a contraction of "isotopic molecule" and this term refers specifically to the 12 distinct isotopic molecules that result when the two isotopes of nitrogen and 3 isotopes of oxygen are combined in every imaginable way. Thus it is incorrect to use isotopocules to describe isotope ratios. Isotopomer refers to the two isotopocules of nitrous oxide that have the same mass but differ in the location of 15N. Isotopologues is not a very useful term as it implies differences in both mass and isotopic composition. Given this, perhaps it would be best to simply use "isotope ratios" to describe both bulk and site specific isotopic information. Secondly, we are concerned with the use of constant values for the kinetic isotope effects (KIE) associated with nitrous oxide reduction in their models. The literature cited in the paper clearly demonstrates that the KIE associated with nitrous reduction is variable and yet the authors chose a single value of 6.6 per mil in their models. Further, the Jinuntuya-Nortman et al (2008) demonstrate that the KIE decreases with increasing water filled pore space. Third, we are concerned with the use of ranges in $\delta$18O of nitrous oxide associated with various sources of nitrous oxide to describe microbial origins. While SP is considered a conservative tracer of the origins of nitrous oxide it is widely know that bulk $\delta$15N and $\delta$18O values are not conservative. Thus while ranges of values can be compiled from the literature it is uncertain how well these values represent what can be expected in the natural environment. It is known that $\delta$18O values in nitrous oxide can be altered by exchange with water and, indeed, the authors estimate that 100% of the O in nitrous oxide has exchanged with water.

[Figure]

Given this high degree of exchange, how reasonable is it to use constant isotope values to infer microbial origins? We don't believe that any of these concerns should result in rejection or major restructuring of the manuscript. Rather, we would like to see the authors acknowledge these concerns and discuss what the implications of variation in KIE's and source isotope values would have on their model results.

Page 4, line 1-2: Abiotic production of N2O can occur by many pathways and it seems the values cited here reflect production from hydroxylamine. We recently reported SP values of 16 per mil for N2O production from NO (Stanton et al., 2018, Geobiology (DOI :10.1111/gbi.12311).

Page 7, line 10: What are the minimum concentrations required to obtain accurate isotope values for nitrate and ammonium?

Page 9, line 29-32. As mentioned above, this is a good representation of the literature $\delta$18O values but given concerns about water exchange can we realistically expect these values to apply to field studies?

Page 10, line 5: It would seem this slope is determined from a single pair of values when a wide range of values for the KIE associated with nitrous oxide reduction can be found in the literature. What is the impact of variation in the slope on the outcomes of this model?

Page 13, line 22: "In the WS treatments, high N2Oemitted fluxes were also associated with lower $\delta$15N signatures". This statement is not entirely accurate. In WS-AWD two peaks of N2O were observed (Figure 3), firsts on June 17, with high $\delta$15N signatures ($\sim$20‰ and the second on June 23 with lower $\delta$15N signatures ($\sim$40‰, both peaks showing similar N2O flux.

Page 18, lines 18-19: The use of "high" net isotope effects can be misleading because the NIE's are negative. A value of -6, for example, is higher than -16 but reflects a lower degree of isotopic discrimination. Perhaps use "greater degree of isotopic

discrimination" or a similar phrase.

Page 18, line 20: The use of a single value to describe the net isotope effect for reduction of nitrous oxide is not very accurate as it is well known that this value varies. Jinuntuya-Nortman et al. (2008) demonstrated that water filled pore space is inversely related to the net isotope effect and at high values of water filled pore space this value approaches zero. Given that this environment is frequently characterized by high and variable water filled pore space how realistic is it to use a single value? What would be the impact on the model outcomes of allowing this value to vary over the range of literature values reported?

Page 19, Line 25: Authors postulates that high SP values relative to $\delta$18O or $\delta$15N observed in N2O pore air from WS treatments, could be explained by greater contributions from abiotic hydroxylamine decomposition. However, in order to produce enough N2O from abiotic hydroxylamine decomposition, to switch or enriched SP values significantly, it wouldnt require high NH4+ concentrations (Rubasinghege et al., 2011; Heil et al., 2015)? In this study, the NH4+ concentrations were very low during the sampling period.

Page 21, line 13: The finding that oxygen exchange is 100% is very concerning. Doesn't 100% exchange compromise the use of $\delta$18O to partition sources of nitrous oxide?

Figure 4: Is there a reason why the reduction and mixing lines are plotted in A but not on the figures in B?
* * *

---

## Author Comment (AC2) · 10 Nov 2018

Authors: We appreciate this thoughtful review and have added some specific changes to the discussion to address the three main areas of concern. We have also addressed the minor comments. We hope these changes are acceptable and make the discussion more robust and valuable to the N2O isotope community. All page and line references refer to our amended manuscript with track changes all accepted.

Referee: There are three significant areas in this manuscript that need to be addressed

and a number of minor issues that we list below. First, we appreciate the authors'
use of the term "isotopocule" to more accurately describe the bulk and site dependent
isotopic composition of nitrous oxide but, regrettably, their use of this term is incorrect
(see Ostrom and Ostrom, 2017). Isotopocule is a contraction of "isotopic molecule" and
this term refers specifically to the 12 distinct isotopic molecules that result when the two
isotopes of nitrogen and 3 isotopes of oxygen are combined in every imaginable way.
Thus it is incorrect to use isotopocules to describe isotope ratios. Isotopomer refers to
the two isotopocules of nitrous oxide that have the same mass but differ in the location
of 15N. Isotopologues is not a very useful term as it implies differences in both mass
and isotopic composition. Given this, perhaps it would be best to simply use "isotope
ratios" to describe both bulk and site specific isotopic information.

Authors: Thank you for this clarification. We have gone through the manuscript and
changed all 'isotopocule' and 'isotopocule signature' terms to 'isotope ratio' as sug-
gested.

Referee: Secondly, we are concerned with the use of constant values for the kinetic iso-
tope effects (KIE) associated with nitrous oxide reduction in their models. The literature
cited in the paper clearly demonstrates that the KIE associated with nitrous reduction
is variable and yet the authors chose a single value of 6.6 per mil in their models.
Further, the Jinuntuya-Nortman et al (2008) demonstrate that the KIE decreases with
increasing water filled pore space.

Referee: Third, we are concerned with the use of ranges in $\delta$18O of nitrous oxide
associated with various sources of nitrous oxide to describe microbial origins. While
SP is considered a conservative tracer of the origins of nitrous oxide it is widely know
that bulk $\delta$15N and $\delta$18O values are not conservative. Thus while ranges of values can
be compiled from the literature it is uncertain how well these values represent what can
be expected in the natural environment. It is known that $\delta$18O values in nitrous oxide
can be altered by exchange with water and, indeed, the authors estimate that 100% of
the O in nitrous oxide has exchanged with water. Given this high degree of exchange,

how reasonable is it to use constant isotope values to infer microbial origins? We don't believe that any of these concerns should result in rejection or major restructuring of the manuscript. Rather, we would like to see the authors acknowledge these concerns and discuss what the implications of variation in KIE's and source isotope values would have on their model results.

Authors: These are valid points and we agree are worthy of brining into the discussion. First, to clarify, we did not use a KIE value of 6.6 per mil in our models used for source partitioning N2O. Our models for source partitioning N2O relied on SP and 18O only, where we did use fixed isotope effects for 18O and SP during N2O reduction, as referenced in Table 2. The use of -6.6 per mil referred to the 15N isotope effect during N2O reduction, which was only used post source partitioning to evaluate the isotope effects for 15N. Here, we used our modeled N2O reduction fraction, rN2O (derived from 18O and SP model) to back calculate plausible ïĄĎ15NN2O/NO3 values if we assumed a fixed value for 15N N2O reduction fractionation and our rN2O rates. Our intent was to determine if this type of correction could bring our measured ïĄĎ15NN2O/NO3 closer to those seen in pure culture or controlled studies, thus adding support to the extent of N2O reduction measured in our model and helping to explain our measured ïĄĎ15NN2O/NO3.

In general, regarding the use of fixed isotope values and isotope effects in our model, we fully agree there is a lot of room for advancement here. Indeed, when first experimenting with our model we played around with a range of 18O values for denitrification/nitrifier-denitrification and nitrification/fungal denitrification derived N2O as we felt there was the largest range in these values in the literature (Author Response Table 1). We found the patterns between treatments to be pretty conservative but the range variable (Author Response Figure 1). An example of a previous test run is given below. In the end, we felt going in this direction was too complex for this paper and would morph it into a monster and distract from our original intent. We feel strongly though that a logical next step would be to advance the model so that isotope ratios

and effects can be drawn from a pool of literature values using Monte Carlo simulation or a similar approach.

We have added a paragraph to P25L21, which discusses this as well as the need to account for known changes in isotope effects based on environmental conditions in more complex models.

"All modeling attempts to date rely on isotope signatures and effects determined in laboratory studies and thus changes in these values in response to environmental or microbial population dynamics in the field remains a large question. As this was an in-situ field experiment, conditions were not constant across treatments or throughout the sampling time frame, yet it has been shown that isotope effects, particularly for N2O reduction change with shifts in environmental conditions such as increasing water filled pore space (Jinuntuya-Northman et al., 2008). Therefore, the use of fixed isotope effects in our model is a simplification. Future modeling efforts may be improved by the incorporation of variable isotope effects based on soil moisture or O2 for example. Careful, controlled experiments across a range of soils with different management histories are necessary to determine if consistent variation in isotope effects in relation to specific environmental parameters can be determined or if such parameters are site specific. The microbial $\delta$18O signatures for denitrification used in our model were calculated relative to $\delta$18O-H2O. We therefore assumed complete exchange between N2O substrates, intermediaries and water during denitrification. We based this off of previous work showing that O exchange is high and that the isotope effect between water and N2O is relatively stable (Lewicka-Szczebak et al., 2016;Lewicka-Szczebak et al., 2017;Snider et al., 2013;Kool et al., 2007). In reality, results over time and between treatments may have been affected by varying degrees of 18O exchange between N2O, intermediaries and water and by variation in $\delta$18O-H2O values. We recommend that future studies measure the $\delta$18O-H2O to better constrain results. Modeling results would also be more robust if complete $\delta$ 15N -N2O, -NH4+ and –NO3- across treatments and times were available, allowing for complimentary modeling of

SP x 15N(N2O/NO3- or N2O/NH4). Employing iterative simulation techniques where a range of literature values for N2O signatures and isotope effects are used to draw from would help to highlight model sensitivity to specific isotope values and improve its accuracy. Lastly, more work needs to be done to validate results such as those generated here which rely on laboratory derived values, with complimentary measurements of microbial community dynamics, such as that by Snider et al. (2015)."

Referee: Page 4, line 1-2: Abiotic production of N2O can occur by many pathways and it seems the values cited here reflect production from hydroxylamine. We recently reported SP values of 16 per mil for N2O production from NO (Stanton et al., 2018, Geobiology (DOI :10.1111/gbi.12311).

Authors: Clarification of hydroxylamine oxidation specifically and this additional reference have been added, P4L1.

Referee: Page 7, line 10: What are the minimum concentrations required to obtain accurate isotope values for nitrate and ammonium?

Authors: Our limit of quantification for 15N-NH4 was 0.75 mgL-1 or $\sim$ 42uM NH4 , this was accidently omitted, but is now added on P9L8. Our limit of quantification for 15N-NO3- was 0.125 mgL-1 or 2.0 uM NO3- (P9L18).

Referee: Page 9, line 29-32. As mentioned above, this is a good representation of the literature $\delta$18O values but given concerns about water exchange can we realistically expect these values to apply to field studies?

Authors: This is a valid point and we agree. We have tried to better acknowledge that isotope methods such as the modeling proposed here are still limited and difficult to apply and interpret in field situations. At the same time, these methods only become really useful if they can be applied in ecological or agronomic studies. No method is perfect, but we feel that given the current knowledge, the methods can be used for ecological studies as long as the uncertainty associated with data interpretation

is acknowledged. We hope this sentiment is now better expressed in our additional discussion paragraph.

Referee: Page 10, line 5: It would seem this slope is determined from a single pair of values when a wide range of values for the KIE associated with nitrous oxide reduction can be found in the literature. What is the impact of variation in the slope on the outcomes of this model?

Authors: We did not test the sensitivity of our model to changes in this slope. We agree this, among other parameters in the model should be further tested and developed in future studies. See above.

Referee: Page 13, line 22: "In the WS treatments, high N2Oemitted fluxes were also associated with lower $\delta$15N signatures". This statement is not entirely accurate. In WS-AWD two peaks of N2O were observed (Figure 3), firsts on June 17, with high $\delta$15N signatures (âĹij20‰ and the second on June 23 with lower $\delta$15N signatures (âĹij40‰ both peaks showing similar N2O flux.

Authors: The sentence has been amended and now reads, "In the WS treatments, high N2Oemitted fluxes on June 23rd, following the second fertilization, were associated with lower $\delta$15N signatures (Fig. 3), this was not the case for a high flux in the WS-AWD on June 17th."

Referee: Page 18, lines 18-19: The use of "high" net isotope effects can be misleading because the NIE's are negative. A value of -6, for example, is higher than -16 but reflects a lower degree of isotopic discrimination. Perhaps use "greater degree of isotopic discrimination" or a similar phrase.

Authors: This this a good observation and the suggested wording has been adopted.

Referee: Page 18, line 20: The use of a single value to describe the net isotope effect for reduction of nitrous oxide is not very accurate as it is well known that this value varies. Jinuntuya-Nortman et al. (2008) demonstrated that water filled pore space is

inversely related to the net isotope effect and at high values of water filled pore space this value approaches zero. Given that this environment is frequently characterized by high and variable water filled pore space how realistic is it to use a single value? What would be the impact on the model outcomes of allowing this value to vary over the range of literature values reported?

Authors: We feel this point is now addressed in our new discussion paragraph on P25L21. It would be interesting to assess the effect of the model outcomes if this value varied, but we feel this would be too speculative and beyond the scope of the current manuscript.

Referee: Page 19, Line 25: Authors postulates that high SP values relative to $\delta$18O or $\delta$15N observed in N2O pore air from WS treatments, could be explained by greater contributions from abiotic hydroxylamine decomposition. However, in order to produce enough N2O from abiotic hydroxylamine decomposition, to switch or enriched SP values significantly, it wouldnt require high NH4+ concentrations (Rubasinghege et al., 2011; Heil et al., 2015)? In this study, the NH4+ concentrations were very low during the sampling period.

Authors: NH4 concentrations in the WS-AWD prior to the second fertilization were between 5-10 mg N/L and around 5 mg/L N in the WS-FLD and were thus higher than the DS-AWD for much of the sampling period. However, you are correct that the times of higher NH4 in the WS treatments don't necessarily correspond to the scattered high SP values and no correlation between these variables was observed for any treatment (Table 3). The plausibility of abiotic hydroxylamine oxidation during coupled nitrification-denitrification is discussed later in this same paragraph. We have amended the wording some. It now reads as follows. If this whole piece remains too speculative, we can omit. "Abiotic hydroxylamine decomposition requires nitrification for the production of NH2OH, and iron or manganese (hyrdr)oxides as electron acceptors to proceed (Bremner et al., 1980). Given the moist conditions, nitrification rates were likely low in the WS treatments. Feasible co-occurrence of these species could really

only occur directly in the rhizosphere of a flooded rice soil, were O2 is transported to the immediate root zone by the aerenchyma. Tightly coupled nitrification-denitrification in the rhizosphere of rice plants has been shown before (Arth and Frenzel, 2000) as has coupling of nitrogen – iron transformations (Ratering and Schnell, 2000) but we cannot say the extent to which this may have occurred in our system. " P21L2

Referee: Page 21, line 13: The finding that oxygen exchange is 100% is very concerning. Doesn't 100% exchange compromise the use of $\delta$18O to partition sources of nitrous oxide?

Authors: We politely disagree. Our modeling used isotope signatures calculated relative d18O of water for denitrification based on results of (Lewicka-Szczebak et al., 2016;Lewicka-Szczebak et al., 2017). We have added the aforementioned discussion paragraph which we hope adequately addresses this issue.

Referee: Figure 4: Is there a reason why the reduction and mixing lines are plotted in A but not on the figures in B?

Authors: Yes, we did not derive a reduction and mixing line for SP x 15N-N2O relationship. To accurately draw such lines we need to use fixed values for the 15N-N2O signature produced from denitrification and nitrification. We have not reviewed the literature for a consensus value for these processes. Because we had limited data for d15N in NH4+ and NO3, we could not use these values in modeling.

―――――――――――――――――――――――

Author Response Figure 1. Denitrification contribution results for Scenario 1, open system modeling across a range of $\delta_o{}^{18}O\text{-}N_2O_{nit}$ and $\delta_o{}^{18}O\text{-}N_2O_{den}$ values. The range of values used in this testing are given in Table 1. The values actually used in the manuscript results are from "A" (black dots). From this analysis we chose to stick with the values derived from Lewicka-Szczebak et al. 2017 for consistency and because they represented the mean. The ranges changed with varying $\delta_o{}^{18}O$ values but the relative patterns were conserved.

[Figure]

Author response Table 1. $\delta_o{}^{18}O$ values used in model testing.

| Identification | Description | $\delta_o{}^{18}O\text{-}N_2O_{nit}$ | $\delta_o{}^{18}O\text{-}N_2O_{den}$ |
|---|---|---|---|
| A (black dots) | Default values (DF) derived from Lewicka-Szczebak *et al.* (2017) | 36.5 | 12.7 |
| C | $N_2O_{nit}$ fixed, $N_2O_{den}$ +5 | 36.5 | 17.7 |
| D | $N_2O_{nit}$ fixed, $N_2O_{den}$ +10 | 36.5 | 22.7 |
| E | $N_2O_{nit}$ fixed, $N_2O_{den}$ +20 | 36.5 | 32.7 |
| F | $N_2O_{nit}$ fixed, $N_2O_{den}$ -5 | 36.5 | 7.7 |
| G | $N_2O_{nit}$ fixed, $N_2O_{den}$ -10 | 36.5 | 2.7 |
| H | $N_2O_{nit}$ fixed, $N_2O_{den}$ -20 | 36.5 | -7.3 |
| I | $N_2O_{den}$ fixed, $N_2O_{nit}$ +5 | 41.5 | 12.7 |
| J | $N_2O_{den}$ fixed, $N_2O_{nit}$ +10 | 46.5 | 12.7 |
| K | $N_2O_{den}$ fixed, $N_2O_{nit}$ +20 | 56.5 | 12.7 |
| L | $N_2O_{den}$ fixed, $N_2O_{nit}$ -5 | 31.5 | 12.7 |
| M | $N_2O_{den}$ fixed, $N_2O_{nit}$ -10 | 26.5 | 12.7 |
| N | $N_2O_{den}$ fixed, $N_2O_{nit}$ -20 | 16.5 | 12.7 |

**Fig. 1.**

---

## Author Response (AR1)

**Complete authors response to all referee comments**

**Manuscript: bg-2018-254**

**Overall:**

We appreciate the detailed and constructive comments of both reviewers. We have made every attempt to address the larger, more general issues brought up by each referee as well as each minor revision. Referee 1 was concerned with a lack of baseline data and also lost the focus of the paper. We have addressed this by adding some additional data, re-arranging the introduction and adding text to the materials and methods as well as to the conclusions to emphasize paddy history. Referee 2 was primarily concerned with our use of fixed isotope effects in our modeling effort. We have added a paragraph to the discussion that discusses more explicitly the assumptions and limitations of the modeling we did and outlines recommendations for future work in this direction. We believe the manuscript is now both clearer and more scientifically robust and we hope the changes are found acceptable.

Below are our responses to each referee.

*note that page references refer to original version with track changes in place.*

**Author response to referee 1:**

Authors: We appreciate the detailed and constructive comments of reviewer 1. We feel that this review has picked up on many issues we struggled with in presenting this data. Our initial and consistent objective was to use the isotopic data as a tool to conclude more about how the management practices affect processes. However, we simply were unable to collect sufficient, season long data in all three treatments to make more robust comparative agronomic conclusions as relates to N2O and N2 emissions. We feel strongly though that the data collected provides valuable insight into detailed process changes under the different water managements and provides a solid and unique dataset to help push forward the interpretation and use of natural abundance isotope methods. Additionally, as you mention later, we do not have baseline, pre-growing season emissions to show these treatments were similar before the season. Rather, our goal was to collect as much data prior to the first in-season fertilization as possible with the aim of analyzing in detail the response to N fertilization between the treatments, as it turns out, the data collected pre-fertilization was often more interesting. It was not possible to install our equipment prior to 2 days before seeding. In fact the treatments likely did NOT have the same basal emissions because this was the 5th year for each of the treatments under alternative water management. In the first three years these treatments were managed slightly differently, as described in (Miniotti et al., 2016; Peyron et al., 2016). In 2015 and 2016 the WS-AWD water management was adjusted and applied as described in this paper and in (Verhoeven et al., 2018). The DS-AWD was managed as dry-seed + flooding (essentially, delayed permanent flood) for the first 4 years and then adjusted to DS-AWD in for the 2016 year (current publication). Text in the materials and methods has been added to emphasize the paddy history. We have also added a sentence in the conclusions reminding readers of this.

*note, significant changes have been made to the manuscript, the page and line numbers referenced below refer to the revised version without any of the track changes visible.

Detailed individual responses:

**Referee 1: One of the objectives is to "semi-quantitatively assess N2O and N2 losses among rice water management treatments". Though this objective is set at prominent position, there is hardly information in form of tables or figures. One would expect such information in view of the objectives.**

Authors: This is a valid point. We have made a minor change to the phrase referenced above by replacing 'losses' with 'loss rates' to avoid implying that we determined cumulative losses. Indeed, at the onset of this work our aim was to comprehensively compare N2O and N2 losses among the different water treatments. In reality we were unable to obtain high enough fluxes or concentrations of N2O throughout the growing season and across treatments to make isotope measurements at many time points. We realized this in the previous year during a separate, lead up study, therefore in the experiment and dataset presented here we decided it was more valuable to concentrate our efforts and resources on the beginning of the growing season when N2O was higher. Given this we do not feel comfortable to extrapolate our results to growing season emissions. We feel that Fig. 1 and Fig. 6 do quantitatively present N2O emissions for the three treatments during the measurement campaign. We elected not to present a graphic of N2 emissions in the main paper because we felt the data was too patchy for the WS treatments (often the N2O emissions were too low for accurate isotope measurements). In this respect, our method failed. In the original manuscript, we included a graphic of these emissions in the Supplementary material, Fig. S13 C and D. It is labeled 'N2O reduction' rather than N2 production, because it was a calculated N2 production based on N2O reduction from our modeling. We are open to other ideas of graphing or presenting this data, we were just trying not to over-interpret our data and to be transparent about what the data is.

**Referee 1: In view of N losses, Crop yields would be very interesting as well. It would probably be wise to add such data in view of objective b**

Authors: The following data has been included in the text at the beginning of the results section, P14L2

| Treatment | Yield (t/ha) | LSD |
| --- | --- | --- |
| DS-AWD | 6.6 | b |
| WS-FLD | 8.9 | a |
| WS-AWD | 8.2 | a |

The effect of lower N demand in the DS-AWD is also mentioned on P19L22 and on P26L2

**Referee 1: The core of the study clearly is the comparison of open and closed system calculations, and their plausibility. The manuscript stops short of clearly presenting and comparing the results of the associated calculations in form of a figure. Such a figure would help the reader to understand why some scenarios were excluded. In addition, the exclusion of open system dynamics could be presented in more detail**

Authors: We politely disagree that the core of this manuscript was the comparison of open and closed system calculations. We feel that in an uncontrolled environment and using in-situ measurements it is very likely that a mixture of open and closed system dynamics existed. Indeed, we chose to statistically

analyze and discuss only the mean of the two dynamics in our discussion (P24L36).  We have added the following text to our materials and methods as well, to emphasize this, P10L28

"In reality, the heterogeneity in microbial microhabitat within the soil most likely results in a mixture of closed versus open system dynamics. Therefore, final data interpretations were made for the average findings across open versus closed systems dynamics.  "

Further more, Figure 5 does present the results of open and closed system modeling and the mean is indicated by a purple line.  Our data shows that open system modeling consistently led to lower rN2O (= higher reduction) and lower denitrification contributions than closed system (Fig. 5).  Likely, some days and/or treatments were more dominated by one scenario or another, but we cannot say.  Therefore, to maintain equality between the treatments, we took the average of the two dynamics.

There may be some confusion between open and closed system models and then scenario 1 and 2.  These are different, both scenarios were applied to open and closed system models, originally resulting in 4 possible rN2O values.  In scenario 1 we assume that N2O produced by denitrification processes is produced and reduced and then mixed with that of un-reduced N2O.  In scenario 2, we assume that un-reduced N2O from both end member pools is mixed and then reduced.  We found few plausible solutions for scenario 2 (Fig. S3 and Table S2) so decided to eliminate this scenario to simplify the discussion.

**Referee 1: The supporting information is frequently used in the manuscript, which is ok, but in view of the complex calculations described in section 2.7, I suggest that an example data point is used to show the calculation procedure, and why a sum of squares of 500 was considered meaningful.**

Authors: A detailed protocol for the calculation of closed system values can be found on ResearchGate (DOI: 10.13140/RG.2.2.17478.52804).  An example of our open system calculations is now given in a supplementary Excel worksheet.  Both of these materials are now referenced in the text on P11L35 and P13L13, respectively.

Examining our values and their distribution, we chose a sum of squares of 500 as a reasonable value, over which solutions tended to be very implausible, i.e. orders of magnitude out the range of other results for at least one value.  Our search for model solutions was set to minimize the sum of squares between our modeled and observed values, therefore it stands to reason that high sum of square values are associated with less robust model values.  At the time, we felt that evaluating results based on sum of squares for the model as a whole rather an outlier analysis of individual values (i.e. for rN2O, denitrification contribution, etc.) was both more just and simpler.  In retrospect, a more standard method of outlier elimination may have been a better choice.  However, we strongly feel that this would not have resulted in a different outcome.  Between 2.6 – 7.3% of values had a sum of squares over 500 (below).  Over all the sub datasets, 3.4% of values had a sum of square > 500.

[Figure]

**Referee 1: The authors present calculated Net isotope effects, however the authors are not clear with regard to their assumptions (open/closed system), and the calculation applied violates some basic assumptions of Rayleigh distillation (details below).**
**Though the authors attempt to provide information why the calculated values do not agree with literature isotope effects, the approach is constructed and in my opinion does not bring the manuscript any further. I suggest considering to skip this section.**

Authors:  We have corrected our terminology and now refer to our calculated fractionation factors as, $\Delta^{15}N_x$. We fell that retaining these calculations is valid and important.  There is a large body of literature reporting isotope effects, net or otherwise under controlled conditions and also from field studies.  We believe it is important to present and contextualize our data for comparison to past work. We agree that this method has limitations and flaws, indeed one of our goals was to try and push forward the development of new methods that do not rely on 15N values in substrates.   We have changed our notation to η, which is more consistent with the literature for net isotope effects.  We have also added the following text to the materials and methods, P10L18.

"The calculation of $\Delta^{15}N_x$ can be compared to the net isotope effects for nitrification and denitrification derived $N_2O$, as found in the literature.  In reality the processes in equations 1 and 2 entail a series of sequential reactions each of which has a unique isotope effect ($\varepsilon_{k,1}$, $\varepsilon_{k,2}$, $\varepsilon_{k,3}$,...).  It is not possible to measure the isotope values of many of the intermediaries in these reactions series, particularly in in-situ field settings, therefore we report the $\Delta^{15}N_x$.   For the calculation of $\Delta^{15}N_x$ we assume open system dynamics because all measurements were in-situ where substrates, products and intermediaries could be replenished by other processes."

**Referee 1: Nutrient concentrations are quite variable. I suggest adding nutrient concentrations and measured fluxes for an appropriate time interval prior to experiment start to show the comparability of the treatments. Please also add seeding dates and all fertilizer applications to the figures 2,3,5 and 6.**

Authors: See general comments as well.  Unfortunately we do not have data for the time period prior to seeding because we were unable to install equipment until all field preparation and leveling was complete. The data collected during the first 3 weeks of the study, prior to the fertilization, were intended to be our background for understanding treatment response to the fertilization.  We have added information to the materials and methods describing the field history.  P5L29.

**Abstract**
**Referee 1: P1L18: please add emissions after N2O**
Authors: We did not make this amendment because as the sentence is worded, we are referring to both emitted and pore air N2O.  We have moved the position of the ( ) so that it does not break up the sentence in an awkward place, and we hope the sentence is now more clear.

The sentence now reads:  "In a field experiment with three water management treatments, we measured $N_2O$ isotopocule signatures of emitted and pore air $N_2O$ ($\delta^{15}N$, $\delta^{18}O$ and site preference, SP) over the course of six weeks in the early rice growing season.  "

**Referee 1: L24: please add and and in front of "fungal denitrification"** Authors: Completed.

**Introduction**
**Referee 1: P2L9: I suggest changing from "biological" to "microbial source processes".** Authors: Good suggestion, done.

**L25: please check the comma after while**.  Authors: The comma has been removed.

**Referee 1: P4L4: the "which serves to enrich" construction of the sentence sounds odd to me. What about "The reduction of N2O to N2 enriches the pool of remaining N2O that is measured in 15N and 18O and, thus changes d15N-N2O, d18O-N2O and SP.**
Authors: This sentence revision has been adopted.

**Referee 1: L9 onwards: This segment on calculation approaches leaves the reader a little confused. Will there be calculations in the manuscript? Why this segment? Please add an explanatory sentence, or consider skipping this segment. It is also not necessarily true that closed system calculations lead to higher substrate enrichment.**
**This depends very much on the amount of reacted substrate.**

Authors: We have eliminated these two sentences on open versus closed system calculation approaches.

**Referee 1: In general, I am missing some background information: Rice is one of the dominant crops in the world, consumes a tremendous amount of water, even in water-scarce regions, and flooded rice production also contributes greatly to the global methane budget. Saving methane may be counterbalanced by N2O emissions . . . .**

Authors: This is a good point. We have flipped the order of our second and third paragraphs as well as re-arranging some of this now second paragraph, see P2L14-36,P3L1-5. We hope this now better addresses the general background. If needed, we are happy to add more.

**Materials and Methods**
**Referee 1: P5L26: why did the DS treatment receive less fertilizer than the WS treatments? At first glance, this does not make a lot of sense. Please clarify**.

Authors: The three treatments received the same amount of total N per season, 160 kg N/ha. However, N was split applied in three applications designed to maximize NUE based on farm management experience. Our experiment was set up within a larger agronomic trial, which was managed under 'best management practices' for each respective water regime. It is known that rice plant development and growth will be slower under dry seeding, therefore the two WS treatments received N rates of 60, 60 and 40 kg N/ha at fertilizations 1,2 and 3 while the DS treatment received a lower initial rate and then higher subsequent rates: 40, 70 and 50 kg N/ha at fertilizations 1,2 and 3. Fertilization 1 and 2 were covered in the measurement campaign included in this manuscript. We fully acknowledge that this can lead to difficulty in directly comparing the treatments at a given timepoint. On the other hand, it makes the data much more realistic and arguably more comparable as N rate was timed to coarsely align with plant demand so as to minimize the residual N in the soil. This data is given in table 1. We have added a line to this table with the July 14$^{th}$ fertilization and have also added the following sentences to the methods for clarification.

P6L23. "A total of 160 kg N ha$^{-1}$ as urea was applied to all treatments, with one pre-plant application on May 16$^{th}$ and two in-season applications on June 21$^{st}$ and July 14$^{th}$ (Table 1). Following best management practices for the three water management practices, a smaller pre-plant urea application was applied in the DS-AWD treatment, followed by a larger application in this treatment at the second and third fertilization. In the DS-AWD treatment, urea was applied at 40, 70 and 50 kg N ha$^{-1}$, while these rates were 60, 60 and 40 kg N ha$^{-1}$ for the WS treatments at fertilization 1, 2 and 3, respectively."

**Referee 1: P6L15: do I understand correctly that the precision of the GC was +- 12ppb / 24 ppb? This would be a quite low precision, however for the fluxes it may be less severe. Chamber height controls the sensitivity of the chamber so that I suggest giving also a detection limit at, for instance, 0.6 ppm maximum headspace concentration.**

Authors: We do scale our GC detection limit based on the concentration in the sample. The samples in our exetainers are drawn directly from the chamber headspace and are assumed to represent chamber headspace at the moment of sampling. Using 10 reps of at least 5 varied concentration standards we created a regression curve of concentration vs stdev and use this to determine the detection limit for a given concentration. The high and low points on this curve are 300ppb (stdev =12 ppb) and 1000ppb (stdev = 24 ppb) and we chose to give these as examples in the text, P7L19-20. When calculating fluxes, we determined fluxes to be below detection if the change over time was less than the stdev associated with the highest concentration of the 4 measurements. Essentially T4-T1 > stdev of T4. We have added a clarifying sentence to this effect on P7L18

**Referee 1: P9L11-14: I am not sure if I understand this correctly: is 15N-N2O in this case the isotopic composition in soil water, or in emitted soil air? Please clarify. I suppose, the authors use 15N-N2O in pore water. I don't agree with the authors that this calculation is valid, for the following reasons:**

Authors: Neither, the 15N-N2O used in the calculation of net isotope effects was pore air N2O taken at the three depths, the 15N-NO$_3^-$ and 15N-NH4 were analyzed in pore water samples taken at the same depths. Sampling for pore air and pore water occurred within 5 hrs of each other on the same day. We have tried to clarify this in the materials and methods P10L14.

1) **15N-N2O is not necessarily formed from exactly the location of which the nitrate originates, and may have formed from no3- / nh4+ as well.**

Authors: We agree, we feel that this is discussed in section 4.2, P20L33

2) **the reaction coordinate is unknown, i.e., there is no knowledge on how much of the nitrate / nh4+ has been transformed. The equation is only valid, if the n2o has formed in an infinitesimal time after consumption of the substrate.**

Authors: We have modified our terminology and now refer to this value Δ15Nx. Further, we assume open system dynamics for these reactions because refreshing of substrate or consumption of product at any point in time cannot be excluded.

**3) there are other possible intermediates in these conversions, all of which obscure this calculation. This needs to be clarified in detail.**

Authors: See earlier response and amendment to materials and methods.

**Referee 1: P9L19: I am not sure what "Additionally" means in this context. I would assume that for both open and closed systems, two possible scenarios were considered. To clarify this I suggest: "For both the open and closed modeling methods, two possible scenarios were considered. . ."**

Authors: This is good suggestion and this phrasing has been adopted.

**Referee 1: P9L25-32: This segment is unclear to me. I guess it is most straightforward to tell my understanding of it, and you clarify in the text: there are 5 publications reporting d18O-N2O for a pure culture experiment during which exclusively N2O was produced, which gives you a good estimate for d18O-N2Oden. You want to add the value measured by Lewicka 2017 to this database (reason remains unclear, I can only encourage mentioning the really careful experiments by Lewicka 2017 as reason to extend the database). However, Lewicka 2017 was corrected for 18O-H2O. Maybe I am right in this assumptions. It became more clear to me after having a look an Figure 1. If so, I suggest you mention Figure 1 in line 22-23, and add 18O-N2Oden, 18O-N2Onit, and the corresponding SP values to figure 1, with an extra tick mark at the corresponding axis, and have the label in the plot region close to the axis. The whole approach may become more clear then. I also suggest not starting with the special case of the 18O-values corrected for water 18O, but start with the general explanation and then describe the detail.**

Authors: We have made the suggested changes to Fig. 1. We have re-arranged and re-written this section and hope that it is now more clear. The section now reads. P10L30

"A schematic of our closed system approach is given in Fig. 1. For both open and closed methods, two possible scenarios were considered as described by (Lewicka-Szczebak et al., 2017); scenario 1 (sc1), where $N_2O$ is produced and reduced by denitrifiers before mixing with $N_2O$ derived from nitrification or scenario two (sc2) where $N_2O$ is produced from both processes, mixed, and then reduced. In both models, $N_2O$ is originally produced from two possible endmembers; denitrification/nitrifier-denitrification (denoted by subscript den) and nitrification/fungal denitrification (denoted by subscript nit). Our SP endmember values ($SP_{den}$ and $SP_{nit}$) and $N_2O$ reduction fractionation factors ($\varepsilon^{18}O_{red}$ and $\varepsilon SP_{red}$) were taken directly from Lewicka-Szczebak et al. (2017) (Table 2). For $\delta^{18}O$-$N_2O_{(x)}$ endmember values we could not directly use the values reported in Lewicka-Szczebak et al. (2017) because these were reported relative to $\delta^{18}O$-$H_2O$ (as $\delta^{18}O$-$N_2O(N_2O/H_2O)$) and we did not measure the isotope signature of water in our study. Therefore, $\delta^{18}O$-$N_2O_{nit}$ was re-calculated using the original mean values ($\delta^{18}O$-$N_2O$ as opposed to $\delta^{18}O$-$(N_2O/H_2O)$) of the six studies referenced by (Lewicka-Szczebak et al., 2017), this yielded a mean of 36.5‰ (Sutka et al., 2006; Sutka et al., 2008; Frame and Casciotti, 2010; Heil et al., 2014; Rohe et al., 2014; Maeda et al., 2015). For $\delta^{18}O$-$N_2O_{den}$ we adjusted the value used in Lewicka-Szczebak et al. (2017) by an estimate of $\delta^{18}O$-$H_2O$ of water for our site rather than re-calculate from the four studies originally referenced by Lewicka-Szczebak et al. (2017) (Sutka et al., 2006; Frame and Casciotti, 2010; Lewicka-Szczebak et al., 2014; Lewicka-Szczebak et al., 2016). We used a $\delta^{18}O$-$H_2O$ value of -8.3‰, as reported by Rapti-Caputo and Martinelli (2009) for an uncontained aquifer of the Po River delta. We chose to do this because some of the mean values used by Lewicka-Szczebak et al. (2017) were themselves calculated from data originally reported. Our intention was to keep endmember values as consistent as possible between this study and Lewicka-Szczebak et al. (2017). "

**Referee 1: In view of the following text, I don't understand why the orange sc2-line does not cross the sample. For my understanding, this is not correct. Please clarify.**

Authors: You are absolutely correct, thanks! The sample point has been moved up to pass through both intercepts.

**Results**
**Referee 1: P13L20: from figure 3, this pattern is not obvious for 15N-N2O. Please clarify.**

Authors: The sentence has been removed.

**Referee 1: P15L2: Nutrient concentrations are quite different for the treatments. Please add an appropriate time period prior to experiment start to show that initial nutrient concentrations were equal.**

Authors: Please see previous comments.

**Referee 1: P15L28: see comments above on net isotope effects.**

Authors: Please see earlier comment.

**Discussion**

**Referee 1: P18L3-4: The sentence starts with while, it seems like the sentence has not been finished correctly.**

Authors: Well noted. The sentence has been revised to read, "In contrast, saturated conditions favoring complete denitrification certainly prevailed in the WS treatments at times" P19L35

**Referee 1: P19L11: it is unclear what you mean with a stronger trajectory towards N2O reduction.**

Authors: The sentence has been revised to read:
"In both SP x $\delta^{18}$O and SP x $\delta^{15}$N plots our sample values mostly fell between the mixing and reduction lines predicted by either isotope relationship (Fig. 4) and somewhat surprisingly showed stronger enrichment, indicative of greater $N_2O$ reduction in the DS-AWD treatment relative to the WS treatments." P22L3

**Referee 1: P19L22: not clear if the denitrifying microsites are assumed to be more abundant in WS treatments? Please clarify.**

Authors: The sentence has been revised to read:
"More $NO_3^-$ was available for denitrification in the DS-AWD treatment, thus for greater enrichment of this pool to occur we propose that more $NO_3^-$ was trapped in denitrifying microsites as the soil dried or $O_2$ was consumed." P22L14

**Referee 1: P19L24: How can abiotoc N2O formation explain the high SP values greater than 30 in WS-FLD, i.e., the scatter? As you point out, this pathway is associated with SP of 35.**

Authors: This is a valid point. We can only really speculate on these high values. It is plausible that N2O produced by abiotic or fungal denitrifiers was further reduced, enriching the SP value somewhat more. However, we would expect to see enrichment of 18O as well, which wasn't always the case. We believe in part, there is just more error in the WS treatments because we were much more often close to the concentration detection limit of our IRMS, most of the values falling above SP 40 per mil were emitted N2O. We have revised the sentence in question. P22L21

**Author Response to Referee 2 (N. Ostrom and J. Lugo)**

We appreciate this thoughtful review and have added some specific changes to the discussion to address the three main areas of concern. We have also addressed the minor comments. We hope these changes are acceptable and make the discussion more robust and valuable to the N2O isotope community. All page and line references refer to our amended manuscript with track changes all accepted.

**Referee 2: There are three significant areas in this manuscript that need to be addressed and a number of minor issues that we list below. First, we appreciate the authors' use of the term "isotopocule" to more accurately describe the bulk and site dependent isotopic composition of nitrous oxide but, regrettably, their use of this term is incorrect (see Ostrom and Ostrom, 2017). Isotopocule is a contraction of "isotopic molecule" and this term refers specifically to the 12 distinct**

**isotopic molecules that result when the two isotopes of nitrogen and 3 isotopes of oxygen are combined in every imaginable way. Thus it is incorrect to use isotopocules to describe isotope ratios. Isotopomer refers to the two isotopocules of nitrous oxide that have the same mass but differ in the location of 15N. Isotopologues is not a very useful term as it implies differences in both mass and isotopic composition. Given this, perhaps it would be best to simply use "isotope ratios" to describe both bulk and site specific isotopic information.**

Authors:  Thank you for this clarification.  We have gone through the manuscript and changed all 'isotopocule' and 'isotopocule signature' terms to either simply 'isotope' or 'isotope ratio' as suggested.

**Referee 2: Secondly, we are concerned with the use of constant values for the kinetic isotope effects (KIE) associated with nitrous oxide reduction in their models. The literature cited in the paper clearly demonstrates that the KIE associated with nitrous reduction is variable and yet the authors chose a single value of 6.6 per mil in their models. Further, the Jinuntuya-Nortman et al (2008) demonstrate that the KIE decreases with increasing water filled pore space.**

**Referee 2: Third, we are concerned with the use of ranges in δ18O of nitrous oxide associated with various sources of nitrous oxide to describe microbial origins. While SP is considered a conservative tracer of the origins of nitrous oxide it is widely know that bulk δ15N and δ18O values are not conservative. Thus while ranges of values can be compiled from the literature it is uncertain how well these values represent what can be expected in the natural environment. It is known that δ18O values in nitrous oxide can be altered by exchange with water and, indeed, the authors estimate that 100% of the O in nitrous oxide has exchanged with water. Given this high degree of exchange, how reasonable is it to use constant isotope values to infer microbial origins? We don't believe that any of these concerns should result in rejection or major restructuring of the manuscript. Rather, we would like to see the authors acknowledge these concerns and discuss what the implications of variation in KIE's and source isotope values would have on their model results.**

These are valid points and we agree are worthy of brining into the discussion.  First, to clarify, we did not use a KIE value of 6.6 per mil in our models used for source partitioning $N_2O$. Our models for source partitioning N2O relied on SP and 18O only, where we did use fixed isotope effects for 18O and SP during N2O reduction, as referenced in Table 2.  The use of -6.6 per mil referred to the 15N isotope effect during N2O reduction, which was only used post source partitioning to evaluate the isotope effects for 15N.  Here, we used our modeled N2O reduction fraction, rN2O (derived from 18O and SP model) to back calculate plausible $\Delta^{15}N_{N2O/NO3}$ values if we assumed a fixed value for 15N N2O reduction fractionation and our rN2O rates.  Our intent was to determine if this type of correction could bring our measured $\Delta^{15}N_{N2O/NO3}$ closer to those seen in pure culture or controlled studies, thus adding support to the extent of N2O reduction measured in our model and helping to explain our measured $\Delta^{15}N_{N2O/NO3.}$

In general, regarding the use of fixed isotope values and isotope effects in our model, we fully agree there is a lot of room for advancement here.  Indeed, when first experimenting with our model we played around with a range of 18O values for denitrification/nitrifier-denitrification and nitrification/fungal denitrification derived N2O as we felt there was the largest range in these values in the literature (Author Response Table 1).  We found the patterns between treatments to be pretty conservative but the range variable (Author Response Figure 1).  An example of a previous test run is given below.  In the end, we felt going in this direction was too complex for this paper and would morph

it into a monster and distract from our original intent.  We feel strongly though that a logical next step would be to advance the model so that isotope ratios and effects can be drawn from a pool of literature values using Monte Carlo simulation or a similar approach.

We have added a paragraph to P27L11, which discusses this as well as the need to account for known changes in isotope effects based on environmental conditions in more complex models.

*"All modeling attempts to date rely on isotope signatures and effects determined in laboratory studies and thus changes in these values in response to environmental or microbial population dynamics in the field remains a large question.  As this was an in-situ field experiment, conditions were not constant across treatments or throughout the sampling time frame, yet it has been shown that isotope effects, particularly for $N_2O$ reduction change with shifts in environmental conditions such as increasing water filled pore space (Jinuntuya-Northman et al., 2008).  Therefore, the use of fixed isotope effects in our model is a simplification.  Future modeling efforts may be improved by the incorporation of variable isotope effects based on soil moisture or $O_2$ for example.  Careful, controlled experiments across a range of soils with different management histories are necessary to determine if consistent variation in isotope effects in relation to specific environmental parameters can be determined or if such parameters are site specific. The microbial δ18O signatures for denitrification used in our model were calculated relative to δ18O-H2O. We therefore assumed complete exchange between $N_2O$ substrates, intermediaries and water during denitrification.  We based this off of previous work showing that O exchange is high and that the isotope effect between water and $N_2O$ is relatively stable (Lewicka-Szczebak et al., 2016;Lewicka-Szczebak et al., 2017;Snider et al., 2013;Kool et al., 2007).  In reality, results over time and between treatments may have been affected by varying degrees of $^{18}O$ exchange between $N_2O$, intermediaries and water and by variation in δ18O-$H_2O$ values.  We recommend that future studies measure the δ18O-$H_2O$ to better constrain results.  Modeling results would also be more robust if complete $δ^{15}N$ -$N_2O$, -$NH_4^+$ and –$NO_3^-$ across treatments and times were available, allowing for complimentary modeling of SP x $^{15}N(N_2O/NO_3^-$ or $N_2O/NH_4)$.  Employing iterative simulation techniques where a range of literature values for $N_2O$ signatures and isotope effects are used to draw from would help to highlight model sensitivity to specific isotope values and improve its accuracy.  Lastly, more work needs to be done to validate results such as those generated here which rely on laboratory derived values, with complimentary measurements of microbial community dynamics, such as that by Snider et al. (2015)."*

Author Response Figure 1.  Denitrification contribution results for Scenario 1, open system modeling across a range of   $δ_0^{18}O$-$N_2O_{nit}$  and $δ_0^{18}O$-$N_2O_{den}$ values.  The range of values used in this testing are given in Table 1.  The values actually used in the manuscript results are from "A" (black dots).  From this analysis we chose to stick with the values derived from Lewicka-Szczebak et al. 2017 for consistency and because they represented the mean.  The ranges changed with varying $δ_0^{18}O$ values but the relative patterns were conserved.

[Figure]

Author response Table 1. $\delta_0{}^{18}O$ values used in model testing.

| Identification | Description | $\delta_0{}^{18}O\text{-}N_2O_{nit}$ | $\delta_0{}^{18}O\text{-}N_2O_{den}$ |
|---|---|---|---|
| A (black dots) | Default values (DF) derived from Lewicka-Szczebak *et al.* (2017) | 36.5 | 12.7 |
| C | $N_2O_{nit}$ fixed, $N_2O_{den}$ +5 | 36.5 | 17.7 |
| D | $N_2O_{nit}$ fixed, $N_2O_{den}$ +10 | 36.5 | 22.7 |
| E | $N_2O_{nit}$ fixed, $N_2O_{den}$ +20 | 36.5 | 32.7 |
| F | $N_2O_{nit}$ fixed, $N_2O_{den}$ -5 | 36.5 | 7.7 |
| G | $N_2O_{nit}$ fixed, $N_2O_{den}$ -10 | 36.5 | 2.7 |
| H | $N_2O_{nit}$ fixed, $N_2O_{den}$ -20 | 36.5 | -7.3 |
| I | $N_2O_{den}$ fixed, $N_2O_{nit}$ +5 | 41.5 | 12.7 |
| J | $N_2O_{den}$ fixed, $N_2O_{nit}$ +10 | 46.5 | 12.7 |
| K | $N_2O_{den}$ fixed, $N_2O_{nit}$ +20 | 56.5 | 12.7 |
| L | $N_2O_{den}$ fixed, $N_2O_{nit}$ -5 | 31.5 | 12.7 |
| M | $N_2O_{den}$ fixed, $N_2O_{nit}$ -10 | 26.5 | 12.7 |
| N | $N_2O_{den}$ fixed, $N_2O_{nit}$ -20 | 16.5 | 12.7 |

**Referee 2: Page 4, line 1-2: Abiotic production of N2O can occur by many pathways and it seems the values cited here reflect production from hydroxylamine. We recently reported SP values of 16 per mil for N2O production from NO (Stanton et al., 2018, Geobiology (DOI :10.1111/gbi.12311).**

Authors: Clarification of hydroxylamine oxidation specifically and this additional reference have been added, P4L1.

**Referee 2: Page 7, line 10: What are the minimum concentrations required to obtain accurate isotope values for nitrate and ammonium?**

Authors: Our limit of quantification for 15N-NH4 was 0.75 mgL-1 or ~ 42uM NH4 , this was accidently omitted, but is now added on P9L8. Our limit of quantification for 15N-NO$_3^-$ was 0.125 mgL-1 or 2.0 uM NO$_3^-$ (P9L32)

**Referee 2: Page 9, line 29-32. As mentioned above, this is a good representation of the literature δ18O values but given concerns about water exchange can we realistically expect these values to apply to field studies?**

Authors: This is a valid point and we agree. We have tried to better acknowledge that isotope methods such as the modeling proposed here are still limited and difficult to apply and interpret in field situations. At the same time, these methods only become really useful if they can be applied in ecological or agronomic studies. No method is perfect, but we feel that given the current knowledge, the methods can be used for ecological studies as long as the uncertainty associated with data interpretation is acknowledged. We hope this sentiment is now better expressed in our additional discussion paragraph.

**Referee 2: Page 10, line 5: It would seem this slope is determined from a single pair of values when a wide range of values for the KIE associated with nitrous oxide reduction can be found in the literature. What is the impact of variation in the slope on the outcomes of this model?**

Authors: We did not test the sensitivity of our model to changes in this slope. We agree this, among other parameters in the model should be further tested and developed in future studies. See above.

**Referee 2: Page 13, line 22: "In the WS treatments, high N2Oemitted fluxes were also associated with lower δ15N signatures". This statement is not entirely accurate. In WS-AWD two peaks of N2O were observed (Figure 3), firsts on June 17, with high δ15N signatures (~20‰ and the second on June 23 with lower δ15N signatures (~40‰, both peaks showing similar N2O flux.**

Authors: The sentence has been amended and now reads, "In the WS treatments, high N$_2$O$_{emitted}$ fluxes on June 23$^{rd}$, following the second fertilization, were associated with lower δ$^{15}$N signatures (Fig. 3), this was not the case for a high flux in the WS-AWD on June 17$^{th}$."

**Referee 2: Page 18, lines 18-19: The use of "high" net isotope effects can be misleading because the NIE's are negative. A value of -6, for example, is higher than -16 but reflects a lower degree of isotopic discrimination. Perhaps use "greater degree of isotopic discrimination" or a similar phrase.**

Authors: This this a good observation and the suggested wording has been adopted.

**Referee 2: Page 18, line 20: The use of a single value to describe the net isotope effect for reduction of nitrous oxide is not very accurate as it is well known that this value varies. Jinuntuya-Nortman et al. (2008) demonstrated that water filled pore space is inversely related to the net isotope effect and at high values of water filled pore space this value approaches zero. Given that this environment is frequently characterized by high and variable water filled pore space how realistic is it to use a single**

**value? What would be the impact on the model outcomes of allowing this value to vary over the range of literature values reported?**

Authors:  We feel this point is now addressed in our new discussion paragraph on P27L11. It would be interesting to assess the effect of the model outcomes if this value varied, but we feel this would be too speculative and beyond the scope of the current manuscript.

**Referee 2: Page 19, Line 25: Authors postulates that high SP values relative to δ18O or δ15N observed in N2O pore air from WS treatments, could be explained by greater contributions from abiotic hydroxylamine decomposition. However, in order to produce enough N2O from abiotic hydroxylamine decomposition, to switch or enriched SP values significantly, it wouldnt require high NH4+ concentrations (Rubasinghege et al., 2011; Heil et al., 2015)? In this study, the NH4+ concentrations were very low during the sampling period.**

Authors:  NH4 concentrations in the WS-AWD prior to the second fertilization were between 5-10 mg N/L and around 5 mg/L N in the WS-FLD and were thus higher than the DS-AWD for much of the sampling period.  However, you are correct that the times of higher NH4 in the WS treatments don't necessarily correspond to the scattered high SP values and no correlation between these variables was observed for any treatment (Table 3).  The plausibility of abiotic hydroxylamine oxidation during coupled nitrification-denitrification is discussed later in this same paragraph.  We have amended the wording a bit.  It now reads as follows.  If this whole piece remains too speculative, we can omit.

*"Abiotic hydroxylamine decomposition requires nitrification for the production of $NH_2OH$, and iron or manganese (hyrdr)oxides as electron acceptors to proceed (Bremner et al., 1980).  Given the moist conditions, nitrification rates were likely low in the WS treatments.  Feasible co-occurrence of these species could really only occur directly in the rhizosphere of a flooded rice soil, were $O_2$ is transported to the immediate root zone by the aerenchyma.  Tightly coupled nitrification-denitrification in the rhizosphere of rice plants has been shown before (Arth and Frenzel, 2000) as has coupling of nitrogen – iron transformations (Ratering and Schnell, 2000) but we cannot say the extent to which this may have occurred in our system. "  P22L28*

**Referee 2: Page 21, line 13: The finding that oxygen exchange is 100% is very concerning. Doesn't 100% exchange compromise the use of δ18O to partition sources of nitrous oxide?**

Authors:  We politely disagree.  Our modeling used isotope signatures calculated relative d18O of water for denitrification based on results of (Lewicka-Szczebak et al., 2016;Lewicka-Szczebak et al., 2017).  We have added the aforementioned discussion paragraph which we hope adequately addresses this issue.

**Referee 2: Figure 4: Is there a reason why the reduction and mixing lines are plotted in A but not on the figures in B?**

Authors: Yes, we did not derive a reduction and mixing line for SP x 15N-N2O relationship. To accurately draw such lines we need to use fixed values for the 15N-N2O signature produced from denitrification and nitrification.  We have not reviewed the literature for a consensus value for these processes. Because we had limited data for d15N in NH4+ and NO3, we could not use these values in modeling.

[revised manuscript text omitted]
 offor NH$_4^+$ oxidation to N$_2$O and were calculated using equation 2 and 3, respectively.

$$\Delta\varepsilon^{15}N_{N_2O-NO_3} \;=\; \delta^{15}N_{N_2O} - \delta^{15}N_{NO_3} \qquad\qquad (2)$$

$$\Delta\varepsilon^{15}N_{N_2O-NH_4} \;=\; \delta^{15}N_{N_2O} - \delta^{15}N_{NH_4} \qquad\qquad (3)$$

The calculation of $\Delta^{15}$N$_x$ can be compared to the net isotope effects for nitrification and denitrification derived N$_2$O,

as found in the literature. In reality the processes in equations 1 and 2 entail a series of sequential reactions each of

20 which has a unique isotope effect ($\varepsilon_{k,1}$, $\varepsilon_{k,2}$, $\varepsilon_{k,3}$,…). It is not possible to measure the isotope values of many of the intermediaries in these reactions series, particularly in in-situ field settings, therefore we report the $\Delta^{15}$N$_x$. For the calculation of $\Delta^{15}$N$_x$ we assume open system dynamics because all measurements were in situ where substrates, products and intermediaries could be replenished by other processes.

**2.7 Determination of N$_2$O source contribution and N$_2$O reduction**

25 **2.7.1 Two endmember mixing models using SP and $\delta^{18}$O signatures: closed and open systems**

We tested used two mixing models where N$_2$O reduction was modeled under 'open' and 'closed' system dynamics following the theory outlined originally by (Fry, 2007) and (Mariotti et al., 1981), respectively. The two modeling methods are henceforth referred to as 'open' and 'closed'. In reality, the heterogeneity in microbial microhabitat within the soil most likely results in a mixture of closed versus open system dynamics. Therefore, final data interpretations

30 were made for the average findings across open versus closed systems dynamics. A schematic of our closed system approach is given in Fig. 1. For both open and closed methods, two possible scenarios were considered as described by (Lewicka-Szczebak et al., 2017); scenario 1 (sc1), where N$_2$O is produced and reduced by denitrifiers before mixing with N$_2$O derived from nitrification or scenario two (sc2) where N$_2$O is produced from both processes, mixed, and then reduced. In both models, N$_2$O is originally produced from two possible endmembers; denitrification/nitrifier-

35 denitrification (denoted by subscript *den*) and nitrification/fungal denitrification (denoted by subscript *nit*). Our intention was to keep the derivation of endmember values consistent between this study and Lewicka-Szczebak et al.

(2017). Our SP endmember values (SP$_{den}$ and SP$_{nit}$) and N$_2$O reduction fractionation factors ($\epsilon^{18}$O$_{red}$ and $\epsilon$SP$_{red}$) were taken directly from Lewicka-Szczebak et al. (2017) (Table 2). For $\delta^{18}$O-N$_2$O$_{(x)}$ endmember values we could not directly use the values reported in Lewicka-Szczebak et al. (2017) because these were reported relative to $\delta^{18}$O-H$_2$O (as $\delta^{18}$O-N$_2$O(N$_2$O/H$_2$O)) and we did not measure the isotope signature of water in our study. Therefore, $\delta^{18}$O-N$_2$O$_{nit}$

5  was re-calculated using the original mean values ($\delta^{18}$O-N$_2$O as opposed to $\delta^{18}$O-(N$_2$O/H$_2$O)) of the six studies referenced by (Lewicka-Szczebak et al., 2017), this yielded a mean of 36.5‰ (Heil et al., 2014;Sutka et al., 2006;Sutka et al., 2008;Frame and Casciotti, 2010;Rohe et al., 2014;Maeda et al., 2015). For $\delta^{18}$O-N$_2$O$_{den}$ we adjusted the value used in Lewicka-Szczebak et al. (2017) by an estimate of $\delta^{18}$O-H$_2$O of water for our site rather than re-calculate from the four studies originally referenced by Lewicka-Szczebak et al. (2017) (Lewicka-Szczebak et al., 2014;Lewicka-

10  Szczebak et al., 2016;Frame and Casciotti, 2010;Sutka et al., 2006). We used a $\delta^{18}$O-H$_2$O value of -8.3‰, as reported by Rapti-Caputo and Martinelli (2009) for an uncontained aquifer of the Po River delta. We chose to do this because some of the mean values used in calculations by Lewicka-Szczebak et al. (2017) were themselves calculated from data originally reported.

15  ~~In both models, N$_2$O is originally produced from two possible endmembers; denitrification/nitrifier denitrification (denoted by subscript *den*) and nitrification/fungal denitrification (denoted by subscript *nit*). In each model we used identical Lewicka-Szczebak et al. (2017)SP endmember values (SP$_{den}$ and SP$_{nit}$) and N$_2$O reduction isotope effects (εSP$_{red}$ and ε$^{18}$O$_{red}$) as those compiled in (Lewicka-Szczebak et al., 2017) (Table 2). For the $\delta^{18}$O-N$_2$O$_{nit}$ we re-calculated the~~

20

25  ~~reported and used our sample $\delta^{18}$O-N$_2$O values as is and then corrected the denitrification isotope signature, $\delta^{18}$O-N$_2$O(N$_2$O/H$_2$O)$_{den}$, reported by (Lewicka-Szczebak et al., 2017) by an assumed $\delta^{18}$O-H$_2$O of water for our site. We used a $\delta^{18}$O-H$_2$O value of -8.3‰, as reported by (Rapti-Caputo and Martinelli, 2009) for an uncontained aquifer of the Po River delta. For the $\delta^{18}$O-N$_2$O$_{nit}$ we re-calculated the mean from the six studies used in (Lewicka-Szczebak et al., 2017), using the original values reported as $\delta^{18}$O-N$_2$O (as opposed to $\delta^{18}$O-(N$_2$O/H$_2$O), this yielded a mean of~~

30

[revised manuscript text omitted]
$_2$O reduction was the largest contributor to our high net isotope effects.  To check this, we estimated *initial* δ$^{15}$N-N$_2$O values before N$_2$O reduction using our modeled N$_2$O reduction fraction (rN$_2$O), measured δ$^{15}$N-N$_2$O values and a $^{15}$N isotope effect during reduction of -6.6‰ (Denk et al., 2017) in the Rayleigh equation.  We could then estimate amended ε$^{15}$N$_{N2O/NO3}$~~

20   ~~values if N$_2$O reduction effects were accounted for, from the difference between our *initial* δ$^{15}$N-N$_2$O estimates and δ$^{15}$N-NO$_2$-.  These calculations yielded a ε$^{15}$N$_{N2O/NO3}$ from -25.0 to -36.5‰, -32.6 to -42.3‰ and -29.0 to -51.1‰ in the DS-AWD, WS-AWD and WS-FLD across depths (Table S6).  These amended ε$^{15}$N$_{N2O/NO3}$ values do decrease and especially for the WS treatments, come relatively close to literature values for ε$^{15}$N$_{N2O/NO3}$ values during denitrification.  Thus, significant N$_2$O reduction can likely explain much of the high ε$^{15}$N$_{N2O/NO3}$ values observed, particularly in the~~

[revised manuscript text omitted]

| Process(s) | $\delta^{18}$O-N$_2$O$_{(x)}$ | SP$_{(x)}$ | references |
|---|---|---|---|
| denitrification, nitrifier-denitrification | 12.7 | -3.9 | $\delta^{18}$O and SP: Lewicka-Szczebak *et al.* (2017) *$\delta^{18}$O uncorrected for $\delta^{18}$O-H$_2$O |
| nitrification, fungal denitrification | 36.5 | 34.8 | SP: Lewicka-Szczebak *et al.* (2017); $\delta^{18}$O: Sutka *et al.* (2006); Sutka *et al.* (2008); Frame and Casciotti (2010); Heil *et al.* (2014); Rohe *et al.* (2014); Maeda *et al.* (2015) |
|  | $\epsilon^{18}$O$_{red}$ | $\epsilon$SP$_{red}$ |  |
| N$_2$O reduction | -15 | -5 | Lewicka-Szczebak *et al.* (2017) |

5  *Lewicka-Szczebak *et al.* (2017) originally report $\delta_0^{18}$O-N$_2$O(N$_2$O/H$_2$O).  Thus, to calculate a pure $\delta_0^{18}$O-N$_2$O, we added the $\delta^{18}$O-H$_2$O value used in our study, -8.3‰.

**Table 3.** Spearman correlations of N$_2$O$_{emitted}$ with N$_2$O$_{emitted}$ isotope ratios, N$_2$O driving variables and N$_2$O$_{poreair}$ isotope ratios measured at 5 cm in the three water management treatments (WS-FLD = water-seeding + conventional flooding; WS-AWD =

10  water-seeding + alternate wetting and drying; DS-AWD = direct dry seeding + alternate wetting and drying).  Significance indicated by: **** <0.0001, *** < 0.001, **<0.01, *<0.05

| | N$_2$O$_{emitted}$ | | | $\delta^{15}$N-N$_2$O$_{emitted}$ | | | $\delta^{18}$O-N$_2$O$_{emitted}$ | | | $\delta$SP-N$_2$O$_{emitted}$ | | |
|---|---|---|---|---|---|---|---|---|---|---|---|---|
| | WS-FLD | WS-AWD | DS-AWD | WS-FLD | WS-AWD | DS-AWD | WS-FLD | WS-AWD | DS-AWD | WS-FLD | WS-AWD | DS-AWD |
| N$_2$O$_{emitted}$ | | | | -0.16 | 0.03 | -0.51*** | -0.46** | -0.45** | -0.58**** | -0.42* | 0.36* | -0.68**** |
| N$_2$O$_{dissolved,\ 5cm}$ | -0.25 | 0.01 | 0.36 | 0.07 | -0.39* | -0.3 | 0.14 | -0.15 | -0.56* | -0.07 | 0.21 | -0.58* |
| N$_2$O$_{poreair,\ 5cm}$ | 0.00 | -0.05 | 0.48*** | 0.11 | 0.15 | -0.60**** | -0.29 | -0.11 | -0.64**** | -0.3 | -0.32 | -0.64**** |
| WFPS$_{5cm}$ | -0.23 | -0.02 | 0.31* | 0.25 | -0.02 | -0.49*** | -0.09 | -0.29 | -0.50**** | -0.22 | -0.3 | -0.64**** |
| Eh$_{5cm}$ | -0.03 | 0.15 | 0.25 | 0.05 | -0.09 | 0.15 | -0.03 | -0.29 | 0.26 | -0.02 | 0.44* | 0.22 |
| DOC$_{5cm}$ | -0.08 | -0.43** | -0.05 | 0.2 | 0.43** | 0.13 | 0.40* | 0.28 | -0.03 | -0.33 | 0.06 | -0.03 |
| NO$_3$-N$_{porewater,\ 5cm}$ | -0.21 | 0.1 | 0.52*** | -0.25 | -0.29 | -0.64**** | -0.23 | -0.15 | -0.27 | -0.13 | -0.11 | -0.21 |
| NH$_4$-N$_{porewater,\ 5cm}$ | -0.29* | -0.32* | -0.31 | 0.05 | -0.02 | 0.23 | 0.29 | 0.43** | 0.01 | 0.07 | -0.16 | -0.03 |
| $\delta^{15}$N-N$_2$O$_{poreair,\ 5cm}$ | 0.24 | 0.09 | -0.51**** | -0.02 | 0.07 | 0.71**** | 0.1 | -0.24 | 0.64**** | 0.1 | 0.1 | 0.65**** |
| $\delta^{18}$O-N$_2$O$_{poreair,\ 5cm}$ | -0.07 | 0.07 | -0.39** | -0.13 | -0.1 | 0.46*** | 0.02 | -0.03 | 0.48*** | 0.33 | 0.47** | 0.41** |
| $\delta$SP-N$_2$O$_{poreair,\ 5cm}$ | -0.27 | -0.1 | -0.55**** | 0.18 | -0.22 | 0.62**** | 0.14 | 0.21 | 0.49*** | 0.47* | 0.55** | 0.67**** |

**Table 4.** Spearman correlations between $\delta^{15}N\text{-}NO_3^-$ and $\delta^{15}N\text{-}NH_4^+$ with $N_2O_{poreair}$ concentration, $\delta^{15}N\text{-}N_2O_{poreair}$, $NO_3^-$ and $NH_4^+$ concentrations in the three water management treatments (WS-FLD = water-seeding + conventional flooding; WS-AWD = water-seeding + alternate wetting and drying; DS-AWD = direct dry seeding + alternate wetting and drying).

| | $\delta^{15}N\text{-}NO_3^-$ | | | $\delta^{15}N\text{-}NH_4^+$ | | |
|---|---|---|---|---|---|---|
| | DS-AWD | WS-AWD | WS-FLD | DS-AWD | WS-AWD | WS-FLD |
| $\delta^{15}N\text{-}NO_3^-$ | | | | -0.54* | -0.03 | -0.05 |
| $\delta^{15}N\text{-}NH_4^+$ | -0.54* | -0.03 | -0.05 | | | |
| $N_2O_{poreair}$ | 0.34** | 0.07 | 0.38** | -0.72*** | 0.04 | 0.22* |
| $\delta^{15}N\text{-}N_2O_{poreair}$ | 0.00 | 0.00 | -0.14 | 0.46* | -0.03 | 0.14 |
| $NO_3^-$ | -0.66**** | -0.01 | -0.28 | -0.41 | 0.11 | 0.27* |
| $NH_4^+$ | 0.01 | 0.13 | -0.06 | -0.54* | -0.23* | -0.12 |

**Table 5.** ANCOVA results of modeled residual $N_2O$ not reduced (gross $rN_2O$), fraction of total $N_2 + N_2O$ production coming from denitrification (gross frac$_{DEN}$) and the fraction of $N_2O$ attributed to denitrification (DenContribution) derived from $N_2O_{emitted}$ and $N_2O_{poreair}$. The Y position was used a co-variate and represents the longitudinal position of each replicate within field.

| | NumDF | $N_2O_{poreair}$ $rN_2O$-gross | $N_2O_{poreair}$ frac$_{DEN}$ -gross | DenContribution ($N_2O_{poreair}$) | NumDF | $N_2O_{emitted}$ $rN_2O$-gross | $N_2O_{emitted}$ frac$_{DEN}$-gross | DenContribution ($N_2O_{emitted}$) |
|---|---|---|---|---|---|---|---|---|
| treatment | 2 | **0.004** | **<0.001** | 0.188 | 2 | 0.146 | 0.931 | **0.016** |
| day | 14 | **<0.001** | **0.001** | **<0.001** | 16 | **<0.001** | **<0.001** | **<0.001** |
| depth | 1 | **0.019** | **0.007** | **0.008** | | | | |
| Y position | 1 | 0.844 | **0.016** | 0.375 | 1 | 0.451 | 0.373 | 0.818 |
| trmt:day | 28 | **0.001** | **<0.001** | **<0.001** | 19 | **0.009** | **0.024** | **<0.001** |
| trmt:depth | 2 | 0.330 | 0.082 | 0.052 | | | | |
| day:depth | 14 | 0.185 | **<0.001** | **0.002** | | | | |
| trmt:day:depth | 23 | **0.022** | **0.047** | 0.189 | | | | |

**Table 6.** Spearman correlations between modeled $r$N$_2$O-gross, frac$_{DEN}$ –gross and *DenContribution* with soil environmental variables and inorganic N substrates and $\delta^{15}$N-N$_2$O. Results are for the mean of open and closed system dynamics. Subsurface correlations were performed on data aggregated across 5 and 12.5 cm depths. Significance indicated by: **** <0.0001, *** < 0.001, **<0.01, *<0.05

| | frac$_{DEN}$ -gross | | | $r$N$_2$O - gross | | | *DenContribution* | | |
|---|---|---|---|---|---|---|---|---|---|
| | DS-AWD | WS-AWD | WS-FLD | DS-AWD | WS-AWD | WS-FLD | DS-AWD | WS-AWD | WS-FLD |
| | | | | *subsurface* | | | | | |
| [N$_2$O$_{poreair}$] | 0.34*** | 0.2 | 0.31* | 0.01 | 0.60**** | 0.17 | 0.67**** | 0.70**** | 0.59**** |
| WFPS | 0.21* | 0.21* | 0.39** | -0.11 | 0 | -0.06 | 0.34*** | 0.22* | 0.47*** |
| Eh | -0.04 | 0.01 | 0.01 | 0.04 | 0.04 | 0.07 | -0.03 | -0.12 | 0.06 |
| NO$_3^-$ | 0.16 | 0.01 | 0.16 | 0.13 | 0.15 | 0.04 | 0.28* | 0.18 | 0.31* |
| NH$_4^+$ | -0.22 | -0.06 | -0.19 | 0.21 | 0.41*** | 0.23 | -0.06 | 0.33** | -0.03 |
| $\delta^{15}$N-N$_2$O$_{poreair}$ | -0.35*** | 0.14 | 0.12 | -0.03 | -0.48**** | -0.34** | -0.61**** | -0.30** | -0.24 |
| | | | | *surface* | | | | | |
| [N$_2$O$_{emitted}$] | -0.21 | -0.73**** | -0.40* | 0.46*** | 0.77**** | 0.74**** | 0.64**** | -0.11 | 0.27 |
| WFPS | -0.12 | -0.24 | 0.18 | 0.39** | 0.29 | 0.1 | 0.60**** | 0.09 | 0.13 |
| Eh | 0.15 | -0.22 | 0.08 | -0.13 | 0.15 | -0.17 | -0.18 | -0.39 | -0.13 |
| NO$_3^-$ | -0.44** | -0.17 | -0.28 | 0.32 | 0.19 | 0.31 | 0.19 | 0.06 | 0.01 |
| NH$_4^+$ | 0.39* | 0.52** | 0.59** | -0.18 | -0.58** | -0.51** | 0.11 | 0.02 | 0.18 |
| $\delta^{15}$N-N$_2$O$_{emitted}$ | 0.60**** | 0.29 | 0.36 | -0.80**** | -0.33 | -0.44* | -0.53**** | 0.19 | -0.11 |